# m⁶A reader Ythdf proteins control retrotransposon B2 repeat expression and safeguard early embryo development

Ruibao Su [ID][1,11 ✉], Di Gao[2,11], Zongchang Du[3,11], Tie-Gang Meng [ID][1,11], Chao Li[1,11], Lei-Ning Chen[1], Xiaoting Lin[1], Changchang Cao [ID][4], Li-Hua Fan[1], Yanbin Dong[5], Sheng Li[5], Shi-Ming Luo[1], Shuai Jiang[6], Zhong Guo [ID][7], Yulu Tian[8], Qing-Yuan Sun [ID][1,9,10 ✉] & Xiang-Hong Ou [ID][1 ✉]

## Abstract

*N*⁶-methyladenosine (m⁶A) and its binding proteins are critical regulators of gene expression and development in mammals. Despite its extent and importance, the regulatory mechanisms of m⁶A-binding reader proteins Ythdf1, Ythdf2, and Ythdf3 during the early stages of mammalian development remain incompletely understood. Here, we show that Ythdf2 and Ythdf3, but not Ythdf1, are required for early embryo development in mice. Mechanically, we demonstrate that all three Ythdf proteins mediate the decay of their target transcripts by binding to similar m⁶A sites, including maternal mRNAs, mid-preimplantation-activated transcripts, and retrotransposon RNAs. Among these, retrotransposon B2 RNAs emerge as one of the primary targets of Ythdf proteins throughout early embryo development, and deficiency in Ythdf1-3 leads to the accumulation of SINE/B2 RNAs, which, in turn, attenuates RNA polymerase II (Pol II) transcription through *trans*-regulatory mechanisms. In parallel, Ythdf1-3 deficiency represses Pol III-driven B2 transcription, thereby modulating RNA polymerase II activity at genomic regions adjacent to B2 loci via *cis*-regulatory effects. Together, the coordinated regulatory axis of Ythdf-SINE/B2-gene expression governs a broad transcriptional network that is crucial for embryogenesis.

**Keywords** Ythdf1/2/3 Proteins; RNA Decay; SINE/B2 Retrotransposons; RNA Polymerase II Transcription; Early Embryo
**Subject Categories** Development; RNA Biology

## Introduction

*N*⁶-methyladenosine (m⁶A) is one of the most important RNA modifications in the eukaryotic transcriptome (Boccaletto et al, 2018). m⁶A and its associated binding proteins play critical roles in regulating the spatiotemporal gene expression during various developmental processes (Mu et al, 2021; Wei et al, 2022; Wu et al, 2022), including the maternal-to-zygotic (MZT) transition, which involves the timely decay of maternal transcripts and subsequent massive activation of zygotic transcripts (Lee et al, 2014). In mammalian oocytes and embryos, m⁶A is deposited on messenger RNAs, repeat RNAs, and long noncoding RNAs (Mu et al, 2021; Wang et al, 2023; Yao et al, 2023; Zhu et al, 2023), regulated by writer and eraser proteins, affecting transcript fate through reader proteins (Shi et al, 2019). The cytosolic m⁶A reader proteins of the YT521-B homology Domain-containing Family (YTHDF), which include YTHDF1, YTHDF2, and YTHDF3, directly recognize m⁶A sites through the YTH domain (Luo and Tong, 2014). In the classical model, each YTHDF protein has different reported functions: YTHDF1 enhances the translation of its mRNA targets (Wang et al, 2015), YTHDF2 promotes the decay of its mRNA targets (Wang et al, 2014), and YTHDF3 facilitates both translation and degradation of its mRNA targets (Li et al, 2017; Shi et al, 2017). Moreover, YTHDF proteins regulate the translation and degradation of m⁶A-modified mRNAs in a context-dependent manner, wherein O-GlcNAcylation of YTHDF1/3 proteins modulates YTHDF1/3-mediated translation (Chen et al, 2023). However, several reports argue that the three YTHDF proteins act redundantly to mediate the same mRNA degradation, because of the similar binding affinity for m⁶A-containing RNA for all three YTHDF proteins. And yet, the function of YTHDF1 and YTHDF3 in enhancing translation is not detectable, considering the

[1]Guangzhou Key Laboratory of Metabolic Diseases and Reproductive Health, Guangdong-Hong Kong Metabolism & Reproduction Joint Laboratory, Reproductive Medicine Center, the Affiliated Guangdong Second Provincial General Hospital of Jinan University, 510317 Guangzhou, China. [2]Department of Obstetrics and Gynecology, Reproductive Medicine Center, The First Affiliated Hospital of Bengbu Medical University, 233004 Bengbu, China. [3]School of Artificial Intelligence, University of Chinese Academy of Sciences, 100049 Beijing, China. [4]State Key Laboratory of Cardiovascular Disease, Fuwai Hospital, National Center for Cardiovascular Diseases, Chinese Academy of Medical Sciences and Peking Union Medical College, 100037 Beijing, China. [5]Key Laboratory of RNA Biology, Institute of Biophysics, Chinese Academy of Sciences, 100101 Beijing, China. [6]Henan Academy of Innovations in Medical Science, 451163 Zhengzhou, China. [7]Center for Biological Science and Technology, Advanced Institute of Natural Sciences, Beijing Normal University, 519087 Zhuhai, China. [8]Shaanxi Key Laboratory of Earth Surface System and Environmental Carrying Capacity, College of Urban and Environmental Sciences, Northwest University, 710127 Xi'an, China. [9]Key Laboratory of Regenerative Medicine of Ministry of Education, Jinan University, 510632 Guangzhou, China. [10]Department of Developmental Biology, School of Basic Medical Sciences, Southern Medical University, 510515 Guangzhou, China. [11]These authors contributed equally: Ruibao Su, Di Gao, Zongchang Du, Tie-Gang Meng, Chao Li. ✉ E-mail: surb@gd2h.org.cn; sunqy@gd2h.org.cn; ouxh@gd2h.org.cn

redundant function of RNA decay on m⁶A-modified mRNA is dosage-dependent (Kontur et al, 2020; Lasman et al, 2020; Zaccara and Jaffrey, 2020; Zou et al, 2023). Therefore, the precise roles of YTHDF proteins remain under debate.

Approximately half of the mammalian genome is derived from transposon elements (TEs), including three major classes of retrotransposons in mammals: long terminal repeat (LTR) elements, long interspersed nuclear elements (LINEs), and short interspersed nuclear elements (SINEs), which are widely recognized as key drivers of genome evolution due to their ability to rewire gene regulatory networks. In embryonic stem (ES) cells and early embryos, many classes of autonomous TEs are still actively expressed and marked with m⁶A by the METTL3-METTL14 complex (Deng et al, 2014). YTHDC1 binds to the transcripts of IAP, ERVK, and LINE1 retrotransposons in mouse ES cells in an m⁶A-dependent manner, mediating chromatin modulation via H3K9me3 and functional regulation through retrotransposon repression (Liu et al, 2021; Xu et al, 2021). Moreover, methylation of ERV mRNAs mainly acts by reducing the half-life of IAP mRNA with the recruitment of the m⁶A reader YTHDF proteins, thereby providing a protective effect in maintaining cellular integrity by clearing reactive ERV-derived RNA species (Chelmicki et al, 2021). FTO mediates m⁶A demethylation of LINE1 RNA in mouse ES cells and early embryos, regulating LINE1 abundance and the local chromatin state, which in turn modulates transcription of LINE1-containing genes (Wei et al, 2022). In brief, m⁶A modifications on chromosome-associated regulatory RNAs, including promoter-associated RNAs, enhancer RNAs, and repeat RNAs, can tune chromatin state and transcription globally (Liu et al, 2020b).

In mice, conditional knockout (cKO) of *Ythdf2* (*Ythdf2*[HA-Fl/HA-Fl]; *Zp3*-Cre) leads to female-specific infertility, and the embryos mainly arrest at the 2-cell (2C) stage (Ivanova et al, 2017). Both *Ythdf1* KO and *Ythdf3* KO mice are fertile and exhibit normal morphology in their reproductive organs (Lasman et al, 2020). Due to the limited research materials from embryos and technical challenges, the regulatory circuitry mechanisms by which Ythdf proteins regulate the transcriptome after fertilization and impact embryonic development in mammals remain unknown. Here, we demonstrate that all Ythdf proteins mediate the decay of their target retrotransposon RNAs and mRNAs. The coordinated regulatory axis of Ythdf-SINE/B2-gene expression is essential for proper early embryonic development.

## Results

### *Ythdf* knockdown leads to defects in embryonic development

To understand the function and mechanism of Ythdf1/2/3 (hereafter referred to as DF1, DF2, and DF3) as regulators in preimplantation development, we first examined their dynamic mRNA expression patterns in oocytes and early mouse embryos. *Ythdf1/2/3* transcripts exhibit distinct expression patterns, with the expression of *Ythdf3* and *Ythdf2* culminating in the early and middle 2-cell (2C) stages, respectively. Afterward, their expression gradually declines until the blastocyst stage. In contrast, the expression of *Ythdf1* remains consistently low throughout early development, with a slight increase in late 2-cell (L2C) embryos

before gradually declining in 8-cell (8C) embryos (Deng et al, 2014) (Fig. 1A). The translation dynamics of *Ythdf1-3* mirrored their corresponding RNA levels during the early development, except for Ythdf3, whose translational pattern in oocytes appeared to diverge from its RNA expression profile (Xiong et al, 2022) (Fig. 1B). Therefore, DF1/2/3 proteins have a different expression pattern during early embryonic development.

To investigate the impact of perturbing *Ythdf* expression on development, we employed short interfering RNAs (siRNAs) to specifically target endogenous *Ythdf1*, *Ythdf2*, and *Ythdf3*, as well as to deplete all three *Ythdf* transcripts simultaneously. We first confirmed that *Ythdf* siRNAs effectively and specifically knock down *Ythdf1*, *Ythdf2*, and *Ythdf3* expression at both the RNA and protein levels from the L2C stage onwards (Fig. EV1A–C; Appendix Fig. S1). We then monitored the developmental effects of knocking down *Ythdf1*, *Ythdf2*, and *Ythdf3* individually, or all three in combination, on early embryogenesis. This was achieved by microinjecting the corresponding siRNA into two-pronuclear (PN2) zygotes and culturing the embryos in vitro until the blastocyst stage (Fig. 1C). Embryos with triple knockdown of *Ythdf1-3* exhibited a significant developmental delay starting from 2.5 days post-coitus (dpc), resulting in a notable decrease in blastocyst formation (Fig. 1D,E). The embryos appear to develop normally through the 2C stage, successfully undergoing the crucial maternal-to-zygotic transition (MZT), irrespective of individual or combined knockdown of *Ythdf1-3*. This contrasts with reports that conditional knockout (KO) of *Ythdf2* leads to embryo arrest primarily at the 2C stage (Ivanova et al, 2017). We reasoned that DF proteins might still be present before the L2C stage when embryos are treated with DF siRNAs at the PN2 stage, allowing the development of knockdown embryos beyond the 2C stage. Indeed, Immunofluorescence and Western blotting assays showed that DF1 and DF2 proteins remain detectable at the E2C stage despite the dramatic decrease in *Ythdf1* and *Ythdf2* RNA levels when using siRNAs to deplete these transcripts (Fig. EV1D,E; Appendix Fig. S1). Of note, knockdown of *Ythdf2* or *Ythdf3* alone led to a modest yet significant reduction in blastocyst formation, whereas knockdown of Ythdf1 had minimal effects on early embryonic development (Fig. 1D,E). In addition, we observed another abnormal phenotype in *Ythdf* triple knockdown embryos: a notable increase in the number of blastomeres per embryo at 3.0 and 3.5 dpc (Fig. 1F,G).

To rule out potential off-target effects associated with siRNA-mediated *Ythdf* knockdown, we employed both the CRISPR/Cas9 and CasRx systems (Guo et al, 2021; Zuo et al, 2017) to individually or combinatorially knock out or knock down *Ythdfs* in early embryos. Following microinjection, we confirmed that knockout or knockdown of *Ythdf2* or *Ythdf3* alone, or all three *Ythdfs* together, significantly impaired blastocyst formation. In contrast, *Ythdf1* knockout or knockdown had minimal effects on early embryonic development (Figs. 1H–J and EV1F–N). Additionally, we observed a notable increase in the number of blastomeres per embryo at 3.0 dpc in DF1-3 KO embryos (Fig.1K). These findings are consistent with those observed in siRNA-mediated *Ythdf* knockdown experiments, reinforcing the conclusion that the phenotypes are specific and not artifacts of the siRNA-mediated knockdown method.

A previous study demonstrated that knockdown of any individual DF protein in HeLa cells induced compensatory upregulation of the remaining DF paralogs, potentially masking

the phenotypic consequences of single or double knockdowns (Zaccara and Jaffrey, 2020). To investigate whether similar compensatory effects underlie the phenotypic differences observed in our *Ythdf* knockdown embryos, we analyzed the expression of each DF at both the RNA and protein levels following single and double DF knockdowns. Our findings reveal that compensatory effects among *Ythdf*s do not occur at the RNA level before the L2C stage (Fig. EV1D). However, at the protein level, DF2 and DF3 can compensate for the loss of DF1 or DF1/2 in L2C embryos. In contrast, DF1 does not compensate for the loss of DF2, DF3, or

DF2/3, consistent with the developmental defects observed in si*Ythdf2* or si*Ythdf3* knockdowns but not in si*Ythdf1* knockdowns (Fig. EV1E). Moreover, simultaneous knockdown of any two *Ythdf* members resulted in significant developmental delays, beginning at 2.5 dpc (Fig. EV1O–Q). Unexpectedly, these defects were exacerbated rather than rescued by overexpressing an additional DF protein in other *Ythdf* double-knockdown embryos, suggesting that compensatory mechanisms among DF proteins are insufficient to restore normal development. Notably, intrinsic properties of DF proteins, such as their ability to undergo liquid-liquid phase

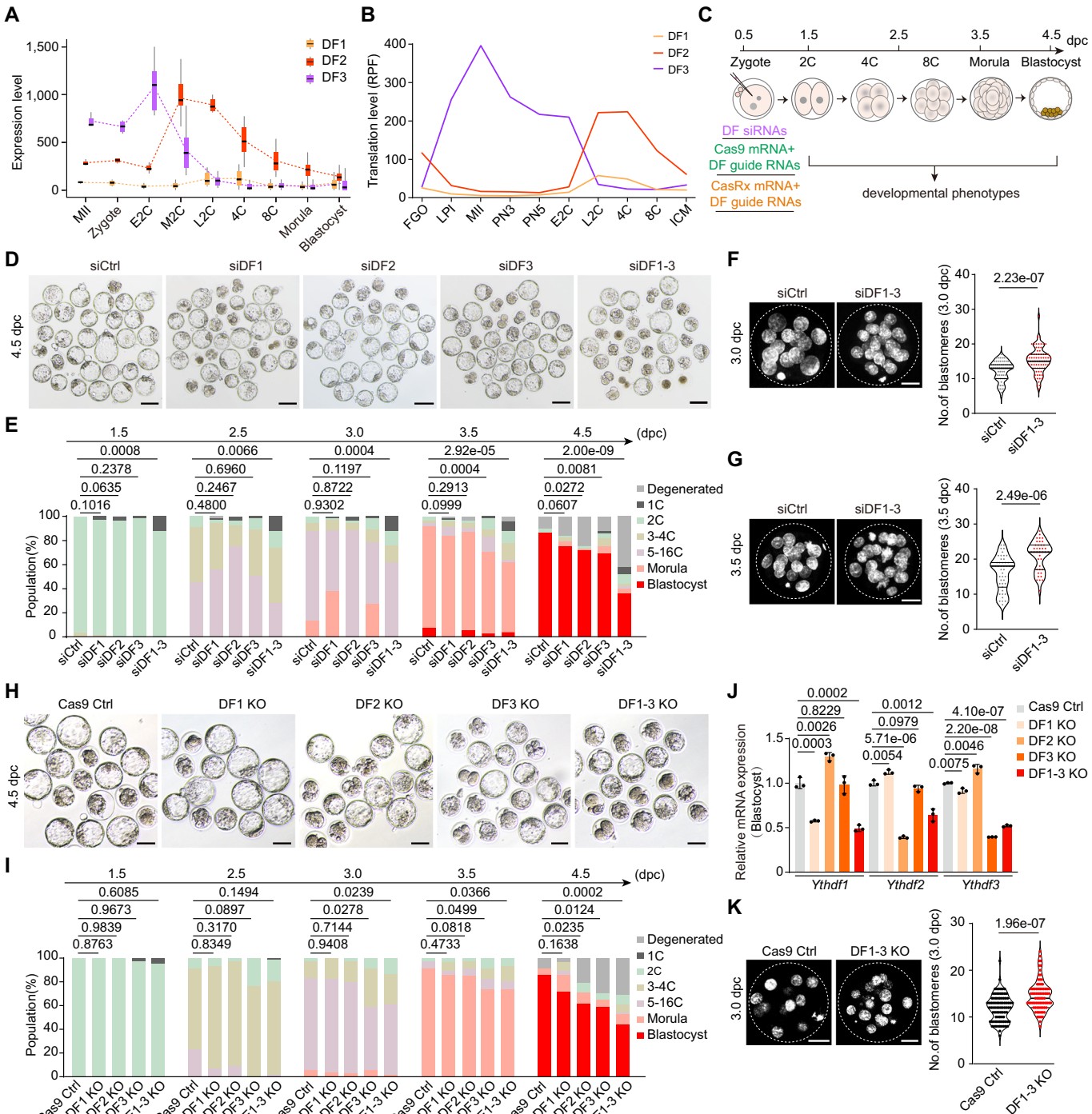

**Figure 1. Developmental effects of *Ythdf* KD or KO.**

(A) Box plots showing *Ythdf1*, *Ythdf2*, and *Ythdf3* mRNA levels in early embryos from a previous publication (Deng et al, 2014). MII oocytes, *n* = 3; Zygote, *n* = 4; Early 2-Cell (E2C), *n* = 8; Middle 2-Cell (M2C), *n* = 12; Late 2-Cell (L2C), *n* = 10; 4-cell (4C), *n* = 14; 8-cell (8C), *n* = 47; Morula (MO), *n* = 58; Blastocyst (BL), *n* = 60. Boxes represent the 25th–75th percentile (line at the median), with whiskers at 1.5× interquartile range (IQR). (B) Line plots showing YTHDF translation levels during early embryo development from a previous publication (Xiong et al, 2022). RPF ribosome-protected fragment. (C) Schematic of the experimental procedure for microinjecting *Ythdf* siRNAs, CRISPR/Cas9 guide RNAs, and CasRx guide RNAs to zygotes and culturing embryos in vitro. (D) Representative images showing embryos treated with *Ythdf* siRNAs or control siRNA at 4.5 dpc. Representative images were selected from two to four independent experiments. Scale bars, 100 μm. (E) Percentages of embryonic stages observed at the indicated time points in siControl (siCtrl, *n* = 90), si*Ythdf1*(siDF1, *n* = 68), si*Ythdf2* (siDF2, *n* = 53), si*Ythdf3* (siDF3, *n* = 65), and si*Ythdf1-3* (siDF1-3, *n* = 50) groups. *P* values were determined by the Chi-square test. (F, G) Representative images of immunofluorescence staining of DAPI (4,6-diamidino-2-phenylindole) in control and *Ythdf1-3* knockdown morula at 3.0 dpc (F) and 3.5 dpc (G). Scale bars, 20 μm. Violin plots showing the total number of blastomeres per embryo in control and *Ythdf1-3* knockdown morulae at 3.0 (siCtrl, *n* = 67; siDF1-3, *n* = 75) and 3.5 (siCtrl, *n* = 47; siDF1-3, *n* = 39) dpc. The upper and lower lines in the violin plots represent upper and lower quartiles (25th and 75th percentiles), and the center line represents the median. (H) Representative images showing embryos treated with *Ythdf* guide RNAs (CRISPR/Cas9) or control guide RNA at 4.5 dpc. Representative images were selected from 2 independent experiments. Scale bars, 100 μm. (I) Percentages of embryonic stages observed at the indicated time points in Cas9 Control (Cas9 Ctrl, *n* = 36), DF1 KO (*n* = 29), DF2 KO (*n* = 35), DF3 KO (*n* = 34), and DF1-3 KO (*n* = 61) groups. *P* values (E, I) were determined by the Chi-square test. (J) Relative expression of *Ythdf1/2/3* quantified by RT-qPCR when single DF or triple DF KO in the blastocyst embryos. Data are mean ± SD, *n* = 3 biological replicates. The *P* value was determined by a two-tailed unpaired *t* test. (K) Representative images of immunofluorescence staining of DAPI in Cas9 control (*n* = 86) and Cas9-mediated DF1-3 KO morula (*n* = 79) at 3.0 dpc. Scale bars, 20 μm. Violin plots showing the total number of blastomeres per embryo in control and DF1-3 KO morulae at 3.0 dpc. The upper and lower lines in the violin plots represent upper and lower quartiles (25th and 75th percentiles), and the center line represents the median. *P* values in (F, G, K) were determined by a two-tailed unpaired *t* test. Source data are available online for this figure.

separation (LLPS), can drive the formation of endogenous LLPS compartments that support compartment-specific mRNA regulation (Fu and Zhuang, 2020; Gao et al, 2019; Ries et al, 2019; Wang et al, 2019). In parallel, post-translational modifications (PTMs), such as glycosylation, modulate protein localization, ligand or partner interactions, and stress granule (SG) assembly (Chen et al, 2023; Sikorski et al, 2023), ultimately shaping the compensatory efficiency of DFs. Together, these findings suggest that DF1, 2, and 3 act in a coordinated manner during early embryogenesis, with each member contributing to developmental progression to varying degrees.

## DF1/2/3 control maternal transcript clearance and 2 C gene expression

To investigate the impacts of DFs on gene expression profiles during embryonic development, we conducted RNA-seq analysis on si*Ythdf* knockdown and control embryos at the L2C and morula stages (Fig. 2A). As expected, knockdown of *Ythdf1*, *Ythdf2*, *Ythdf3*, or all three *Ythdf1-3* in morula-stage embryos resulted in a substantial number of differentially expressed genes (DEGs). In contrast, at the L2C stage, only mild changes in gene expression were observed (Fig. 2B,C; Dataset EV1), which is consistent with the observation that embryos develop normally through the 2-cell stage despite si*Ythdf* knockdown. Notably, RNA-seq analysis confirmed that each *Ythdf* siRNA effectively and specifically depleted its target transcript, as shown in Appendix Fig. S2A,B. Importantly, the targeted *Ythdf* depletion does not affect the expression of the other two *Ythdf* members, further confirming that compensatory effects among DFs do not boost at the RNA level. Additionally, the knockdown of *Ythdf1-3* results in a greater number of DEGs compared to single *Ythdf* knockdown embryos at the morula stage (Fig. 2C), which correlates with the most severe embryonic phenotype observed in si*Ythdf* triple knockdown, marked by the lowest blastocyst development rate (36%). In contrast, at the L2C stage, there is a small amount of increased DEGs in si*Ythdf1-3* knockdown embryos compared to single *Ythdf* knockdown embryos. This suggests that knockdown of any DF protein has minimal impact on the 2C program, thus supporting our

results that residual DF proteins before the L2C stage allow knockdown embryos to develop beyond the 2C stage.

Subsequently, we categorized the early embryonic-expressed genes into 13 clusters based on their expression patterns, including maternal RNA, minor ZGA, major ZGA, mid-preimplantation gene activation (MGA, with expression peaking at the 4–8 cell stage), late-preimplantation gene activation (LGA, with expression peaking at the morula or blastocyst stage), and others (Appendix Fig. S2C; Dataset EV2), using publicly available datasets of the dynamic transcriptome during early development (Wang and Dey, 2006). Among the detected DEGs from si*Ythdf* knockdown embryos, a great number were associated with the clusters of maternal RNA, major ZGA, MGA, and 2-cell transient (Fig. 2D), suggesting that si*Ythdf* knockdown disrupts the normal expression of zygotic and mid-preimplantation activated genes, as well as the clearance of maternal mRNA, contributing to developmental defects. In particular, the upregulated genes in different DF-deficient morulae exhibit similar expression patterns, while the number of unique DEGs identified in each si*Ythdf* knockdown embryo increases from the L2C to the morula stage (Appendix Fig. S2D,E), reflecting the extent of developmental defects associated with individual DF knockdowns. Interestingly, the upregulated genes in the si*Ythdf1-3* triple knockdown morulae include well-known markers of 2C genes, such as *Gm4340*, *Zscan4*, *Zfp352*, and *Tdpoz2/4*, along with the 2C-specific transposon MERVL (murine endogenous retrovirus-L) (Macfarlan et al, 2012) (Fig. 2E; Appendix Fig. S2F). Consistently, both Cas9-mediated DF1-3 KO and CasRx-mediated DF1-3 KD blastocysts exhibited upregulation of 2C genes and MERVL (Appendix Fig. S2G,H). This likely reflects a delayed clearance of 2C-stage transcripts, suggesting that DF1-3 triple-deficient embryos retain a 2C/totipotent-like transcriptomic profile even at the LGA stage. In addition, double knockdown of any two *Ythdf* members also results in varying degrees of upregulation of 2C genes, comparable to triple knockdown of *Ythdf1-3* (Fig. 2E; Appendix Fig. S2I). Consistently, single si*Ythdf* knockdowns lead to a reduction in both the number and expression levels of the 2C genes upregulated in si*Ythdf1-3* triple knockdown morulae (Appendix Fig. S2J,K), suggesting that the

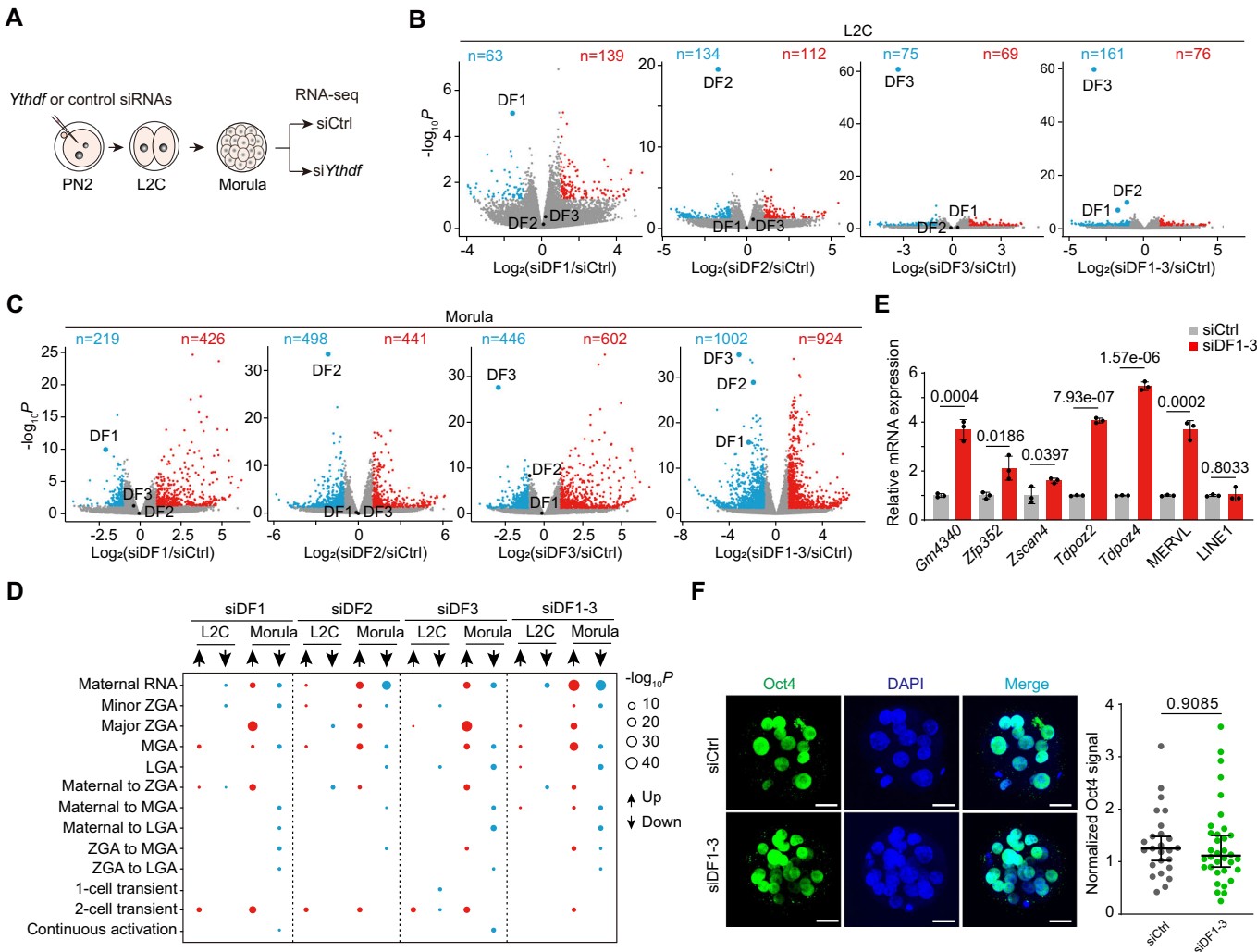

**Figure 2. Transcriptomic effects of si*Ythdf* knockdown.**

(A) Schematic of the experimental procedure for performing RNA-seq at L2C and morula. (B, C) Volcano plots showing gene expression changes upon si*Ythdf* knockdown at the L2C (B) and morula (C) stages (3 biological replicates for each knockdown condition). The P value was determined using DESeq2 (Love et al, 2014), with a threshold of 0.05. (D) Bubble plot showing the overlaps between all DEGs in *Ythdf* knockdown embryos and the expression patterns of the developmental transcriptome, based on datasets from previous publications (Deng et al, 2014). The bubble plot sizes show the $-\log_{10}[P$ values] derived from Fisher's exact test. (E) RT-qPCR validation of upregulated 2C genes in si*Ythdf* knockdown morula embryos. Data are mean ± SD, $n = 3$ biological replicates. The P value was determined by a two-tailed unpaired t test. (F) Representative images of immunofluorescence staining of OCT4 with DAPI (4,6-diamidino-2-phenylindole) counterstain in control and si*Ythdf1-3* knockdown morula at 3.5 dpc. Scale bars, 20 μm. Dot plots showing the relative intensity for OCT4 in control ($n = 25$) and si*Ythdf1-3* knockdown morula ($n = 34$) at 3.5 dpc, normalized to the DAPI signal. The error bar represents the mean ± SD. P value was determined by a two-tailed unpaired t test. Source data are available online for this figure.

increase in 2C gene transcripts correlates positively with the severity of the developmental phenotype observed in si*Ythdf* knockdown embryos.

Next, we were interested in exploring the potential impact of DF proteins on early cell lineage specification during preimplantation development, considering the report that m⁶A-marked mRNA facilitates the process of naïve pluripotency toward differentiation from post-implantation E5.5 embryos (Geula et al, 2015). To address this, we examined mRNA and protein levels of genes related to the inner cell mass (ICM), such as *Oct4* (also known as *Pou5f1*) and *Nanog*, as well as trophectoderm differentiation-related gene (*Cdx2*), in *Ythdf1-3* triple knockdown embryos at 3.5 dpc. As expected, there is no significant difference in the mRNA

and protein levels of Oct4, Nanog, and Cdx2 between si*Ythdf1-3* KD embryos and the control group (Fig. 2F; Appendix Fig. S2L–O), suggesting that DF proteins do not play a major role in early embryonic lineage differentiation. This observation is consistent with previous reports showing that Mettl3 KO blastocysts at E3.5 exhibit normal morphology and expression of pluripotency markers, and that *Ythdf* triple KO embryos develop normally until E7.5 but fail to progress beyond that stage (Geula et al, 2015; Lasman et al, 2020). Gene ontology analysis of the DEGs identified from the si*Ythdf1-3* knockdown morula shows that these genes are predominantly involved in crucial pathways, including regulation of cell adhesion, phosphate metabolic process, and cell development (Appendix Fig. S2P,Q).

## Phenotypes observed in *Ythdf1-3* knockdown embryos are associated with m⁶A marks

Considering that DF proteins regulate transcripts by recognizing the m⁶A marks on RNAs, we sought to investigate whether the phenotype caused by si*Ythdf* knockdown is associated with the m⁶A marks. To this end, we employed a METTL3 inhibitor (STM2457) to treat embryos (Wu et al, 2022), which greatly reduced m⁶A levels and blastocyst formation (Fig. EV2A–C). Next, we conducted RNA-seq analysis on STM2457-treated and control embryos at the L2C and morula stages (Fig. EV2D–F). Specifically, STM2457-treated embryos exhibited marked upregulation of 2C genes and MERVL (Fig. EV2G). Of note, the expression of pluripotency and differentiation-associated genes remained unchanged, in line with the report that Mettl3 KO blastocysts at E3.5 display normal morphology and expression of pluripotency markers and do not exhibit defects in early embryonic lineage differentiation (Geula et al, 2015) (Fig. EV2H; Dataset EV3). Unexpectedly, there was a notable increase in blastomere number per embryo at 3.0 and 3.5 dpc (Fig. EV2I), consistent with the characteristics observed in si*Ythdf1-3* knockdown embryos. Moreover, the development defects resulting from *Ythdf1-3* KD can be largely rescued by STM2457 treatment (Fig. EV2J–L). These findings indicate that the phenotype of si*Ythdf1-3* knockdown is dependent on m⁶A marks. We further analyzed the overlap of DEGs between si*Ythdf1-3* knockdown and STM2457-treated embryos. The shared upregulated genes include well-known 2C markers such as *Gm8994*, *Tdpoz2*, and *Tdpoz4*, whereas the shared downregulated genes, including *Trp53*, *Notch1*, and *Trp73* (Fig. EV2M), are enriched in the p53 signaling pathway (Fig. EV2N). These findings suggest a potential mechanism by which *Ythdf1-3* knockdown promotes enhanced cell proliferation and accelerated blastomere division in early embryos.

## DF1/2/3 proteins bind similar m⁶A sites

Although the precise roles of DF proteins are still under discussion in various cell lines and organisms, we aim to identify the binding targets and potential roles of DF1/2/3 proteins during early development. To address this, we utilized the LACE-seq method (Su et al, 2021), previously developed by our group for efficient capture of RNA-binding protein targets from limited starting materials, to investigate the binding profiles of DF1, DF2, and DF3 in E2C, L2C, and morula embryos (Figs. 3A and EV3A). To ensure rigorous control over our findings, we conducted DF1/2/3 LACE-seq to explore their binding profiles in both control and STM2457-treated embryos across these three early developmental stages (Table EV2). The binding of DF1/2/3 is dramatically decreased in STM2457-treated embryos, characterized by a preference for binding the m⁶A classic motif DRACH (D = G/A/T, R = A/G, H = A/T/C) (Fig. EV3B,C), indicating that the binding of DF1/2/3 is dependent on m⁶A modification.

Next, we utilized DF1/2/3 LACE-seq datasets from STM2457-treated embryos as negative controls to mitigate potential background signals in DF1/2/3 binding observed in corresponding control embryos. Furthermore, to identify high-confidence m⁶A-dependent DF targets for downstream functional characterization, we rigorously filtered DF binding peaks for the presence of the canonical m⁶A consensus motif. After filtering out noise signals,

we observed a dynamic distribution of DF1/2/3-bound peaks across the genome, with enrichment around mRNA stop codons observed for all three m⁶A readers across all stages (Fig. EV3D,E), consistent with the binding features of DF proteins in cell lines (Patil et al, 2016; Shi et al, 2017; Wang et al, 2014; Wang et al, 2015). It is noteworthy that the DF1/2/3 LACE-seq libraries were generated using total RNA from cell lysates, thereby enabling the detection of DF1/2/3 binding events across exonic, intronic, and intergenic regions. This is consistent with previous findings showing that total RNA samples, as opposed to poly(A)⁺ RNA, contain a greater proportion of m⁶A sites within promoter, intronic, and intergenic regions (Linder et al, 2015). A major contributor to the observed DF1/2/3 enrichment in intronic and intergenic regions is the abundant expression of retrotransposon RNAs; notably, 20%–44% of DF1/2/3 binding peaks are associated with transposable elements (TEs) situated within intronic and intergenic regions (Fig. EV3F,G). We identified dynamic binding peaks of DF1/2/3 proteins during early development (Fig. EV3H; Dataset EV4), characterized by an increase in DF2/3 binding peaks and a concomitant decrease in DF1 binding peaks from the E2C to morula stages.

At the non-intronic gene level, we observed a reciprocal overlap of DF targets between two DF proteins, ranging from 23.6 to 72.2% (Fig. 3B–D). Among the DF-bound genes in 2C and morula embryos, a great number are associated with maternal RNA and MGA genes (Fig. 3E). This suggests that DF proteins predominantly regulate maternal mRNA and MGA genes, rather than ZGA genes, even though numerous major ZGA genes are significantly upregulated in DF-deficient morulae (Fig. 2D). Furthermore, we confirmed a substantial overlap of peaks and transcripts between DF-bound coding RNAs and m⁶A-occupied coding RNAs in 2C embryos (Wang et al, 2023) (Appendix Fig. S3A,B), implying that the DF binding to transcripts is closely associated with m⁶A modification, as exemplified by the m⁶A-modified regions of *N4bp1* in 2C embryos (Fig. 3F).

To explore whether DF1/2/3 proteins bind to overlapping m⁶A sites, we compared DF1/2/3 LACE-seq binding signals at m⁶A peaks identified from the picoMeRIP-seq datasets (Wang et al, 2023). DF proteins exhibited a high degree of correlation with each other over m⁶A peaks at individual stages (E2C, L2C, or Morula) or across overall early developmental stages (combined DF LACE-seq data from the E2C, L2C, and morula stages), with Pearson correlation coefficients ranging from $R = 0.54$ to $0.91$ (Fig. 3G), suggesting that DF1/2/3 largely bind to overlapping sets of m⁶A sites during early embryogenesis. Collectively, these results suggest that DF proteins recognize similar m⁶A sites at each stage of early embryonic development.

## DF proteins promote targeted mRNA decay

Given the ongoing debate regarding the cellular functions of DF1/2/3 proteins, particularly whether DF1, like DF2 and DF3, contributes to transcript decay, we investigated whether these DF proteins mediate targeted mRNA degradation to regulate transcript decay during early embryonic development, with a specific focus on DF1. To achieve this, we conducted RNA-seq on L2C embryos with *Ythdf1*, *Ythdf2*, or *Ythdf3* knockdown, respectively, accompanied by 5,6-dichloro-1-β-d-ribofuranosylbenzimidazole (DRB) treatment of these knockdown and control embryos from the PN2 one-cell stage, which inhibits RNA polymerase II (Pol II) transcription elongation

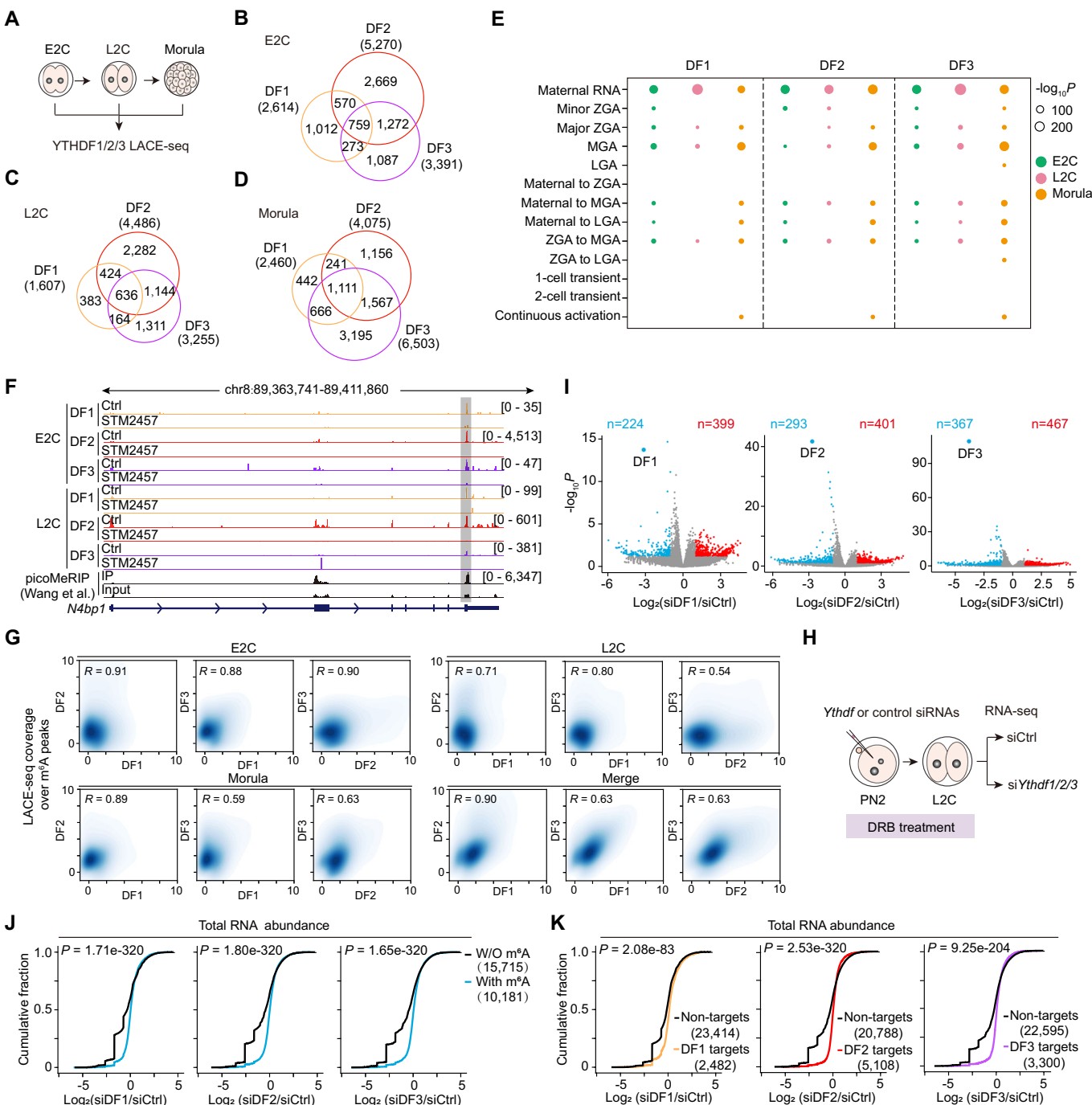

**F** chr8:89,363,741-89,411,860

picoMeRIP (Wang et al.)

during minor and major ZGA (Abe et al, 2018) (Fig. 3H; Appendix Fig. S3C). As expected, the knockdown of *Ythdf1*, *Ythdf2*, and *Ythdf3* in DRB-treated embryos led to a greater number of DEGs compared to embryos where *Ythdf* knockdown occurred without DRB treatment, particularly among the subsets of upregulated genes (Figs. 2B and 3I; Appendix Fig. S3D; Dataset EV5), indicating that DRB treatment effectively mitigates the interference of Pol II transcription in DF-mediated RNA decay. Consequently, a cluster of genes shows extended lifespan in si*Ythdf* knockdown embryos treated with DRB. Thus, these data strongly indicate that each DF

protein promotes mRNA decay at the 2C stage, with individual DF proteins regulating partially distinct subsets of target genes (Appendix Fig. S3E).

Then, we categorized the evaluated transcripts based on m⁶A peaks inferred from the picoMeRIP-seq dataset (Wang et al, 2023) in 2C embryos into two groups: m⁶A-unmodified and m⁶A-modified transcripts. Compared to m⁶A-unmodified transcripts, a noticeable increase in the abundance of m⁶A-modified transcripts was observed in si*Ythdf1*, si*Ythdf2*, and si*Ythdf3* embryos subjected to DRB treatment, respectively (Fig. 3J), suggesting that DF1, similar to DF2 and DF3, plays

Figure 3. DF1/2/3 proteins promote transcript decay.

(A) Schematic of the experimental procedure for performing DF1/2/3 LACE-seq in the E2C, L2C, and morula embryos. (B–D) Venn diagram showing the overlap of target genes identified in DF1/2/3 LACE-seq replicates at the E2C (B), L2C (C), and morula (D) stages. (E) Bubble plot showing the overlap between DF-binding genes and the expression patterns of the developmental transcriptome, based on datasets from previous publications (Deng et al, 2014). The bubble plot sizes show the −log10[*P* values] derived from Fisher's exact test. (F) Genome browser snapshot showing the read coverage of *N4bp1* in DF1/2/3 LACE-seq datasets and the picoMeRIP-seq dataset (Wang et al, 2023). (G) Density plots showing the pairwise comparison of normalized Ythdf1/2/3 binding signals (LACE-seq reads) surrounding individual m6A peaks (picoMeRIP-seq datasets from 2C and 8C embryos) across the E2C, L2C, morula, and combined developmental stages. The DF1/2/3 LACE-seq datasets from E2C and L2C embryos were referenced against the m6A peaks identified in the 2C picoMeRIP-seq datasets; the DF1/2/3 LACE-seq datasets from morula embryos were referenced against the m6A peaks identified in the 8C picoMeRIP-seq datasets. The combined DF1/2/3 LACE-seq datasets from E2C, L2C, and morula embryos were compared with the combined m6A peaks from the 2C and 8C picoMeRIP-seq datasets. (H) Schematic of the experimental procedure for detecting the transcriptome stability of DF-depleted embryos treated with DRB (5,6-dichloro-1-β-d-ribofuranosylbenzimidazole). (I) Volcano plot showing changes in transcript stability upon knockdown of *Ythdf* expression in DRB-treated embryos at the L2C stage (3 biological replicates for each knockdown condition). The *P* value was determined using DESeq2(Love et al, 2014), with a threshold of 0.05. (J) Cumulative distributions of the log2 fold changes in RNA abundance between si*Ythdf* and siControl for m6A-modified and unmodified RNAs in DRB-treated L2C embryos, m6A-modified datasets from previous publications (Wang et al, 2023). *P* values were determined by the Kolmogorov–Smirnov test. (K) Cumulative distributions of the log2 fold changes in RNA abundance between si*Ythdf* and siControl for non-targets and DF LACE-seq targets at the E2C stages. *P* values were determined by the Kolmogorov–Smirnov test.

a role in facilitating the decay of m6A-modified transcripts. Subsequently, we classified 2C transcripts into non-target RNAs and mRNAs occupied by DF1, DF2, or DF3 based on binding peaks defined by DF1/2/3 LACE-seq datasets from E2C and L2C embryos, respectively. Compared to corresponding non-target RNAs, dramatic increases in mRNA stabilization of DF1/2/3-binding RNAs were observed in si*Ythdf1*, si*Ythdf2*, and si*Ythdf3* embryos following DRB treatment, respectively (Fig. 3K; Appendix Fig. S3F), suggesting that DF1, along with DF2 and DF3, facilitates the decay of its target transcripts by recognizing m6A-modified sites. Conservation analysis revealed that the mouse DF1 protein shows higher similarity to mouse DF2 and human DF2 proteins, rather than human DF1 protein, despite the similarity calculations between two mouse DF proteins falling within the range of 65.87–68.37% (Appendix Fig. S3G,H), indicating that the function of the mouse DF1 protein is more akin to mouse DF2 and human DF2. Thus, these results demonstrate that DF1/2/3 proteins facilitate RNA decay with binding to m6A-marked transcripts in the 2C embryos.

## DF1/2/3 proteins promote B2 RNA decay

During early embryo development, numerous retrotransposons are activated and involved in developmental regulation (Guo et al, 2023; Percharde et al, 2018; Sakashita et al, 2023). We wondered whether DF proteins could directly bind to and regulate the transcripts of active retrotransposons during early embryonic development, as previous reports have shown a high abundance of m6A modification on LINE1, LTR, and SINE transcripts (Wang et al, 2023; Wu et al, 2022). We first examined the DF1/2/3 binding on the major subfamilies of transposable elements (TEs) and the expression changes of these TEs in embryos with si*Ythdf1-3* knockdown. Although DF1/2/3 proteins broadly target retrotransposon RNAs, B2 RNAs, as well as the 2C-specific retrotransposon MERVL, were markedly upregulated in si*Ythdf1-3* knockdown morula embryos (Fig. 4A,B). We also observed an increase in B2 RNA levels in DF1-3 KO mESCs (Lasman et al, 2020) and STM2457-treated embryos, but not in DF1-3 KD HeLa cells (Zou et al, 2023) (Fig. EV4A), consistent with the fact that B2 elements are mouse-specific. In contrast, Alu elements are their closest counterparts in humans (Mariner et al, 2008). These findings suggest that DF-mediated regulation of B2 elements may be integral to the mESC and mouse early embryo context. The colocalization of immunofluorescence and fluorescent in situ RNA

hybridization (IF/RNA-FISH) and quantitative reverse transcription polymerase chain reaction (RT-qPCR) results confirmed that B2 RNAs are greatly increased in *Ythdf*1-3 knockdown embryos (Fig. EV4B–F). Additionally, the binding signals of each DF protein were well-localized to the m6A-marked regions on B2 loci in early embryos (Figs. 4C and EV4B,D). Subclasses of the B2 family, derived from distinct genomic loci, exhibited high levels of m6A methylation (Fig. 4D). Furthermore, the consensus sequences surrounding the putative m6A sites conformed to the canonical DRACH motif (Fig. 4D), as evidenced by published GLORI datasets from mESCs, supporting the notion that DF proteins target B2 RNAs in an m6A-dependent manner.

SINE elements are highly abundant retrotransposons in the mammalian genome, and B2 is a distinct subfamily of SINEs dispersed throughout the rodent genome. its expression is highly observed in germ cells, early embryos, and cultured somatic cells induced by specific cellular stresses (Allen et al, 2004; Bachvarova, 1988; Mangiavacchi et al, 2021). During early embryogenesis, the expression level of B2 RNA declines at the E2C stage and then rapidly increases, remaining at a higher level until the blastocyst stage, distinct from the expression pattern of MERVL (Fig. EV4G), suggesting B2 RNA is rapidly cleared before ZGA, and then reactivated expression along with ZGA occurs. Of note, the B2_Mm1a is the most highly expressed B2 RNA subfamily at each developmental stage (Fig. EV4H), and we designed B2 antisense oligonucleotides (ASOs) targeting B2_Mm1a to knock down B2 RNAs for further study (Table EV1). Then, we observed that B2 RNA exhibited nuclear enrichment with cytoplasmic distribution up to the morula stage and became evenly distributed between the nucleus and cytoplasm at the blastocyst stage (Fig. 4E). These results raise the possibility that nuclear B2 transcripts may play roles as regulators in gene expression and cytosol B2 transcripts may hijack the LINE-encoded proteins to generate and integrate a cDNA copy of the SINE RNA into the genome or influence mRNA translation (Elbarbary et al, 2016).

To shed light on the functional effects of DF proteins on B2 RNAs, we classified B2 transcripts according to their m6A modification and DF binding in 2C embryos. As expected, we observed a significant increase in the stabilization of both m6A-modified and DF-bound B2 transcripts compared to other B2 transcripts in si*Ythdf* knockdown embryos following DRB treatment (Fig. 4F,G). Accordingly, the increased stabilization of

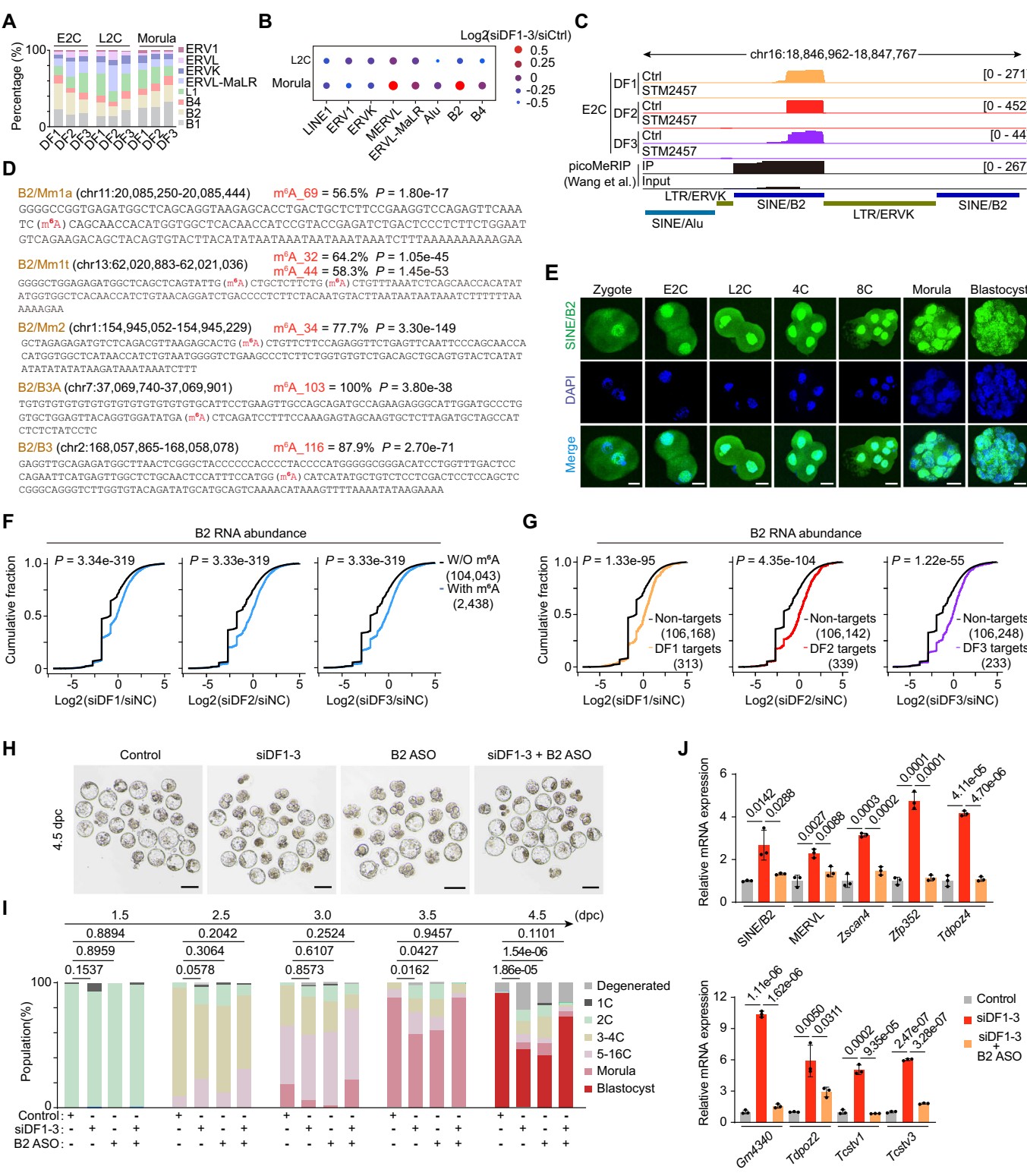

m⁶A-modified and DF-bound MERVL was also observed in the same embryos (Fig. EV4I,J). Likewise, DF proteins also promote B2 RNAs and MERVL decay in L2C embryos (Fig. EV4K,L). Indeed, the abundance of retrotransposon RNAs, including MERVL and B2, is markedly upregulated in DRB-treated embryos

with individual si*Ythdf* knockdown (Fig. EV4M), once again confirming that DF1 can promote RNA decay in embryos. Taken together, these findings suggest that DF proteins can mediate the degradation of B2 RNA, along with MERVL, in an m⁶A-dependent manner.

**Figure 4. B2 transcripts are major targets of DF1/2/3 proteins.**

(A) Histogram showing the enrichment score of DF-binding reads in major LINE, LTR, and SINE subfamilies at the E2C, L2C, and morula stages. (B) Bubble plot showing changes in TE transcript abundance (si*Ythdf1-3*/siControl) at the L2C and morula stages. (C) Genome browser snapshot showing the read coverage of the B2 locus in DF1/2/3LACE-seq datasets and the picoMeRIP-seq datasets. (D) Alignment of consensus sequences from B2 subfamilies, with annotated putative m6A sites and quantification of their methylation stoichiometries derived from published GLORI datasets in mESCs (Liu et al, 2022). P values were determined based on the binomial distribution. (E) Representative images of RNA-FISH for B2 RNA with DAPI counterstain during preimplantation development. Scale bars, 20 μm. (F) Cumulative distribution of the log2 fold changes in B2 RNA abundance in DRB-treated L2C embryos between si*Ythdf* and siControl for m6A-modified and unmodified RNAs. P values were determined with the Kolmogorov–Smirnov test. (G) Cumulative distribution of the log2 fold changes in B2 RNA abundance in DRB-treated L2C embryos between si*Ythdf* and siControl for non-targets and DF LACE-seq targets. P values were determined with the Kolmogorov–Smirnov test. (H) Representative images showing embryos treated with *Ythdf* siRNAs, B2 ASO, *Ythdf* siRNAs + B2 ASO, or control siRNA at 4.5 dpc. Representative images were selected from two independent experiments. Scale bars, 100 μm. (I) Percentages of embryonic stages observed at the indicated time points in Control (n = 42), B2 ASO (n = 35), siDF1-3 (n = 52), and siDF1-3 + B2 ASO (n = 51) groups. P values were determined by the Chi-square test. (J) RT-qPCR validation of B2 transcripts and 2C genes in Control, siDF1-3, and siDF1-3 + B2 ASO-treated embryos at 4.5 dpc. Data are mean ± SD, n = 3 biological replicates. P values were determined by a two-tailed unpaired t test. Source data are available online for this figure.

## Depletion of B2 RNA largely rescues the phenotype observed in *Ythdf1-3* knockdown embryos

Given that DF proteins mediate B2 RNA decay in embryos and that B2 RNA is markedly upregulated in si*Ythdf1-3* knockdown morulae, we investigated whether overexpressing B2 RNA could mimic the developmental defects observed with si*Ythdf1-3* knockdown. As expected, embryos microinjected with 1× B2 RNA (without m6A modification, ~170 ng/μl) exhibited notable developmental delays beginning at 1.5 dpc and a significant reduction in blastocyst formation (46.6%). In contrast, embryos injected with a lower concentration of B2 RNA showed milder developmental delays starting at 2.5 dpc, suggesting that the phenotype induced by B2 RNA is dose-dependent (Appendix Fig. S4A–C). Next, we conducted RNA-seq analysis on B2 overexpression (OE) and control embryos at the L2C and morula stages (Appendix Fig. S4D–F). B2 OE embryos showed significant downregulation of genes related to ribosome biogenesis and nuclear localization at the L2C and morula stages, respectively (Appendix Fig. S4G; Dataset EV6). Notably, genes upregulated in both B2 overexpression (OE) and *Ythdf1-3* knockdown embryos include well-established 2C markers such as *Gm4340*, *Tdpoz1/4/5*, *Zfp352*, and *Zscan4c*. In contrast, the shared down-regulated genes are mainly involved in developmental pathways, including *Creb3l2*, *Cdh3*, *Dhh*, *Fstl3*, *Heg1*, *Notch1*, *Lrp1*, *Plxnb1*, and *Rgs14* (Appendix Fig. S4H). Indeed, we observed significant upregulation of 2C marker genes such as *Gm4340*, *Zscan4*, *Zfp352*, and *Tdpoz2/4*, as well as MERVL, in B2 OE morula embryos (Appendix Fig. S4I,J), resembling the upregulation of these 2C genes in si*Ythdf1-3* knockdown embryos. Similarly, the protein levels of Oct4 and Nanog showed no apparent difference between B2 OE embryos and the control group, despite downregulated mRNA levels of *Oct4* and *Nanog* in B2 OE embryos (Appendix Fig. S4J–L). In addition, there was no observable difference in the number of blastomeres per embryo between B2 OE embryos and the control group at 3.5 dpc (Appendix Fig. S4M). These results indicate that overexpression of B2 RNA partially recapitulates the characteristics observed in *Ythdf1-3* knockdown early embryos, independent of m6A-associated effects.

Then, we investigated whether reducing endogenous B2 RNAs could alleviate the developmental abnormalities observed in si*Ythdf1-3* knockdown embryos. To achieve this, we concurrently depleted B2 RNA and DF1-3 by microinjecting B2-specific antisense ASOs and *Ythdf1-3* siRNAs into the PN2 zygotes (Fig. EV5A). As expected, depletion of B2 RNA significantly promoted the development and blastocyst formation of *Ythdf1-3*

knockdown embryos, accompanied by a marked reduction in the expression of 2C genes at the blastocyst stage (Fig. 4H–J), suggesting that reducing the abundance of B2 RNAs could effectively rescue the defects observed in si*Ythdf1-3* knockdown embryos. Markedly, depleting B2 RNAs alone in wild-type embryos also resulted in developmental delay and reduced blastocyst formation (Fig. 4H,I), hinting that the modest abundance of B2 RNA is crucial for embryonic development. In contrast, we observed that LINE1 ASO aggravated the defects of si*Ythdf1-3* knockdown embryos rather than rescuing the blastocyst rate, and MERVL ASO slightly improved the blastocyst rate of si*Ythdf1-3* knockdown embryos (Fig. EV5B–D). The expression levels of 2C genes in both LINE1 ASOs and MERVL ASOs rescuing embryos are higher or comparable to si*Ythdf1-3* knockdown embryos (Fig. EV5E), suggesting that knocking down either LINE1 or MERVL RNAs does not effectively rescue *Ythdf1-3* knockdown embryos. To further validate the rescue effect of MERVL, we increased the concentration of MERVL ASOs in si*Ythdf1-3* knockdown embryos. Surprisingly, instead of rescuing the defects, the embryos exhibited aggravated phenotypes, with 2C gene expression levels remaining comparable to those in *Ythdf1-3* knockdown embryos (Fig. EV5F–H). Thus, these results demonstrate that DF proteins promote retrotransposon RNA decay in early embryos, and the phenotype caused by *Ythdf1-3* knockdown is predominantly attributed to B2 RNA accumulation.

## B2 RNA regulates Pol II transcription in a *trans*-regulatory manner

B2 RNAs are known to play roles in stress responses, such as during heat shock, by integrating into the pre-initiation complex of RNA Polymerase II (Pol II) at gene promoters and inhibiting transcriptional elongation, thereby functioning as a *trans*-regulator (Ponicsan et al, 2015; Zovoilis et al, 2016). We investigated whether the excessive accumulation of B2 RNAs blocks Pol II elongation in si*Ythdf1-3* knockdown embryos. Initially, we observed that B2 RNAs primarily localized in the nucleolus, separate from Pol II, in L2C embryos. However, at the morula stage, B2 RNAs colocalized with Pol II in the nucleus (Fig. 5A), suggesting a potential interaction between B2 RNAs and Pol II in morula embryos. To decipher the underlying mechanism, we performed CUT&Tag (Kaya-Okur et al, 2019) for Pol II in si*Ythdf1-3* knockdown and control embryos at the L2C and morula stages (Fig. 5B; Appendix Fig. S5A). At the L2C stage, Pol II binding at promoters was globally similar in both *Ythdf1-3* knockdown and control embryos (Fig. 5C), indicating that

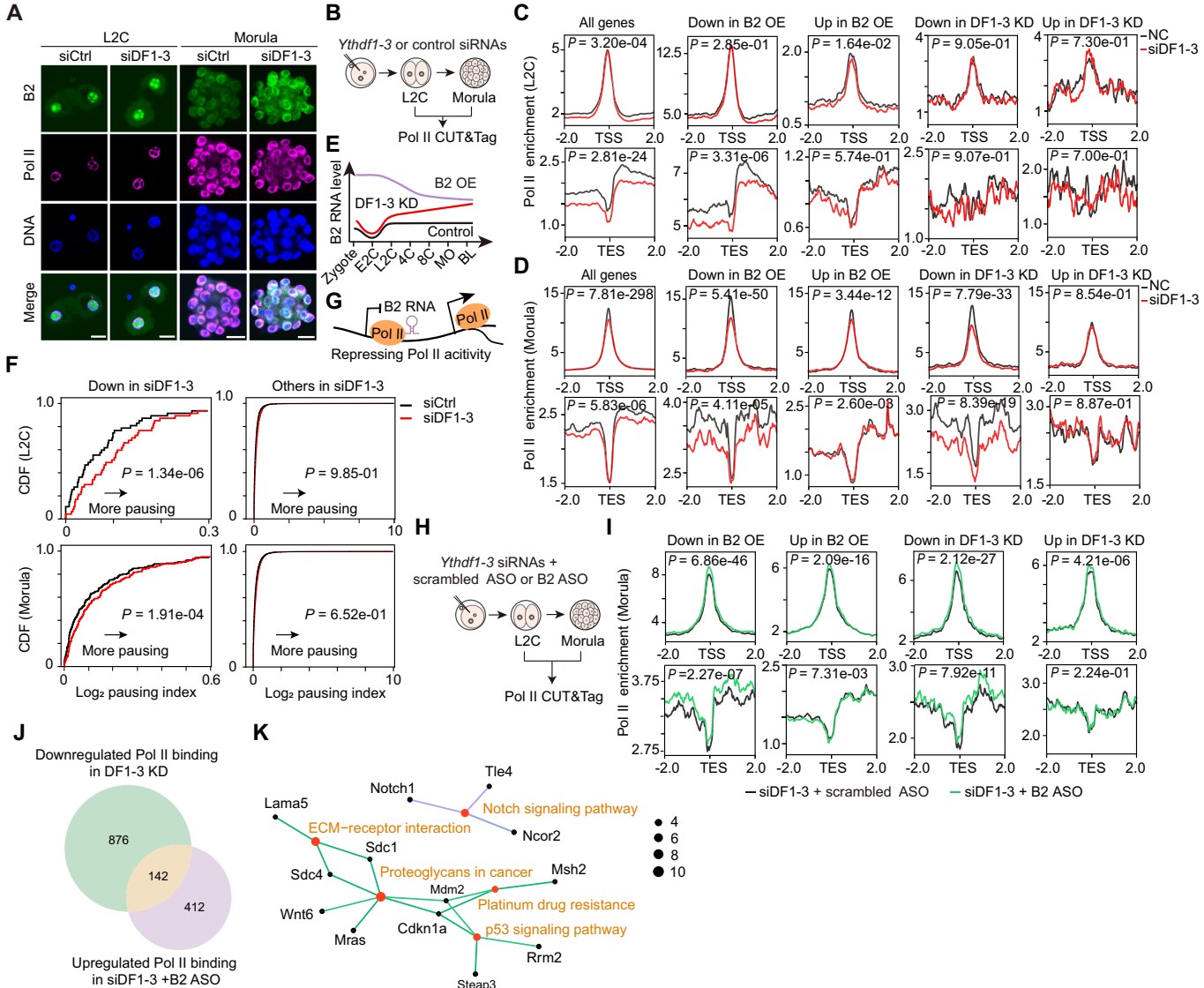

**Figure 5. B2 regulates Pol II transcription in a *trans*-active manner upon *Ythdf1-3* knockdown.**

(A) Representative images of IF/RNA-FISH for B2 RNA with Pol II counterstain in control and si*Ythdf1-3* knockdown embryos at the L2C and morula stages. Scale bars, 20 μm. (B) Schematic of the experimental procedure for performing Pol II CUT&Tag in si*Ythdf1-3* knockdown and control embryos at the L2C and morula stages. (C, D) Profiles of Pol II levels on all gene loci (left), DEG loci of B2 overexpression (middle), and DEG loci of si*Ythdf1-3* knockdown (right) from 2.0 kb up- to downstream of the transcription start site (TSS) or the transcription end site (TES) in control and si*Ythdf1-3* knockdown L2C (C) and morula (D) embryos. *P* values were determined with the Kolmogorov–Smirnov test. (E) Schematic depicting the dynamic expression levels of B2 RNAs in early embryos with si*Ythdf1-3* knockdown or B2 overexpression. (F) Cumulative distributions of the log₂ pausing index in downregulated genes (left) and other genes (right) for Pol II binding between si*Ythdf1-3* knockdown and control embryos at the L2C (upper) and morula (bottom) stages. *P* values were determined with the Kolmogorov–Smirnov test. (G) Schematic of B2 RNA repressing Pol II transcription. (H) Schematic of the experimental procedure for performing Pol II CUT&Tag in si*Ythdf1-3* knockdown and B2 rescue embryos at the L2C and morula stages. (I) Profiles of Pol II levels on DEG loci of B2 overexpression (left) and *Ythdf1-3* knockdown (right) from 2.0 kb up- to downstream of the TSS (upper) or the TES (bottom) in si*Ythdf1-3* knockdown and B2 ASO rescue embryos at the morula stage. *P* values were determined with the Kolmogorov–Smirnov test. (J) Venn diagram showing the overlapping genes identified from downregulated Pol II binding signals in si*Ythdf1-3* knockdown embryos and upregulated Pol II binding signals in B2 ASO rescue embryos. (K) KEGG analysis of the overlapping genes identified from downregulated Pol II binding signals in si*Ythdf1-3* knockdown embryos and upregulated Pol II binding signals in B2 ASO rescue embryos. Source data are available online for this figure.

initial Pol II binding is not affected by si*Ythdf1-3* knockdown. However, transcriptional elongation of Pol II on the gene body was impaired in si*Ythdf1-3* knockdown embryos compared to control embryos. Conversely, in morula embryos, the Pol II binding features were reversed (Fig. 5D), with a reduction of Pol II binding signals at the promoter region but not at the transcription termination sites (TES)

region in si*Ythdf1-3* knockdown embryos, suggesting that the transcriptional control of Pol II becomes more complex with prolonged si*Ythdf1-3* knockdown embryos. Intriguingly, the downregulated genes identified from si*Ythdf1-3* knockdown and B2 OE embryos were more likely to exhibit repressed transcriptional elongation (around the TES regions) of Pol II in si*Ythdf1-3* knockdown embryos compared to controls. In

contrast, the upregulated genes did not exhibit enhanced Pol II transcriptional elongation. Note that the number of up-and down-regulated genes (76 and 161, respectively) identified from si*Ythdf1-3* knockdown L2C embryos was so small that evaluating the difference in Pol II binding between si*Ythdf1-3* knockdown and control embryos was challenging. The repression of Pol II elongation for downregulated genes is enhanced by the accumulation of B2 RNA from L2C to morulae with si*Ythdf1-3* knockdown. Conversely, this repression is diminished by the decrease of B2 RNA from the L2C stage to morulae in B2 OE embryos (Fig. 5E). Furthermore, we evaluated the changes of the pausing index (Williams et al, 2015) (the ratio of Pol II signal density near a gene promoter to signal density within the gene body) for downregulated genes from si*Ythdf1-3* knockdown embryos in our Pol II CUT&Tag datasets. Downregulated genes showed a significantly increased pausing index in si*Ythdf1-3* knockdown embryos compared to other genes (unaffected and upregulated genes) (Fig. 5F), suggesting that B2 RNA can sequester Pol II at the gene promoter, thereby inhibiting Pol II elongation. Thus, these data suggest that the excessive accumulation of B2 RNA is more likely to repress Pol II elongation, contributing to gene downregulation at the transcriptional level in si*Ythdf1-3* knockdown embryos. However, the upregulated genes in si*Ythdf1-3* knockdown embryos result from the loss of RNA degradation rather than increased transcription, indicating that post-transcriptional regulation plays a more significant role for these genes. To confirm the role of B2 RNAs in Pol II transcription repression (Fig. 5G), we performed CUT&Tag for Pol II in si*Ythdf1-3* knockdown and B2 ASO rescue (siDF1-3 + B2 ASO) embryos at the L2C and morula stages (Fig. 5H; Appendix Fig. S5B). The downregulated genes from both si*Ythdf1-3* knockdown and B2 OE embryos showed enhanced Pol II transcriptional elongation in B2 ASO rescue embryos compared to si*Ythdf1-3* knockdown embryos, whereas the upregulated genes did not exhibit increased Pol II elongation (Fig. 5I), confirming that DF-mediated RNA decay plays a more significant role in the upregulation of these genes. Additionally, in L2C embryos, there were no apparent changes in Pol II elongation for both up-and-downregulated genes, aligning with the marginal colocalization of Pol II with B2 RNAs at the L2C stage (Appendix Fig. S5C). We next identified 142 genes whose expression was repressed by B2 RNA, based on changes in Pol II binding signals observed in *Ythdf1-3* knockdown and B2 ASO rescue embryos (Fig. 5J; Dataset EV7). Analysis of the 142 B2-repressed genes at the morula stage reveals their association with critical pathways involved in embryonic development, specifically the p53 signaling pathway and the Notch signaling pathway (Adiga et al, 2007; Huppert et al, 2000) (Fig. 5K). Additionally, beyond the repressive effect of B2 RNA on Pol II transcription, we found that m⁶A-marked genes also exhibited reduced Pol II transcription in *Ythdf1-3* knockdown embryos (Appendix Fig. S5D), concordance with the report that the transcription activation induced by m⁶A depletion in mESCs (Liu et al, 2020b). This result supports that the abundance of DF-targeted m⁶A-marked transcripts is susceptible to transcription regulation, despite their tendency to exhibit increased stability upon DF protein knockdown.

## B2 transcription modules RNA Pol II activity at genomic regions adjacent to B2 loci in early embryos

In mammals, B2 RNAs are transcribed by Pol III, and it has been suggested that Pol III-occupied SINEs act as promoters or enhancers for nearby protein-coding genes (Ferrigno et al, 2001; Zhang et al, 2019). We sought to explore the potential *cis*-

regulatory activity of SINE/B2 elements on Pol II transcription in *Ythdf1-3* knockdown embryos. To address this, we conducted CUT&Tag for Pol III in si*Ythdf1-3* knockdown and control embryos at the L2C and morula stages (Fig. 6A; Appendix Fig. S5E). Pol III binding signals were preferentially enriched in the tDNA (transfer RNA gene) locus compared to the genic regions, in contrast to Pol II binding signals, in line with the known roles of Pol III in transcribing transfer RNAs (tRNAs), and Pol II primarily transcribes mRNAs, despite the cross-regulatory effects observed between these RNA polymerases (Jiang et al, 2022) (Appendix Fig. S5F,G). At the L2C stage, *Ythdf1-3* knockdown embryos showed reduced transcription of Pol III-driven B2 RNAs, which corresponded with diminished Pol II transcription near SINE/B2 loci (Fig. 6B). In contrast, B2 RNA transcription was mildly elevated in si*Ythdf1-3* knockdown embryos, along with the Pol II transcription increase around SINE/B2 loci at the morula stage (Fig. 6b), implying that Pol II transcription around SINE/B2 loci is positively associated with B2 RNA transcription. Moreover, we observed a significant reduction in Pol II signal near SINE/B2 loci (≤50 kb), whereas the regions located farther away (>50 kb) remained unaffected at the L2C stage (Fig. 6C). Notably, in si*Ythdf1-3* knockdown morulae, Pol II signals appear largely unaffected by distance from SINE/B2 loci (Appendix Fig. S5H), consistent with a mild increase in Pol II transcription near these loci (Fig. 6B). These findings support a model in which B2 RNA transcription positively regulates adjacent Pol II activity in a *cis*-acting manner. In addition, Pol II-driven MERVL transcription activity is similar to Pol III-driven B2 RNA transcription in si*Ythdf1-3* knockdown embryos at the L2C and morula stages (Fig. 6D), suggesting that the abundance of B2 RNAs, along with MERVL, is regulated by both transcriptional and post-transcriptional levels in early embryos. In summary, B2 RNAs repress protein-coding gene transcription by *trans*-regulating Pol II elongation; meanwhile, they also repress Pol II transcription around the B2 locus through *cis*-regulation, contributing to gene downregulation in si*Ythdf1-3* knockdown embryos.

## Discussion

The DF1/2/3 proteins serve as primary cytosolic m⁶A-binding proteins, playing key roles in mediating the effects of m⁶A-marked transcripts in the cytosol (Shi et al, 2019). However, the exact mechanisms by which m⁶A and its binding proteins regulate gene expression and development are still largely unknown. In this study, we have systematically presented findings highlighting the critical role of m⁶A modification in RNA decay during mouse embryonic development, mediated by its reader proteins DF1/2/3 (Fig. 6E). Depletion of DF2 or DF3, but not DF1, compromised murine preimplantation development, with the most severe phenotype observed upon *Ythdf1-3* simultaneous knockdown, accompanied by abnormal gene expression and accumulation of retrotransposon RNAs. Notably, compensatory effects and redundant targets among DF proteins may underscore the dispensable role of DF1 in early embryos, despite its capacity, alongside DF2 and DF3, to promote RNA decay. Our results are consistent with a previous study showing that the knockdown of *Ythdf2* or *Ythdf3* delays somatic cell reprogramming, whereas *Ythdf1* knockdown does not affect this process (Liu et al, 2020a). For the DF paralogs, several post-

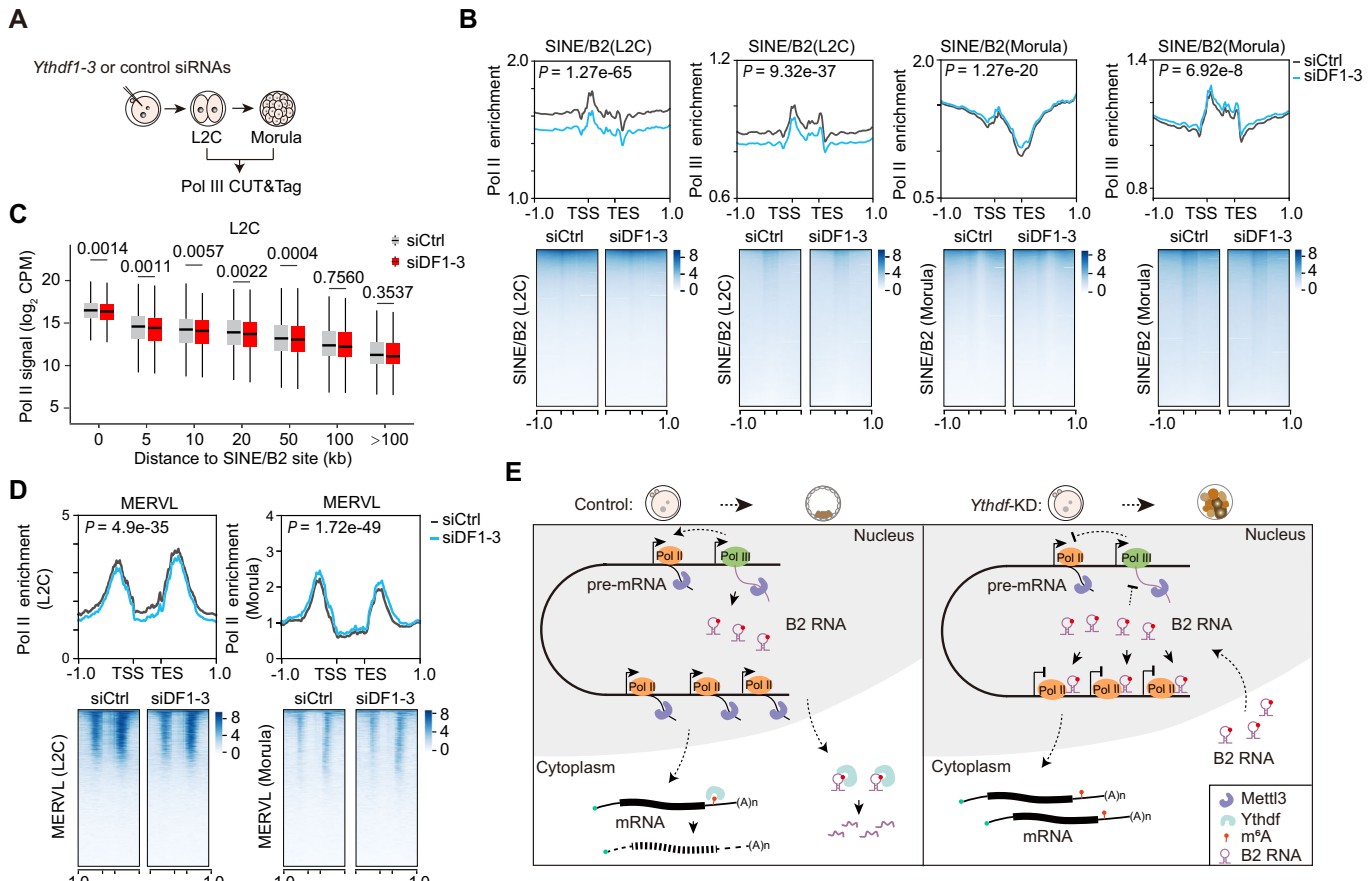

**Figure 6. B2 transcription modulates Pol II activity adjacent to B2 loci.**

(A) Schematic of the experimental procedure for performing Pol III CUT&Tag in si*Ythdf1-3* knockdown and control embryos at the L2C and morula stages. (B) Profiles and heatmaps of Pol II and Pol III binding on B2 RNA loci from 1.0 kb upstream of the TSS to 1.0 kb downstream of the TES in control and si*Ythdf1-3* knockdown embryos at the L2C and morula stages. *P* values were determined with the Kolmogorov–Smirnov test. (C) Box plots showing Pol II signal fold changes between si*Ythdf1-3* knockdown and control embryos at the L2C stage. The Pol II signals were categorized according to their genomic distance to the nearest B2 loci. B2 loci (0 kb), n = 7052; B2 loci (0–5 kb), n = 4503; B2 loci (5–10 kb), n = 3099; B2 loci (10–20 kb), n = 4064; B2 loci (20–50 kb), n = 6177; B2 loci (50–100 kb), n = 4468; B2 loci ( > 100 kb), n = 6961. Boxes represent the 25th-75th percentile (line at the median), with whiskers at 1.5× interquartile range (IQR). *P* values were determined with the Kolmogorov–Smirnov test. (D) Profiles and heatmaps of Pol II binding on MERVL loci from 1.0 kb upstream of the TSS to 1.0 kb downstream of the TES in control and si*Ythdf1-3* knockdown embryos at the L2C and morula stages. *P* values were determined by the Kolmogorov–Smirnov test. (E) A model illustrating that DF1/2/3 proteins mediate the interaction between B2 RNA and Pol II activity in early embryos. Each DF protein can promote the decay of mRNAs and B2 RNAs. Knockdown of *Ythdf1-3* leads to the retention of maternal RNAs, persistent expression of 2C genes, and accumulation of B2 RNAs, which in turn broadly repress RNA polymerase II transcription in a *trans*-acting manner.

translational modifications (PTMs), including phosphorylation, ubiquitination, SUMOylation, and glycosylation, have been reported to modulate their subcellular localization, ligand or partner interactions, and stress granule (SG) assembly (Chen et al, 2023; Sikorski et al, 2023). Recent studies further demonstrate that m⁶A-modified mRNAs can undergo liquid-liquid phase separation (LLPS) with DF proteins, resulting in the formation of endogenous LLPS compartments that facilitate compartment-specific mRNA regulation (Fu and Zhuang, 2020; Gao et al, 2019; Ries et al, 2019; Wang et al, 2019). Together, these intrinsic properties of DF proteins may ultimately influence their compensatory efficiency during early embryonic development. During early development, we observe a distinct expression pattern of DF proteins, with each protein contributing to the regulation of RNA decay and exhibiting similar m⁶A recognition capabilities. These findings support the unified model of YTHDF-mediated mRNA decay and reveal a critical YTHDF–SINE/B2–gene regulatory axis operating

during early embryogenesis. The potential translation-promoting role of DF proteins in early embryos, however, warrants further investigation.

The enhanced stability of RNAs in si*Ythdf1-3* knockdown embryos correlated with the accumulation of transcripts from 2C genes, maternal RNAs, MGA transcripts, MERVL RNAs, and B2 RNAs. Furthermore, the developmental abnormalities induced by si*Ythdf1-3* knockdown were significantly alleviated by silencing B2 RNA, underscoring the critical role of B2 RNAs as targets of DF proteins. Although B2 RNAs are abundant in germ cells and early embryos, their functional roles are less reported. Previous studies have shown that B2 RNAs can be incorporated into the Pol II pre-initiation complex at gene promoters, thereby inhibiting transcriptional elongation both in vitro and in vivo, particularly under stress conditions (Ponicsan et al, 2015; Zovoilis et al, 2016). Given the accumulation of B2 RNAs and the widespread downregulation of

genes observed in si*Ythdf1-3* knockdown embryos, we hypothesized that elevated B2 RNA levels could repress the transcription of a subset of genes, thereby contributing to their downregulation. Our analysis of Pol II and Pol III CUT&Tag datasets confirms that downregulated genes in *Ythdf1-3* knockdown embryos are more likely to exhibit suppressed Pol II transcription than upregulated genes. In addition to their *trans*-regulatory role, we observed that SINE/B2 elements repress Pol II transcription adjacent to SINE/B2 loci through *cis*-regulatory activity at the L2C stage, suggesting they have dual or multiple roles in RNA metabolism during early development.

In embryonic stem cells and early embryos, various classes of autonomous TEs are actively transcribed and marked with m⁶A, including IAP, ERVK, and LINE1. The m⁶A modification, along with its binding proteins, plays critical roles in regulating chromatin modifications, gene expression, and the repression of retrotransposons (Chelmicki et al, 2021; Liu et al, 2020b; Liu et al, 2021; Wei et al, 2022; Xu et al, 2021). These mechanisms are essential for maintaining cellular identity and supporting proper development. B2 RNA is highly expressed in germ cells and early embryos but is rarely explored. We demonstrated that DF1/2/3 proteins bind to L1, LTR, and SINE retrotransposon RNAs, which are recognized depending on m⁶A modification. However, only MERVL and B2 RNAs show significant accumulation in *Ythdf1-3* knockdown embryos, indicating that the turnover of retrotransposon RNAs is regulated at both the transcriptional and post-transcriptional levels. The excessive accumulation of MERVL and B2 RNAs may contribute to the phenotypic abnormalities observed in *Ythdf1-3* knockdown embryos. Our rescue assays demonstrated that only B2 RNAs, but not LINE1 or MERVL, significantly contribute to the defects observed in *Ythdf1-3* knockdown embryos. Previous studies have highlighted the essential *cis*-acting functions of MERVL RNA in preimplantation development (Sakashita et al, 2023), which are not dramatically affected in *Ythdf1-3* knockdown embryos. Therefore, B2 RNAs emerge as the primary candidate retrotransposon RNAs impacting developmental potency.

In summary, we propose an expanded model in which the coordinated regulatory axis of Ythdf-SINE/B2-gene expression plays a critical role in safeguarding the embryonic transcriptome from the detrimental effects of dysregulated retrotransposon RNAs, 2C transcripts, maternal RNAs, and development-associated genes during early development. This protective role promotes embryogenesis, expanding our understanding of the regulatory functions of m⁶A in development.

# Methods

### Reagents and tools table

| Reagent/resource | Reference or source | Identifier or catalog number |
|---|---|---|
| **Experimental models** | | |
| C57BL/6J (*M. musculus*) | Guangdong Zhiyuan Biomedical Technology | ZY001 |
| DBA2 (*M. musculus*) | Beijing HFK Bioscience | 2018041222 |
| **Antibodies** | | |
| Anti-β-Tubulin | Tianjin Sungene Biotech | KM9007 |

| Reagent/resource | Reference or source | Identifier or catalog number |
|---|---|---|
| Anti-YTHDF1 | Abcam | ab220162 |
| Anti-YTHDF1 | Proteintech | 17479-1-AP |
| Anti-YTHDF2 | Abcam | ab246514 |
| Anti-YTHDF2 | Cell Signaling Technology | 80014S |
| Anti-YTHDF3 | Abcam | ab220161 |
| Anti-OCT4 | Santa Cruz Biotechnology | sc-5279 |
| Anti-NANOG | Cell Signaling Technology | 8822 |
| Anti-CDX2 | BioGenex | MU392A-UC |
| Anti-flag | Abclonal | AE005 |
| Anti-RNA pol II | Active motif | 39497 |
| Anti-POLR3A | Abcam | ab96328 |
| Alexa Fluor 594-conjugated goat anti-rabbit secondary antibody | Thermo Scientific | A11012 |
| Alexa Fluor 488-conjugated goat anti-mouse secondary antibody | Thermo Scientific | A11029 |
| AF647-labeled Goat Anti-Mouse IgG (H + L) | Beyotime | A0473 |
| Anti-m6A | Synaptic Systems | 202003 |
| **Oligonucleotides and other sequence-based reagents** | | |
| siRNA sequences | This study | Table EV1 |
| Cas9 guide RNA sequences | This study | Table EV1 |
| CasRx guide RNA sequences | This study | Table EV1 |
| PCR primers | This study | Table EV1 |
| B2 RNA-FISH probe sequences | This study | Table EV1 |
| B2 ASO sequence | This study | Table EV1 |
| LINE1 ASO sequence | Percharde et al, 2018 | Table EV1 |
| MERVL ASO sequence | Sakashita et al, 2023 | Table EV1 |
| **Chemicals, enzymes and other reagents** | | |
| KSOM medium | Millipore | MR-107-D |
| T7 High Yield RNA Transcription Kit | Ambion | AM1344 |
| Poly(A) polymerase tailing kit | Epicentre | PAP5104H |
| DRB | Sigma | D1916 |
| STM2457 | Selleck | S9870 |
| PVDF membrane | Millipore | 1620177 |
| Wes Separation Module microplate | ProteinSimple | SM-W002 |
| PicoPure RNA Isolation Kit | QIAGEN | 74034 |
| Maxima H Minus First Strand cDNA Synthesis Kit | Thermo Scientific | K1682 |
| Power SYBR Green Master Mix | Vazyme | Q111 |
| KAPA HiFi HotStart Ready Mix | KAPA Biosystems | KK2601 |
| GenElute™ Single Cell RNA Purification Kit | Sigma | RNB300 |
| Hybond-N+ membrane | GE Healthcare | RPN203B |

| Reagent/resource | Reference or source | Identifier or catalog number |
|---|---|---|
| 20× SSC | Thermo Scientific | AM9770 |
| Dextran Sulfate 50% Solution | Sigma | S4030 |
| Formamide | Sigma | 47671-250ML-F |
| Ribonucleoside vanadyl complexes (RVC) | Sigma | R3380-5ML |
| Tyrode's solution | Sigma | T1788 |
| Pierce chip-grade protein A/G magnetic beads | Thermo Scientific | 26162 |
| RNase inhibitor | Thermo Scientific | EO0381 |
| RQ1 DNase | Promega | M6101 |
| Micrococcal nuclease | New England BioLabs | M0247S |
| FastAP alkaline phosphatase | Thermo Scientific | EF0651 |
| T4 RNA ligase 2 | New England Biolabs | M0242 |
| RNase H | Thermo Scientific | EN0202 |
| Dynabeads™ MyOne™ Streptavidin C1 beads | Thermo Scientific | 65002 |
| Ampure XP beads | Beckman Coulter | A63881 |
| T7 RNA Polymerase | New England BioLabs | M0251 |
| TURBO DNase | Thermo Scientific | AM2238 |
| Agencourt RNA Clean beads | Beckman Coulter | A63987 |
| Gel Extraction Kit | Qiagen | 28604 |
| Hyperactive Universal CUT&Tag Assay Kit | Vazyme Biotech | TD903 |
| Superscript II Reverse Transcriptase | Thermo Scientific | 18064071 |
| RiboLock RNase inhibitor | Thermo Scientific | EO0382 |
| 10 × PBS buffer | Thermo Scientific | AM9624 |
| dNTP Solution Mix | New England BioLabs | N0447 |
| Ribonucleotide Solution Mix | New England BioLabs | N0466 |
| Qubit RNA HS Assay Kit | Thermo Scientific | Q32852 |
| Qubit dsDNA HS Assay Kit | Thermo Scientific | Q32854 |
| Glycogen | Thermo Scientific | AM9510 |
| IGEPEL (NP-40) | Sigma | 18896 |
| Triton X-100 | Sigma | T8787 |
| T4 RNA Ligase 1 | New England BioLabs | M0437M |
| Adenosine 5'-Triphosphate (ATP) | New England BioLabs | P0756S |
| Tween-20 | Sigma | P9416 |
| Roche Complete Protease Inhibitor EDTA-Free tablets | Roche | 4693132001 |
| **Software** | | |
| GraphPad Prism v9.5.0 | GraphPad Software | |
| Fiji: ImageJ | https://imagej.net/software/fiji/ | |
| HISAT2 v2.2.0, | Kim et al, 2019 | |
| featureCount v2.0.1 | Liao et al, 2014 | |
| DESeq2 | Love et al, 2014 | |

| Reagent/resource | Reference or source | Identifier or catalog number |
|---|---|---|
| TEtranscripts | Jin et al, 2015 | |
| Cutadapt v1.8.1 | Martin, 2011 | |
| fastp v0.21.0 | Chen et al, 2018 | |
| Bowtie2 v2.5.1 | Langmead and Salzberg, 2012 | |
| STAR58 v2.5.2b | Dobin et al, 2013 | |
| PureCLIP v 1.3.1 | Krakau et al, 2017 | |
| MACS3 v3.0.0b3 | Zhang et al, 2008 | |
| deepTools v3.5.4. | Ramírez et al, 2016 | |
| Homer | http://homer.ucsd.edu/homer/ | |
| WGCNA | Langfelder and Horvath, 2008 | |
| ChIPSeeker | Yu et al, 2015 | |
| **Other** | | |
| Roche LightCycler 480 instrument | Roche | |
| Nikon TI2-U microscope | Nikon | |
| Illumina NovaS4 sequencer | Illumina | |
| Abby™ Simple Western™ system | ProteinSimple | |

## Animals

Eight-week-old female C57BL/6 and male DBA/2 mice were maintained under SPF conditions at the Animal Center of Guangdong Second Provincial General Hospital. All animal maintenance and experimental procedures were conducted following the guidelines of the Institutional Animal Care and Use Committee of Guangdong Second Provincial General Hospital, Guangzhou, China.

## Embryo collection and in vitro culture

For embryo collection, C57BL/6 female mice were injected with 7.5 IU of pregnant mare serum gonadotropin (PMSG), followed by an injection of human chorionic gonadotropin (hCG) 48 h later, with a dosage of 5 IU. Then, female mice were mated with DBA/2 males after hCG administration. Zygotes were collected from the swollen upper part of the oviduct 18 h after hCG administration. Cumulus cells were removed from zygotes by briefly incubating them in the M2 medium with hyaluronidase. The zygotes were cultured in KSOM (Millipore, MR-107-D) droplets at 37 °C under 5% $CO_2$.

## In vitro transcription (IVT)

The Cas9 sgRNAs were designed using the online gene editing tool (https://design.synthego.com/), and CasRx sgRNAs were designed using the online tool (https://cas13design.nygenome.org/). All sgRNA sequences were listed in Table EV1. A T7 promoter was added to the sgDNA, B2_Mm1a, Cas9, and CasRx DNA templates by PCR amplification. The T7-DNA PCR products were purified and used as the DNA templates for IVT at 37 °C for 4 h using the T7 High Yield

RNA Transcription Kit (Ambion, AM1344). The Cas9 and CasRx mRNAs were then polyadenylated using a Poly(A) polymerase tailing kit (Epicentre, PAP5104H). The RNAs were purified using RNA Clean Beads according to the manufacturer's protocols.

## Embryo microinjection

Zygote microinjection was performed on the heating stage of a phase-contrast inverted microscope (Nikon). Approximately 10 pl of siRNA solution (30 μM), B2 RNA (170-1.7 ng/μl), L1 ASO (0.2–20 μM), MERVL ASO (0.2–20 μM), and B2 ASO (0.2–20 μM), were microinjected into the cytoplasm of zygotes. For Cas9-mediated KO, Cas9 mRNA (150 ng/μl), mixed with different sgRNAs (300 ng/μl), was microinjected into the cytoplasm of zygotes. For CasRx-mediated KD, CasRx mRNA (150 ng/μl), mixed with different sgRNAs (300 ng/μl), was microinjected into the cytoplasm of zygotes. The zygotes were incubated in KSOM at 37 °C with 5% $CO_2$.

## DRB and STM2457 treatment of mouse embryos

One-cell embryos were obtained at 18 h post-hCG and cultured in a KSOM medium containing 60 μM DRB (Sigma, D1916). These embryos were collected after 25 h (L2C).

For STM2457 treatment, we used 10 μM STM2457 (Selleck, S9870) in the KSOM medium and 0.1% DMSO in the KSOM medium as a control.

## Western blotting

Embryos were collected in 10 μl lysis buffer and stored at −80 °C until use. Samples were then mixed with protein sample buffer (Beyotime, P0012B) and heated at 95 °C for 10 min. Protein samples were separated by SDS-PAGE using electrophoresis systems and transferred to a polyvinylidene fluoride (PVDF) membrane (Millipore, 1620177). Then, the membrane was blocked with a blocking buffer (Beyotime, P0252) for 15 min and incubated with primary antibodies overnight at 4 °C. After rinsing three times in PBST, the membranes were incubated for 1 h with secondary antibodies. After rinsing three times in PBST, the membrane was imaged with Tanon ECL Ultra (Tanon), and the images were scanned using the VersaDoc Imaging System (Bio-Rad).

## Simple western analysis

Embryos were collected in a 4.5 μl loading buffer and denatured at 95 °C for 5 min. After cooling on ice for 5 min, 3 μl lysate was loaded to the Wes Separation Module microplate (SM-W002) for analysis according to the manufacturer's instructions (ProteinSimple).

## Quantitative RT-PCR

Total RNA was extracted from embryos using the PicoPure RNA Isolation Kit with on-column genomic DNA digestion (QIAGEN, 74034) and reverse transcribed using the RT Reagent Kit (Thermo Scientific, K1682) with oligo dT and random hexamer primers. For Quantitative RT-PCR, the resulting cDNA was quantified by real-time PCR on a LightCycler 480 instrument (Roche) using Power

SYBR Green Master Mix (Vazyme, Q111). Relative gene expression was analyzed based on the $2^{-\Delta\Delta Ct}$ method, normalized with GAPDH. For RT-PCR, the resulting cDNA was amplified by KAPA HiFi HotStart Ready Mix (KAPA Biosystems, KK2601). The PCR products were separated and imaged on a 2.5% agarose gel. All RT-qPCR primers used in this study are listed in Table EV1.

## Dot blot

$m^6A$ Dot blot assay was performed following the previous protocol (Shen et al, 2017). Total RNA was extracted from embryos using the GenElute™ Single Cell RNA Purification Kit (Sigma, RNB300). RNA was denatured at 95 °C for 3 min and transferred to ice immediately. After dropping 3 μl of RNA directly onto the Hybond-N+ membrane (GE Healthcare, RPN203B), the spotted RNA was cross-linked to the membrane in a Crosslinker twice with UV light: 200 mJoule/$cm^2$ at 254 nM. The membrane was washed in Wash buffer (1x PBS, 0.02% Tween-20) for 5 min at room temperature (RT) and incubated in Blocking buffer (1x PBS, 0.02% Tween-20, 5% non-fat milk) for 1 h at RT. The membrane was then incubated with anti-$m^6A$ antibody (1:250 dilution) in antibody dilution buffer (1× PBS 0.02% Tween-20 5% non-fat milk) overnight at 4 °C. After washing the membrane three times for 5 min each in Wash buffer, the membrane was incubated with goat anti-rabbit IgG-HRP (1:10,000 dilution; 20 ng/ml) in Antibody dilution buffer for 1 h at RT. After washing the membrane four times for 10 min each in Wash buffer, the membrane was incubated with ECL Western Blotting Substrate for 5 min in the dark at RT. Then, the membrane was imaged with Tanon ECL Ultra (Tanon), and the images were scanned using the VersaDoc Imaging System (Bio-Rad).

## Immunostaining

Embryos were fixed in 4% paraformaldehyde at room temperature for 30 min. Then, the embryos were permeabilized with 0.5% Triton X-100 at room temperature (RT) for 30 min. The samples were then blocked with 10% FBS at room temperature for 1 h. After blocking, the embryos were incubated overnight at 4 °C with the primary antibody. The following day, the embryos were washed three times with 0.1% PVP in PBS for 5 min each and then incubated with Alexa Fluor 488-conjugated anti-mouse and/or 594-conjugated anti-rabbit immunoglobulin G secondary antibodies (Thermo Scientific, A11029, A11012; 1:500) for 1 h at RT. After washing the embryos three times with 0.1% PVP in PBS for 10 min each, they were counterstained with DAPI for 15 min at RT and then loaded onto glass slides. Images were obtained with a confocal laser scanning microscope (Leica DMi8) and processed using ImageJ (NIH).

## RNA-FISH and immunofluorescence co-staining

Embryos were fixed in 4% paraformaldehyde for 10 min at RT, infiltrated in PBST/RVC buffer (1× PBS, 2 mM ribonucleoside vanadyl complexes (RVC), 0.5% Triton X-100) for 10 min at 4 °C. After that, the samples were permeabilized twice with 70% ethanol, 90% ethanol, and 100% ethanol for 5 min each at 4 °C. Then, they were hybridized overnight at 37 °C in a hybridization buffer (55% formamide, 11% dextran, 11% 20× SSC, 0.11% BSA, and 22 mM

RVC) using probes specifically designed to target the entire length of B2_mm1a RNA (Table EV1). After washing twice with FA/2×SSC buffer (50% Formamide, 2×SSC, DEPC-treated water) and 2×SSC for 10 min each, the nuclei of the embryos were counterstained with DAPI for 10 min at RT. All images were collected using a confocal laser scanning microscope (Leica DMi8) and processed with ImageJ (NIH).

B2 FISH/immunofluorescence co-staining was performed following the previous protocol (Namekawa and Lee, 2011). Embryos were fixed in 1% paraformaldehyde on ice for 10 min. After that, the samples were permeabilized in 70% (vol/vol) ethanol at 4 °C until use. After incubating with a blocking buffer (1% BSA, 0.1% TWEEN 20, 0.4 U/µl SUPERase•In™ RNase Inhibitor in PBS), the primary antibody at a 1:100 dilution in blocking buffer was incubated for 1 h at room temperature. Samples were washed with blocking buffer and stained with Alexa 594/647-conjugated secondary antibody (Thermo Scientific, A11012, 1:500; Beyotime, A0473, 1:500) for 1 h. Then samples were fixed again with 4% PFA for 10 min. After dehydrating slides by serial treatment with 70%, 80%, and 100% (vol/vol) ethanol, samples were hybridized with FISH probes at 42 °C overnight. After washing twice with FA/2×SSC buffer (50% Formamide, 2×SSC, DEPC-treated water) and 2×SSC, the nuclei of the embryos were counterstained with DAPI for 10 min at RT. For all the experiments mentioned in this section, the images were collected using a confocal laser scanning microscope (Leica DMi8). The intensity of FISH signals was quantified using Fiji (ImageJ).

## RNA-seq

RNA-seq libraries were generated following the Smart-seq2 protocol (Picelli et al, 2014), as described previously. The zona pellucida was gently removed with Tyrode's solution (Sigma, T1788). Oocytes and embryos were washed three times with 0.1% BSA in PBS and then lysed in 4 µl lysis buffer containing RNase inhibitor.

## LACE-seq

LACE-seq was performed based on our previously described protocol (Su et al, 2021) with minor modifications. Briefly, 250–300 ZP-free two-cell or morula embryos were cross-linked with 0.1% formaldehyde at RT for 10 min and then quenched with 150 mM glycine at RT for 10 min. The cross-linked samples were collected in a 1.5 ml low-binding tube.

For each sample, 10 µl protein A/G magnetic beads (Thermo Scientific, 26162) were blocked with block buffer (1× PBS, 0.2 mg/ml glycogen, 0.2 mg/ml BSA) at RT for 1 h, then washed once with 0.1 M Na-phosphate buffer (93.2 mM $Na_2HPO_4$, 6.8 mM $NaH_2PO_4$, 0.05% Tween 20, pH 8.0). The blocked beads were resuspended in 50 µl of 0.1 M Na-phosphate buffer containing 2 µg YTHDF antibody at RT for 1 h. The antibody-coupled beads were washed twice with wash buffer (1× PBS, 0.1% SDS, 0.5% NP-40, 0.5% sodium deoxycholate) and resuspended in 10 µl of wash buffer per sample.

The cross-linked samples were lysed on ice with 50 µl of wash buffer for 10 min. Then, 1 µl of RNase inhibitor (Thermo Scientific, EO0381) and 4 µl of RQ1 DNase (Promega, M6101) were added, and the mixture was then incubated at 37 °C for 3 min. After snap-chilling the tube on ice for 3 min, 10 µl of beads coupled with antibodies were added to the lysate and rotated for 1 h at 4 °C. The immunoprecipitated RNAs were fragmented with $1 \times 10^{-8}$ U micrococcal nuclease (MNase, New England BioLabs, M0247S) for 3 min at 37 °C. The fragmented RNA 3' ends were dephosphorylated on beads using FastAP alkaline phosphatase (Thermo Scientific, EF0651) at 37 °C for 10 min. The fragmented RNA 3′ linker was ligated with T4 RNA ligase 2 (New England Biolabs, M0242) at RT for 2.5 h.

After converting the RNA fragment to cDNA using a biotin-modified primer, the cDNA was released from Protein A/G beads by treating it with RNase H (Thermo Scientific, EN0202) and then captured using streptavidin C1 beads (Thermo Scientific, 65002). After the cDNA 3' linker was ligated with T4 RNA ligase 1 (New England BioLabs, M0437) overnight at RT, the cDNA products were subjected to pre-PCR using KAPA HiFi HotStart Ready Mix (KAPA Biosystems, KK2601). The PCR products were purified using Ampure XP beads (Beckman Coulter, A63881) and subjected to in vitro transcription with T7 RNA Polymerase (New England BioLabs, M0251) at 37 °C for 24 h. DNA template was removed by treating it with TURBO DNase (Thermo Scientific, AM2238) at 37 °C for 30 min and the RNA was purified using Agencourt RNA Clean beads (Beckman Coulter, A63987). After performing reverse transcription and indexed PCR, the PCR products were subjected to size selection using a 2% agarose gel. Regions ranging from 250 to 500 bp were then purified using a Gel Extraction Kit (Qiagen, 28604).

## CUT&Tag

CUT&Tag was performed as previously described (Kaya-Okur et al, 2019) using the Hyperactive Universal CUT&Tag Assay Kit for Illumina (Vazyme Biotech, TD903). In brief, 50 embryos were incubated with 10 µl pre-washed ConA beads in a 1.5-ml low-binding tube. 50 µl of pre-cooled antibody buffer containing 0.8 µg primary antibody was added to the tube and incubated overnight at 4 °C. After removing the supernatant, 50 µl dig-wash buffer with 0.5 µg secondary antibody was added to resuspend the ConA beads and incubated at RT for 1 h. After washing three times with 200 µl dig-wash buffer, 1 µl pA/G-Tnp was added with 100 µl dig-300 buffer. Samples were incubated at room temperature for 1 h and then washed thrice with 200 µl dig-300 buffer. In total, 40 µl dig-300 buffer was added with 10 µl 5× TTBL, and the samples were incubated at 37 °C for 1 h. The reaction was stopped by adding 5 µl Proteinase K, 100 µl Buffer L/B, and 20 µl DNA Extract Beads. PCR was performed to amplify the libraries after extracting the DNA with DNA extraction beads. The libraries were extracted using VAHTS DNA Clean Beads and sequenced on the Illumina NovaSeq platform, following the manufacturer's instructions.

## Data analyses

### RNA-seq data analysis

All RNA-seq data were aligned to the mm9 genome by HISAT2 (Kim et al, 2019) (version 2.1.0). The read count was calculated by featureCount (Liao et al, 2014) (version 2.0.1) with the Gencode mm9 annotation. The count data was normalized using the estimateSizeFactors function and then subjected to the DESeq2(-Love et al, 2014) package for differential expression analysis. For

TE analysis, multi-aligned reads were retained and subjected to TEtranscripts (Jin et al, 2015) for quantifying TE families and conducting differential expression analysis. For published RNA-seq data, the expression values were calculated using a similar method.

### LACE-seq data analysis

The adapter sequences at both ends of the raw reads were removed using the Cutadapt (Martin, 2011) program (v1.15) with the following parameters: -m 18 -j 8 --max-n 4 --trim-n --times 2 -e 0.1 -O 3. The sequences for parameters -a, -A, and -G are listed in Table EV3. Paired-end reads were merged into single reads using fastp (Chen et al, 2018) (version 0.21.0) if there was an overlap of more than 30 nucleotides. After extracting the UMI sequence, the clean reads were initially aligned to the mouse pre-rRNA using Bowtie2 (Langmead and Salzberg, 2012) software (version 2.5.1), and the remaining unmapped reads were subsequently aligned to the mouse (mm9) reference genome using STAR (Dobin et al, 2013) (version 2.5.2b) with the following parameters: --outSJfilter-Reads Unique --alignEndsType Extend5pOfRead1 --outFilterMismatchNoverLmax 0.04 --outFilterMismatchNmax 999 --outFilterMultimapNmax 1. UMI sequences were used to remove PCR duplicates, and the retained reads were utilized for peak identification using PureCLIP (Krakau et al, 2017) (version 1.3.1). The 5' ends of LACE-seq reads within a genomic distance shorter than 100 bp were merged for motif analysis. Merged ends with more than 10 unique reads were further selected as clusters, and the summit was defined as the position with the highest read end coverage. Genomic regions 500 bp around the summit were selected as the background region. Base enrichment around the summit was defined as the ratio over the corresponding background region. The well-known $m^6A$ motif, DRACH, was used to filter potential false-positive binding events for DF proteins. For motif analysis, LACE-seq peaks were first extended by 30 nt upstream, and over-represented motifs in the extended sequences were identified using the findMotifsGenome.pl function in Homer (http://homer.ucsd.edu/homer/). The Pearson correlation coefficient between LACE-seq replicates was calculated as described previously.

### picoMeRIP data analysis

We reanalyzed the picoMeRIP data from mouse 2C and 8C embryos (Wang et al, 2023), following the instructions of the original paper (https://github.com/Augroup/ MeRipBox). Briefly, sequencing adapters and low-quality bases were trimmed using Cutadapt (v1.8.1), with the parameters: -q 20,20 -m 20 -max-n 0.01 -trim-n. Trimmed reads were aligned to the mouse reference genome (mm9) using HISAT2 (v2.1.0) with the following parameters: -5 8 -no-mixed -no-discordant. Only the uniquely aligned reads were retained. PCR duplicates were removed using SAMtools fixmate & markdup, and the reads mapped to rRNAs were also removed. $m^6A$ peaks were called using MACS3 (Zhang et al, 2008) with the following parameters: -keep-dup all -B -nomodel -call-summits. Only the peaks with a $q$ value < 0.05 were used for the subsequent analyses.

### CUT&Tag data analysis

CUT&Tag raw reads were filtered using fastp (version 0.21.0) with the following options: -g -x -q 20 -l 36 -y -c -u 40 -n 4 --detect_adapter_for_pe -w 8. The filtered reads were then aligned to the indexed mouse genome (mm9) using Bowtie2 (version 2.5.1) with the following options: --local --very-sensitive-local --no-mixed --no-discordant --phred33 -I 10 -X 700 -p 10. Uniquely mapped reads were used for peak calling using MACS3 (version 3.0.0b3) with the following parameters: -nolambda -nomodel. A cut-off of q value ≤ 0.05 was applied. The average tag density plots were generated using the plotHeatmap program implemented in deepTools (Ramírez et al, 2016) (version 3.5.4). To visualize read enrichment over representative genomic loci, BigWig files were created from sorted BAM files using bamCoverage from deepTools. Genomic distributions were displayed using ChIPSeeker (Yu et al, 2015). Pausing index analysis was conducted following the instructions provided at https://github.com/MiMiroot/PIC. For the analysis of differential binding, we first generated a set of regions called peaks using MACS3. These peaks were identified in at least one of the samples. Then, we utilized the mergeBed function from BEDTools to combine the peak regions from all biological replicates. Using the merged peak file and CUT&Tag data, read counts for each region were quantified using featureCounts. Subsequently, the data were subjected to DESeq2 for differential analysis.

### Gene expression pattern analysis

We reanalyzed the RNA-seq data (Deng et al, 2014) from MII oocytes and early embryos, following the instructions of the previous report (Park et al, 2013). We defined 13 types of stage-related gene sets. Specifically, gene expression was first averaged across developmental stages, and then hierarchical clustering was performed based on the Pearson correlation distance. The cutreeDynamic and mergeCloseModules functions from WGCNA (Langfelder and Horvath, 2008) were used to generate gene modules. Finally, the average expression pattern across the developmental stages for each gene module was used as a feature to classify them into the 13 types of stage-related gene sets.

## Statistics and reproducibility

All experiments were independently repeated at least twice, and no inconsistent results were observed. Statistical analyses were carried out using the GraphPad software or R Studio. The box borders in the boxplots represent upper and lower quartiles (25th and 75th percentiles), and the center line represents the median. The violin plots depict the distribution of the data, with the center line representing the median and the upper and lower quartiles corresponding to the 75th and 25th percentiles, respectively. The statistical tests and $P$ values are indicated in the figure legends. $P$ values < 0.05 were considered significant. All data are reproducible, and the details of replicates are stated in the figure legends.

## Data availability

All the sequencing data generated in this paper have been deposited in the Genome Sequence Archive under project PRJCA021648, with accession CRA013737 (https://ngdc.cncb.ac.cn/gsub/submit/gsa/subCRA021795, and the referee access codes: https://ngdc.cncb.ac.cn/gsa/s/dXRgF890). Publicly available datasets downloaded and used in this work were from NCBI GEO accession number GSE192440.

The source data of this paper are collected in the following database record: biostudies:S-SCDT-10_1038-S44318-026-00728-w.

## Peer review information

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

## Acknowledgements

This work was supported by the National Key R&D Program of China (2022YFC2702200 to XHO, RBS, and SML; 2021YFA1100302 to QYS; 2021YFC2700100 to XHO), the National Natural Science Foundation of China (32230028 to QYS; 32200664 to RBS; 32300952 to LHF; 32471171 to TGM, and 81971357 and 82271728 to XHO), and the Science and Technology Program of Guangzhou, China (2023A03J0258 to QYS).

## Author contributions

**Ruibao Su**: Conceptualization; Data curation; Supervision; Funding acquisition; Validation; Investigation; Visualization; Methodology; Writing—original draft; Project administration; Writing—review and editing. **Di Gao**: Validation; Investigation; Methodology. **Zongchang Du**: Data curation; Software; Formal analysis; Visualization; Methodology. **Tie-Gang Meng**: Funding acquisition; Validation; Investigation; Methodology. **Chao Li**: Validation; Methodology. **Lei-Ning Chen**: Writing—review and editing. **Xiaoting Lin**: Validation; Methodology. **Changchang Cao**: Writing—review and editing. **Li-Hua Fan**: Funding acquisition; Validation. **Yanbin Dong**: Writing—review and editing. **Sheng Li**: Validation. **Shi-Ming Luo**: Funding acquisition; Writing—review and editing. **Shuai Jiang**: Writing—review and editing. **Zhong Guo**: Writing—review and editing. **Yulu Tian**: Writing—review and editing. **Qing-Yuan Sun**: Resources; Supervision; Funding acquisition; Project administration; Writing—review and editing. **Xiang-Hong Ou**: Resources; Supervision; Funding acquisition; Project administration; Writing—review and editing.

Source data underlying figure panels in this paper may have individual authorship assigned. Where available, figure panel/source data authorship is listed in the following database record: biostudies:S-SCDT-10_1038-S44318-026-00728-w.

## Disclosure and competing interests statement

The authors declare no competing interests.

# Expanded View Figures

**Figure EV1.  Associated with Fig. 1.**

(A) RT-qPCR validation of *Ythdf* knockdown in L2C embryos. #1#2#3, three different *Ythdf* siRNAs. Data are mean ± SD, $n = 3$ biological replicates. (B, C) Western blotting showing the knockdown efficiency of *Ythdf* in L2C (B) and morula (C) embryos. (D) Relative expression of *Ythdf1/2/3* measured by RT-qPCR following single, double, or triple DF knockdown in E2C and L2C embryos. Data are mean ± SD, $n = 3$ biological replicates. (E) Simple Western immunoblotting assays showing DF1/2/3 protein levels when single DF, double DF, or triple DF knockdown in the E2C and L2C embryos. (F–H) Representative images of immunofluorescence staining of DF1 (F), DF2 (G), and DF3 (H) with DAPI counterstain in Cas9 control and DF1-3 KO morula at 3.5 dpc. Scale bars, 20 μm. Violin plots showing the DF1 (Control, $n = 31$; DF1-3 KO, $n = 27$), DF2 (Control, $n = 30$; DF1-3 KO, $n = 23$), and DF3 (Control, $n = 25$; DF1-3 KO, $n = 29$) intensity in Cas9 control and DF1-3 KO morula at 3.5 dpc. The upper and lower dotted lines in the violin plots represent upper and lower quartiles (25th and 75th percentiles), and the center line represents the median. (I) Representative images showing embryos treated with *Ythdf* guide RNAs (CasRx) or control guide RNA at 4.5 dpc. Representative images were selected from two independent experiments. Scale bars, 100 μm. (J) Percentages of embryonic stages observed at the 4.5 dpc in CasRx Control (CasRx Ctrl, $n = 28$), DF1 KD ($n = 28$), DF2 KD ($n = 27$), DF3 KD ($n = 54$), and DF1-3 KD ($n = 28$) groups. (K) Relative expression of *Ythdf1/2/3* measured by RT-qPCR following single, double, or triple DF knockdown in L2C embryos. Data are mean ± SD, $n = 3$ biological replicates. (L–N) Representative images of immunofluorescence staining of DF1 (L), DF2 (M), and DF3 (N) with DAPI counterstain in CasRx control and DF1-3 KD morula at 3.5 dpc. Scale bars, 20 μm. Violin plots showing the DF1 (Control, $n = 33$; DF1-3 KD, $n = 21$), DF2 (Control, $n = 30$; DF1-3 KD, $n = 17$), and DF3 (Control, $n = 18$; DF1-3 KD, $n = 21$) intensity in CasRx control and DF1-3 KD morula at 3.5 dpc. The upper and lower dotted lines in the violin plots represent upper and lower quartiles (25th and 75th percentiles), and the center line represents the median. (O) Representative images showing embryos treated with *Ythdf* siRNAs, *Ythdf* mRNA, or control siRNA at 4.5 dpc. Representative images were selected from 2 to 4 independent experiments. Scale bars, 100 μm. (P) Percentages of embryonic stages observed at the indicated time points in siCtrl ($n = 128$), siDF1/2 ($n = 55$), siDF1/2 + DF3 ($n = 30$), siDF1/3 ($n = 57$), siDF1/3 + DF2 ($n = 36$), siDF2/3 ($n = 96$), and siDF2/3 + DF1($n = 81$) conditions. P values in (J, P) were determined by the Chi-square test. (Q) RT-qPCR validation of *Ythdf* mRNA in blastocysts with double *Ythdf* knockdown and another *Ythdf* overexpression. Data in (A, E) are mean ± SD, $n = 3$ biological replicates. The P value in (A, D, F, G, H, K, L, M, N, Q) was determined by a two-tailed unpaired *t* test. Source data are available online for this figure.

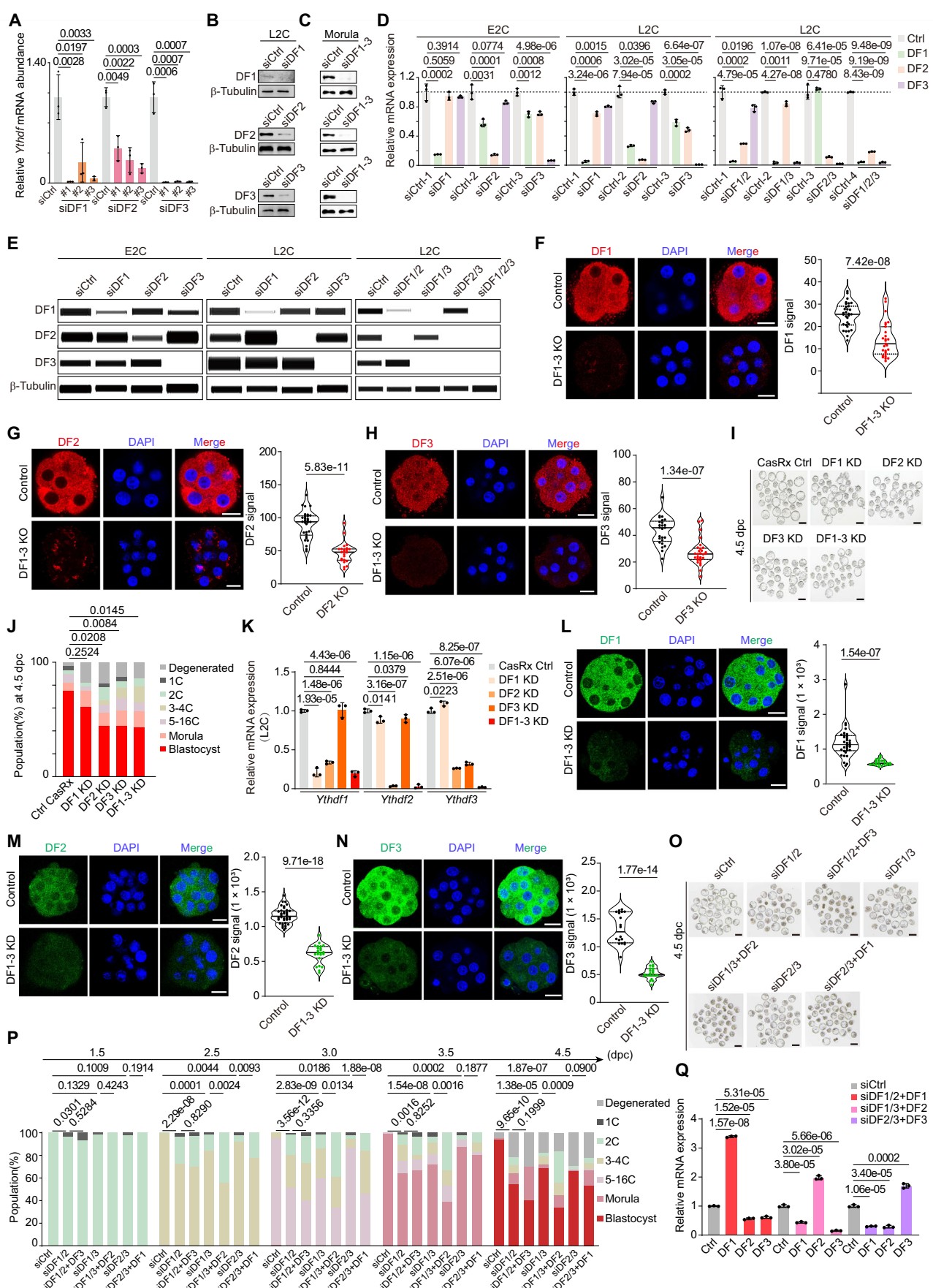

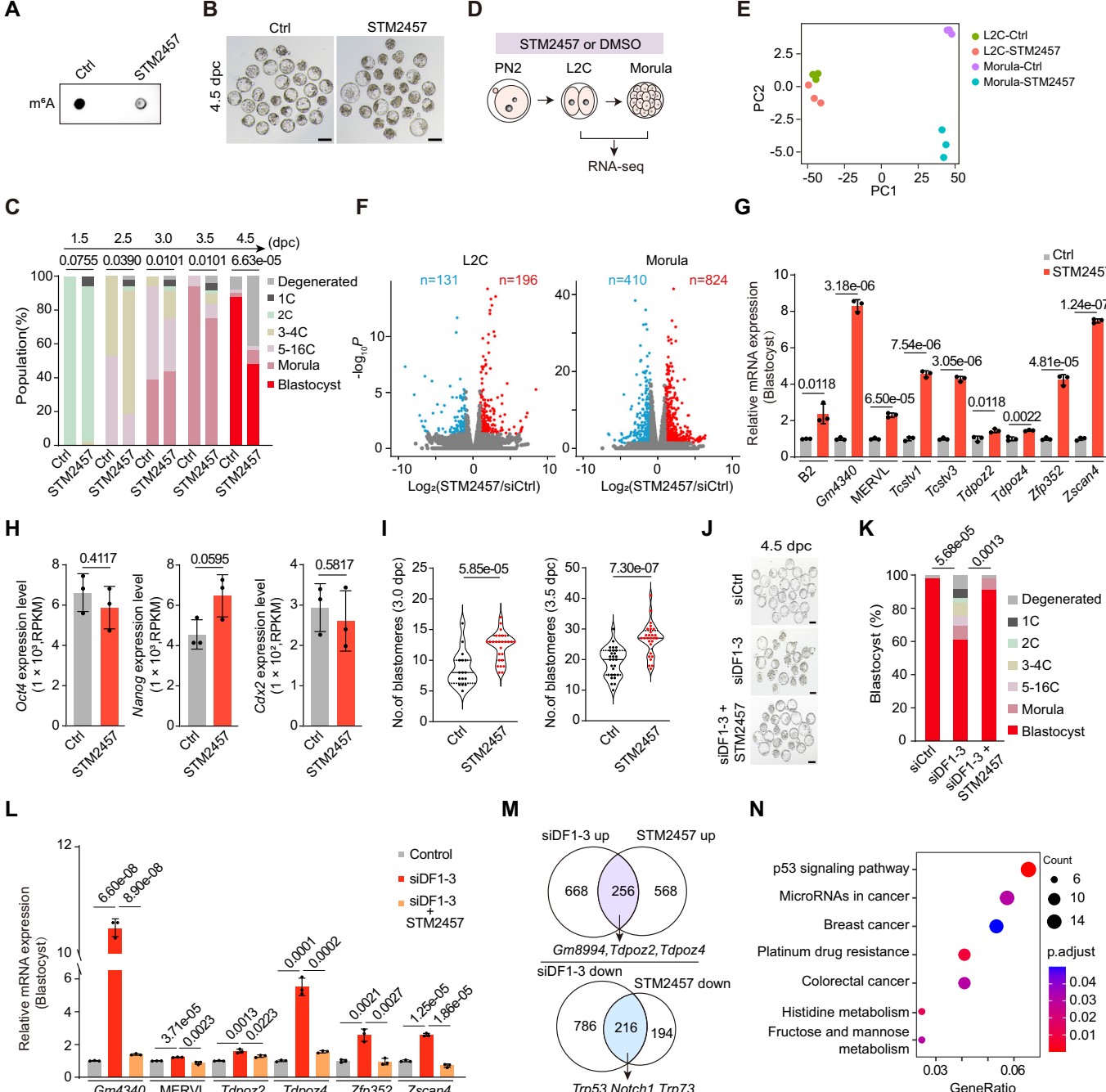

**Figure EV2. Associated with Fig. 1.**

(A) Dot blotting showing the m⁶A level of total RNAs in control and STM2457-treated morula embryos. (B) Representative images showing embryos treated with STM2457 or DMSO control at 4.5 dpc. Scale bars, 100 μm. (C) Percentages of embryonic stages observed at the indicated time points in control ($n = 49$) and STM2457-treated ($n = 48$) groups. $P$ values were determined by the Chi-square test. (D) Schematic of the experimental procedure for detecting the transcriptome changes of STM2457-treated embryos at the L2C and morula stages. (E) PCA showing STM2457-treated and control samples at the L2C and morula stages. (F) Volcano plots showing gene expression changes upon STM2457-treated embryos at the L2C (left) and morula (right) stages (3 biological replicates for each knockdown condition). The $P$ value was determined using DESeq2 (Love et al, 2014), with a threshold of 0.05. (G) Relative 2C gene and B2 RNA expression measured by RT-qPCR at the blastocyst stage. Data are mean ± SD, $n = 3$ biological replicates. (H) Bar chart showing gene expression from STM2457-treated and control RNA-seq data at the morula stage. Data are mean ± SD, $n = 3$ biological replicates. (I) Violin plot showing the total number of blastomeres per embryo in control and STM2457-treated morulae at 3.0 (Ctrl, $n = 16$; STM2457, $n = 27$) and 3.5 (Ctrl, $n = 28$; STM2457, $n = 29$) dpc. The upper and lower dotted lines in the violin plots represent upper and lower quartiles (25th and 75th percentiles), and the center line represents the median. (J) Representative images showing siDF1-3 KD embryos rescued with STM2457 treatment at 4.5 dpc. Scale bars, 100 μm. (K) Percentages of embryonic stages observed at the 4.5 dpc in Control ($n = 47$), siDF1-3 ($n = 36$), and siDF1-3 + STM2457 ($n = 45$) groups. $P$ values were determined by the Chi-square test. (L) RT-qPCR validation of 2C genes in Control, siDF1-3, and siDF1-3 + STM2457 treated embryos at 4.5 dpc. Data are mean ± SD, $n = 3$ biological replicates. $P$ values in (G–I, L) were determined by a two-tailed unpaired $t$ test. (M) Venn diagram showing the overlap of upregulated (upper) and downregulated (bottom) genes identified between si*Ythdf1-3* and STM2457-treated RNA-seq data at the morula stage. (N) GO analysis of common DEGs identified in both si*Ythdf1-3* knockdown and STM2457-treated morula embryos. The $P$ value was determined by Fisher's exact test. Source data are available online for this figure.

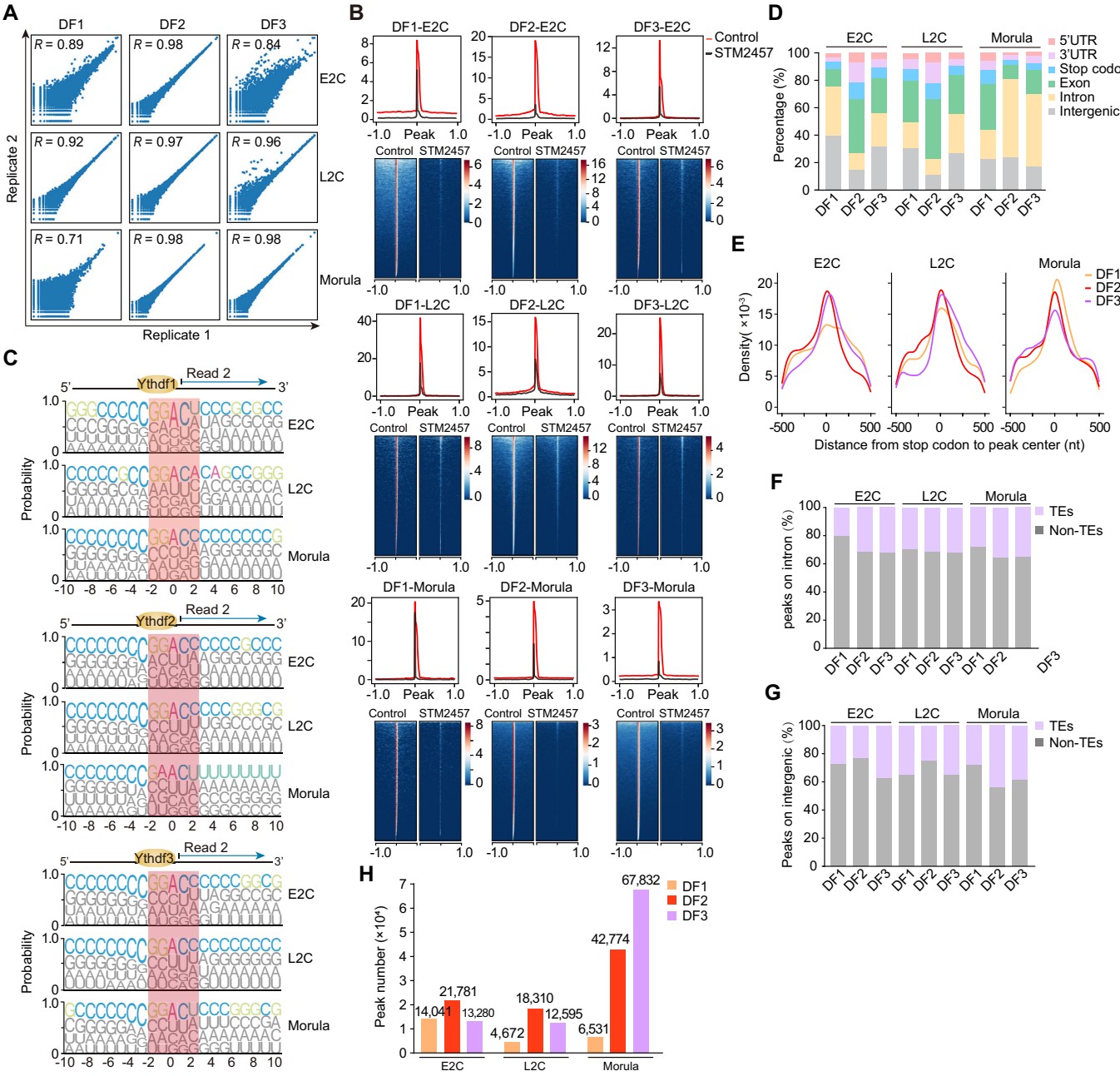

**Figure EV3.  Associated with Fig. 3.**

(A) The correlations of DF1/2/3 LACE-seq between two biological replicates. *R* refers to Pearson's correlation coefficient. (B) Density plots and heatmaps showing the distribution of peak centers within a 1 kb window in the genome for DF1/2/3 LACE-seq data at the E2C, L2C, and morula stages. (C) WebLogo showing the base frequency at and around the DF-RNA crosslinking sites. (D) Bar chart showing the distribution of DF1/2/3 binding peaks in genomic regions at the E2C, L2C, and morula stages. (E) Density plots showing the distribution of peak centers within a 500-nt window in the genome flanking the mRNA stop codon for DF1/2/3 LACE-seq data at the E2C, L2C, and morula stages. (F, G) Bar chart showing the distribution of DF binding peaks upon the intron (F) and intergenic (G) region at the E2C, L2C, and morula stages. (H) Bar plots summarizing the number of DF1/2/3 binding peaks in the E2C, L2C, and morula embryos. Source data are available online for this figure.

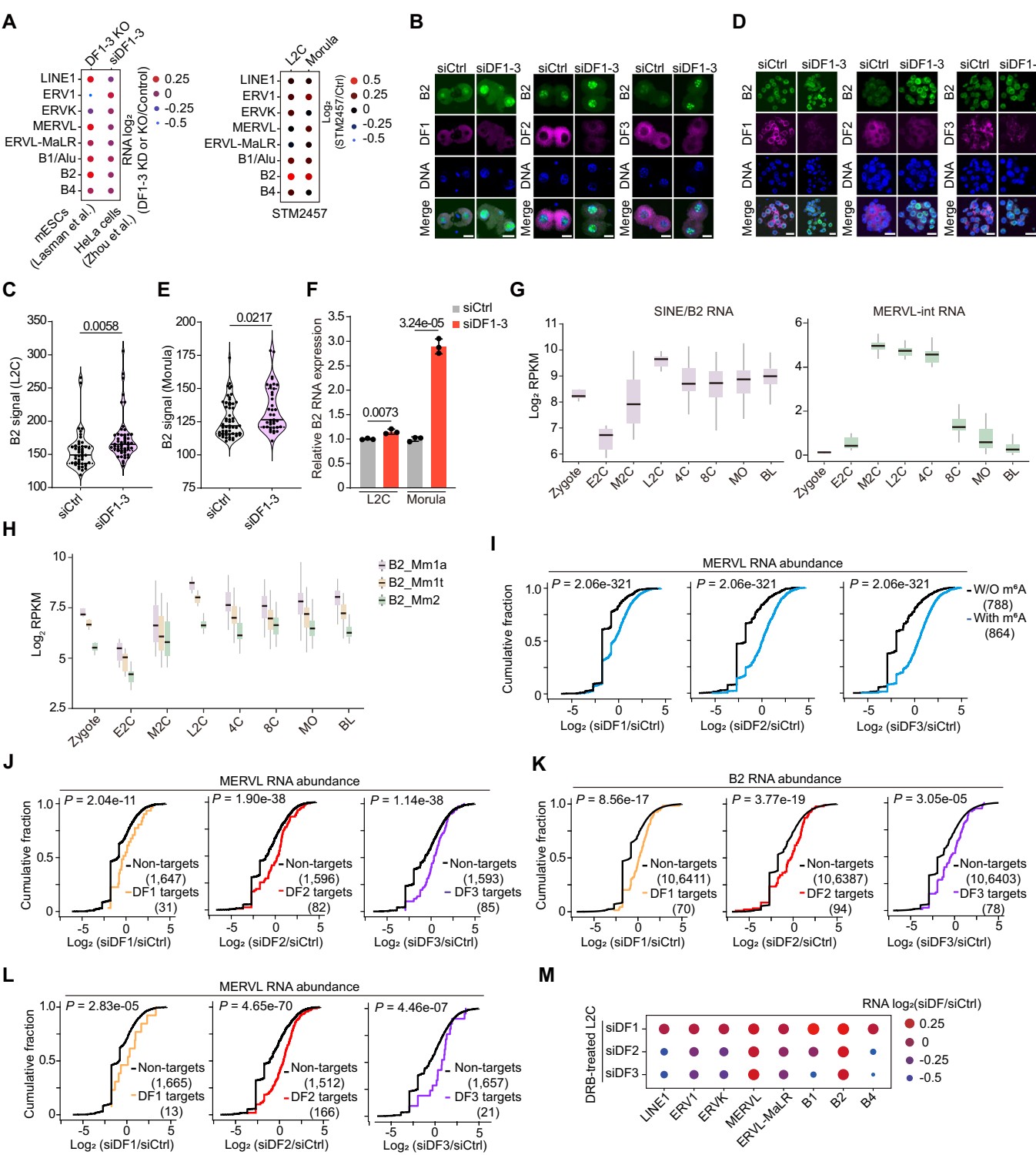

**Figure EV4. Associated with Fig. 4.**

(A) Bubble plot showing changes in TE transcript abundance in siDF1-3 KD HeLa cells, DF1-3 KO mESCs, and STM2457-treated L2C and morula embryos. (B) Representative images of IF/RNA-FISH for B2 RNA with DF counterstain in control and si*Ythdf1-3* knockdown embryos at the L2C stage. Scale bars, 20 μm. (C) Violin plot showing the intensity of the B2 signal in control ($n = 45$) and si*Ythdf1-3* knockdown embryos ($n = 53$) at the L2C stage. The upper and lower dotted lines in the violin plots represent upper and lower quartiles (25th and 75th percentiles), and the center line represents the median. (D) Representative images of IF/RNA-FISH for B2 RNA with DF counterstain in control and si*Ythdf1-3* knockdown embryos at the morula stage. Scale bars, 20 μm. (E) Violin plot showing the intensity of the B2 signal in control ($n = 55$) and si*Ythdf1-3* knockdown embryos ($n = 39$) at the morula stage. The upper and lower dotted lines in the violin plots represent upper and lower quartiles (25th and 75th percentiles), and the center line represents the median. (F) Relative B2 RNA expression measured by RT-qPCR at the L2C and morula stage. Data are mean ± SD, $n = 3$ biological replicates. The *P* values in (C, E, F) were determined by a two-tailed unpaired *t* test. (G) Box plots showing expression levels of the B2 transcript (left) and MERVL-int (right) during preimplantation development. Zygote, $n = 4$; Early 2-Cell (E2C), $n = 8$; Middle 2-Cell (M2C), $n = 12$; Late 2-Cell (L2C), $n = 10$; 4-cell (4C), $n = 14$; 8-cell (8C), $n = 47$; Morula (MO), $n = 58$; Blastocyst (BL), $n = 60$. Boxes represent the 25th-75th percentile (line at the median), with whiskers at 1.5× interquartile range (IQR). (H) Box plots showing expression levels of B2 subfamilies (B2_Mm1a, B2_Mm1t, and B2_Mm2) during preimplantation development. Zygote, $n = 4$; Early 2-Cell (E2C), $n = 8$; Middle 2-Cell (M2C), $n = 12$; Late 2-Cell (L2C), $n = 10$; 4-cell (4C), $n = 14$; 8-cell (8C), $n = 47$; Morula (MO), $n = 58$; Blastocyst (BL), $n = 60$. Boxes represent the 25th-75th percentile (line at the median), with whiskers at 1.5 × interquartile range (IQR). (I) Cumulative distribution of the $\log_2$ fold changes in MERVL abundance between si*Ythdf* and siControl for m6A-modified and unmodified MERVLs in DRB-treated L2C embryos, based on datasets from a previous publication(Wang et al, 2023). *P* values were determined by the Kolmogorov–Smirnov test. (J) Cumulative distribution of the $\log_2$ fold changes in MERVL abundance between si*Ythdf* and siControl for DF1/2/3 LACE-seq targets (E2C) and corresponding non-targets in DRB-treated L2C embryos. *P* values were determined by the Kolmogorov–Smirnov test. (K) Cumulative distribution of the $\log_2$ fold changes in B2 RNA abundance between si*Ythdf* and siControl for DF1/2/3 LACE-seq targets (L2C) and corresponding non-targets in DRB-treated L2C embryos. *P* values were determined by the Kolmogorov–Smirnov test. (L) Cumulative distribution of the $\log_2$ fold changes in MERVL abundance between si*Ythdf* and siControl for DF1/2/3 LACE-seq targets (L2C) and corresponding non-targets in DRB-treated L2C embryos. *P* values were determined by the Kolmogorov–Smirnov test. (M) Bubble plot showing changes in TE transcript abundance (si*Ythdf*/siControl) in DRB-treated L2C embryos. Source data are available online for this figure.

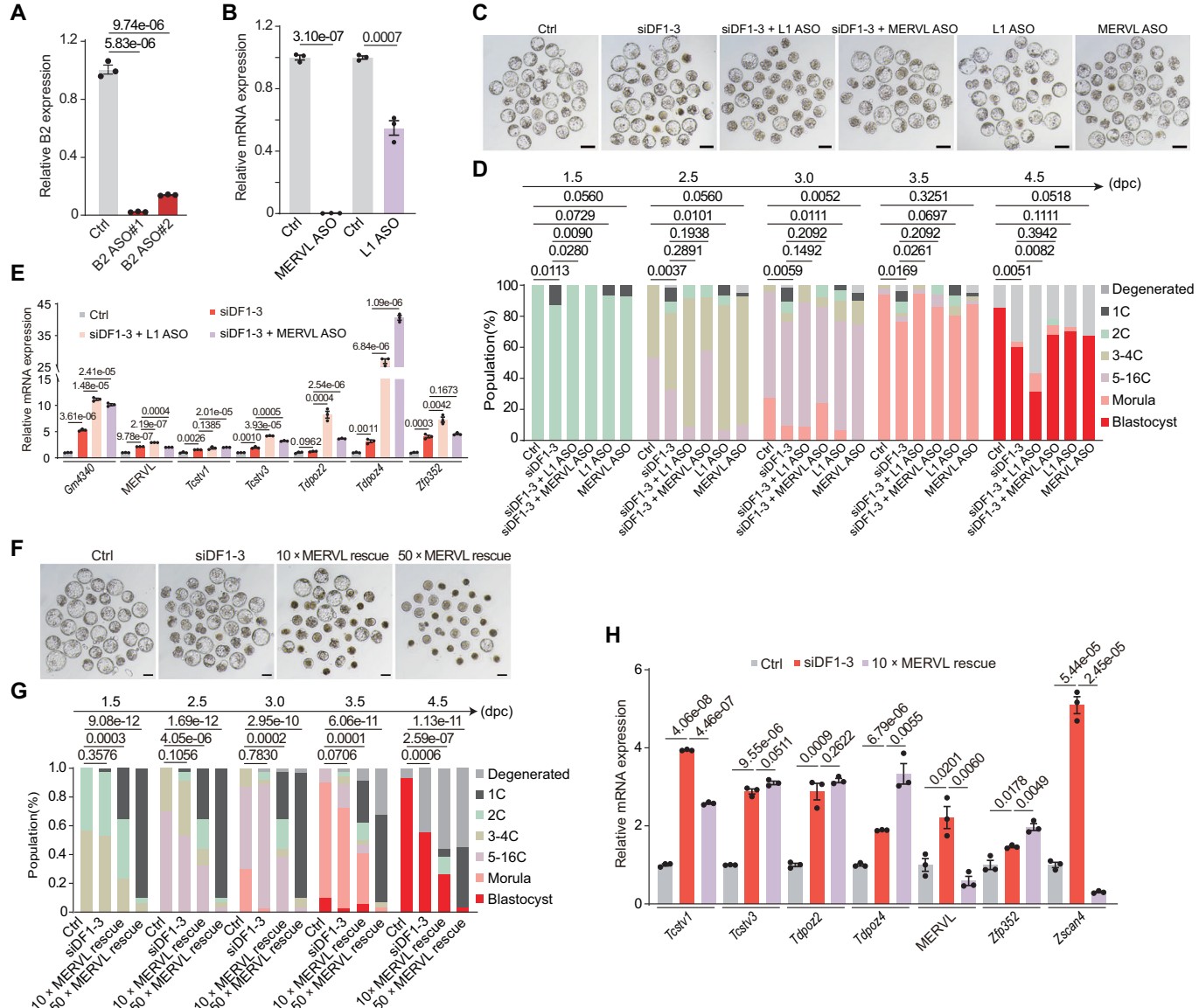

**Figure EV5.** Associated with Fig. 4.

(**A**) RT-qPCR validation of B2 knockdown in L2C embryos. #1#2, two different B2 ASOs. (**B**) RT-qPCR validation of L1 and MERVL knockdown in L2C embryos. We used the same ASO sequences of L1 and MERVL as the public papers (Percharde et al, 2018; Sakashita et al, 2023). (**C**) Representative images showing embryos treated with *Ythdf*1-3 siRNAs, *Ythdf*1-3 siRNAs + L1 ASO (0.2 μM), *Ythdf*1-3 siRNAs + MERVL ASO (0.2 μM), L1 ASO (0.2 μM), MERVL ASO (0.2 μM), or control siRNA at 4.5 dpc. Scale bars, 100 μm. (**D**) Percentages of embryonic stages observed at the indicated time points in Control (*n* = 47), siDF1-3 (*n* = 55), siDF1-3 + L1 ASO (*n* = 35), siDF1-3 + MERVL ASO (*n* = 50), L1 ASO (*n* = 30), MERL ASO (*n* = 40) groups. (**E**) RT-qPCR validation of 2 C genes in control, siDF1-3, siDF1-3 + L1 ASO, and siDF1-3 + MERVL ASO-treated embryos at 4.5 dpc. (**F**) Representative images showing embryos treated with DF1-3 siRNAs, 10×MERVL rescue (siDF1-3 + 2 μM of MERVL ASO), 50×MERVL rescue (siDF1-3 + 10 μM of MERVL ASO), or control siRNA at 4.5 dpc. Scale bars, 100 μm. (**G**) Percentages of embryonic stages observed at the indicated time points in control (*n* = 30), siDF1-3 (*n* = 36), 10×MERVL rescue (*n* = 34), and 50×MERVL rescue (*n* = 31) groups. *P* values in (**D**, **G**) were determined by the Chi-square test. (**H**) RT-qPCR validation of 2 C genes in control, siDF1-3, and 10×MERVL rescue embryos at 4.5 dpc. Data in (**A**, **B**, **E**, **H**) are mean ± SD, *n* = 3 biological replicates. The *P* value was determined by a two-tailed unpaired *t* test. Source data are available online for this figure.

