## [Peer Review File · The EMBO Journal]

m6A reader Ythdf proteins control retrotransposon B2 repeat expression and safeguard early embryo development

Ruibao Su, Di Gao, Zongchang Du, Tie-Gang Meng, Chao Li, Lei-Ning Chen, Xiaoting Lin, Chang-Chang Cao, Li-Hua Fan, Yanbin Dong, Sheng Li, Shi-Ming Luo, Shuai Jiang, Zhong Guo, Yulu Tian, Qing-Yuan Sun, and Xiang-Hong Ou

Corresponding author(s): Ruibao Su (surb@gd2h.org.cn), Qing-Yuan Sun (sunqy@gd2h.org.cn), Xiang-Hong Ou (ouxh@gd2h.org.cn)

Review Timeline:

Transfer Date:	5th Aug 25
Editorial Decision:	1st Oct 25
Revision Received:	27th Nov 25
Editorial Decision:	12th Dec 25
Revision Received:	17th Dec 25
Accepted:	20th Jan 26

Editor: Cornelius Schneider

Transaction Report: This manuscript was transferred to The EMBO Journal following peer review at another journal.

REVIEWER COMMENTS

We sincerely thank the reviewers for their insightful and constructive comments, which have greatly helped us improve the quality of our manuscript. In response to the reviewers' suggestions, we have undertaken additional experiments to validate further the robustness and reproducibility of the embryonic phenotypes observed upon siRNA-mediated knockdown (KD) of YTHDFs. Specifically, we have implemented two complementary approaches to deplete YTHDF expression during preimplantation development independently: (1) CRISPR/Cas9-mediated knockout (KO) via microinjection of Cas9 mRNA and gene-specific guide RNAs into zygotes targeting individual *Ythdf* genes, and (2) CasRx-mediated knockdown of YTHDFs. These results are presented in newly generated figures, including **New Figure 1**, **Extended Figure 2**, and **Extended Figure 3**. These additional approaches consistently recapitulate the developmental phenotypes observed in siRNA-mediated YTHDF-KD embryos, supporting the robustness of our findings. In response to major comment 2 from Reviewer #1, we also reanalyzed our DF1/2/3 LACE-seq datasets in conjunction with the GLORI datasets to assess whether YTHDFs bind to distinct target sites. The updated analysis (**New Figure 3**) demonstrates that YTHDF binding sites are largely shared throughout early development, although the accessibility of m⁶A sites to YTHDF proteins may vary across specific developmental stages. We have revised the manuscript accordingly to reflect these new findings.

Please find our point-by-point responses to all specific comments on the following pages. Reviewers' comments are presented in black, and our responses are highlighted in blue. All corresponding changes in the manuscript are marked in red for clarity.

Reviewer #1 (Remarks to the Author):

This is a very interesting paper about the role of the YTHDF m⁶A-binding proteins in mESC development. Key highlights include experiments supporting functional redundancy of the YTHDF1,2, and 3 proteins (which is still somewhat unresolved), the finding of B2 SINE RNA element RNA elevations as mediating phenotypes of YTHDF depletion (which is very novel).

Generally I think this is a quite good study and the new discovery of B2 RNA expression alterations/regulation by YTHDFs is important and may have broad importance.

We sincerely thank the reviewer for recognizing the significance of our contributions to the study of m⁶A reader function and retrotransposon regulation during early embryonic development. We have carefully addressed all the comments with additional analyses and expanded discussions.

I have some suggestions:

1. Do B2 levels increase with YTHDF depletion in other cell lines or is this a unique mechanism for mESC. It should be easy for the authors to mine existing datasets of YTHDF triple knockdown cells to see if B2 increases (and other SINES, etc) are similarly regulated. Absolute amounts should be compared between mESC and other cell lines that show increases in B2.

We appreciate the reviewer's thoughtful evaluation and constructive suggestions. In response to the reviewer's comment, we reanalyzed existing datasets from YTHDF

triple knockdown (KD) HeLa cells and YTHDF triple knockout (KO) mESCs to assess the expression changes of B2 and other retrotransposons. Our analysis revealed that the levels of B2, MERVL, and LINE1 retrotransposons are elevated in YTHDF triple KO mESCs, but not in YTHDF triple KD HeLa cells (noting that B2 elements are mouse-specific, while human Alu elements are their closest counterparts) (**New Extended Figure 7a**). These findings suggest that YTHDF-mediated regulation of B2 elements may be specific to the mESC context. This new analysis has been incorporated into the revised manuscript.

2. In Fig 3, and ED5, the authors explore the target sites of YTHDF proteins and find distinct targets. This has been a controversial question, so the authors' experiments are useful. However, they use an approach of comparing peak calling rather than using the methods of comparing LACE signals at m6A sites. m6A sites have been called at single-nucleotide resolution using GLORI. I would recommend that the authors take a look at the arguments presented in <https://rnajournal.cshlp.org/content/30/5/468.short> - they should then do their analysis by comparing signals (not called peaks) at m6A sites. I suspect that this analysis will show that YTHDF proteins are indeed binding overlapping sites. If not, then the study mentioned above suggested that gel shift assays be used to show that sites that are specific for YTHDF1 only bind YTHDF1 and not YTHDF2, and YTHDF3, and similar for YTHDF2 and YTHDF3. Since the issue of whether the YTHDFs bind different sites is a quite important issue in the field, it is important that the authors very carefully take into account the criticisms/pitfalls that have been described in this field, and use the methods proposed to definitively validate that YTHDFs have the same or similar sites. I suspect that re-analysis will show largely overlapping sites, and the manuscript should be modified accordingly.

We appreciate the reviewer's careful consideration of our manuscript and constructive suggestions. As suggested by the reviewer, we performed additional analyses by comparing YTHDF binding signals at m⁶A sites identified from the mESC GLORI dataset. Our results revealed that YTHDF proteins exhibit a high degree of correlation with each other at m⁶A sites across the early development (combined data from the E2C, L2C, and morula stages), with Pearson correlation coefficients ranging from $R = 0.48$ to 0.70 (**New Figure 3b**), suggesting that YTHDFs largely bind to overlapping sets of m⁶A sites. However, when analyzing the correlation around m⁶A sites at individual stages (E2C, L2C, or Morula), the correlation coefficients were more variable and relatively lower ($R = 0.30$ to 0.81) (**New Figure 3b**). This indicates that the accessibility of m⁶A sites to DF proteins differs depending on the specific developmental stage, likely due to differences in expression levels and subcellular localization of the DF proteins throughout early embryonic development. Thus, while the YTHDF proteins appear to share similar m⁶A recognition capabilities, their stage-specific accessibility to m⁶A-marked transcripts may explain the distinct phenotypic outcomes observed upon individual YTHDF KD or KO. Specifically, DF2 and DF3 KD/KO result in developmental defects, whereas DF1 KD/KO does not (**Figure 1e**, **New Figure 1i**, and **New Extended Figure 2g**). Moreover, overexpression of a single YTHDF protein fails to rescue the developmental defects caused by KD of the other two YTHDF members (**Extended Figure 2m**). These findings have now been

incorporated into the revised manuscript.

3. I was surprised to see m⁶A and YTHDF binding to B2 since B2 RNAs are made from a Pol III promoter and Pol III RNAs are rarely modified with m⁶A, especially in a METTL3-dependent manner (as suggested by the authors' use of the METTL3 inhibitor). Can the authors show the sequences and the location of the putative m⁶A site, and preferably provide methylation stoichiometries based on published GLORI datasets of mESC?

We thank the reviewer for these comments. Given that SINE/B2 elements are multicopy sequences within the genome, we analyzed five representative subclasses of the B2 family derived from distinct genomic loci. For each subclass, we identified putative m⁶A sites and quantified their methylation stoichiometries using published GLORI datasets from mESCs. Our analysis revealed that B2 transcripts are highly methylated at these genomic loci, and the consensus sequence surrounding the putative m⁶A sites conforms to the canonical DRACH motif (New Figure 4d), a known substrate sequence for the Mettl3 methyltransferase. These results have been included in the revised manuscript.

Minor

1. Did I miss western blots to show degree of FTO knockdown - it would be good to show these. RNA-seq measurements for each YTHDF at each time point that was analyzed should be mentioned/shown in a bar graph. The authors should already have this data (and some was shown in Fig 2).

We thank the reviewer for pointing this out. As this manuscript primarily focuses on the role of YTHDF proteins in early embryos and does not include analyses related to FTO, we assume the reviewer's comment pertains to the western blot results of YTHDF proteins. We have demonstrated the extent of YTHDF knockdown at the L2C and morula stages in Extended Figure 1b and New Extended Figure 1c. In addition, in line with the reviewer's suggestion, we have now included bar graphs showing the expression dynamics of each *Ythdf* gene at corresponding developmental time points, based on our RNA-seq datasets (New Extended Figure 3a and 3b).

Reviewer #2 (Remarks to the Author):

N⁶-methyladenosine (m⁶A) is a crucial RNA modification that plays a significant role in development and disease. The YT521-B homology domain-containing family (YTHDF), consisting of YTHDF1, YTHDF2, and YTHDF3, functions as m⁶A readers by binding to m⁶A-modified RNAs. Recent in vivo studies using *Ythdf* knockout (KO) mice have demonstrated that *Ythdf2* KO results in embryonic lethality, whereas *Ythdf1* KO and *Ythdf3* KO mice remain viable. Single and triple *Ythdf* KOs further reveal functional redundancy among YTHDF1, YTHDF2, and YTHDF3 during early development. However, whether these three YTHDF proteins exhibit functional redundancy from the zygote to morula stage remains unclear. Given the limited availability of material at these early developmental stages, the authors employed LACE-Seq, a technique developed in their previous study, to identify YTHDF-bound transcripts during early embryogenesis. By integrating YTHDF LACE-Seq data with differential gene expression analyses following YTHDF depletion, the authors identified transposable elements, maternal RNAs, and mid-preimplantation gene activation (MGA) RNAs as direct targets of YTHDF proteins.

Focusing on the SINE B2 family of transposable elements, they further suggested that upregulation of B2 RNAs modulates early embryogenesis, as evidenced by gain- and loss-of-function experiments. Mechanistically, B2 RNAs might regulate RNA polymerase II (Pol II) transcriptional activity, influencing the expression of genes essential for embryonic development. This study presents extensive data that provide significant insights into (1) the function and target specificity of YTHDF proteins in early embryogenesis and (2) the regulatory role of B2 RNAs in early development. However, several major concerns need to be addressed before publication. These include issues related to knockdown specificity, YTHDF protein subcellular expression and its correlation to the observed phenotypes, and the presentation and interpretation of statistical analyses. Below, I outline my specific critiques:

We sincerely thank the reviewer for their thorough evaluation and insightful feedback on our manuscript. In response to the reviewer's suggestions, we have conducted additional experiments and expanded the discussion to address all the points raised.

Major

1. Immunostaining and RNA-FISH data (Extended Data Fig. 1c, 1e, 1g, 7a, and 7c) clearly indicate cytoplasmic localization of YTHDF proteins, with nuclear segregation of B2 RNAs. The authors also note that B2 RNAs are present in both the cytoplasm and nucleus at the late morula stage (lines 358–361). However, a major concern is how a predominantly cytoplasmic YTHDF protein can regulate nuclear B2 RNA function prior to the morula stage. If the authors conducted functional assays in early morula, it raises the question of how cytoplasmic YTHDF proteins could affect nuclear B2 RNAs, as depicted in their model. Additionally, in lines 365–377, the authors examine YTHDF functional effects at the 2-cell stage, where B2 RNAs are nuclear, while YTHDF proteins remain largely cytoplasmic. This discrepancy warrants clarification—what is the proposed mechanism by which cytoplasmic YTHDF proteins influence nuclear B2 RNAs at this stage?

We apologize for the lack of clarity in our previous explanation. As shown in **Extended Figure 7g**, B2 transcription is inactive before zygotic genome activation (ZGA). Before ZGA, cytoplasmic DF proteins mediate B2 RNA degradation, contributing to its downregulation (**Figure 4f and g**, and **Extended Figure 7k**). After ZGA, B2 transcription occurs concurrently with its degradation. In this context, nuclear B2 RNA colocalizes with RNA polymerase II, suggesting a role in transcriptional regulation (**Figure 5a**), whereas cytoplasmic B2 RNA colocalizes with DF proteins and undergoes degradation at the morula stage (**Extended Figure 7d**). We acknowledge that our initial data presentation may have overemphasized B2 signals in the nucleus while underrepresenting cytoplasmic localization in the immunostaining and RNA-FISH data. We have now updated these datasets (**Figure 4e** and **Extended Data Figure 7b**) and revised the corresponding text to reflect a more balanced view of B2 RNA localization: “We observed that B2 RNA exhibited nuclear enrichment with cytoplasmic distribution up to the morula stage and became evenly distributed between the nucleus and cytoplasm at the blastocyst stage.”

2. Most of the major experiments rely on siRNA-mediated knockdown, which raises concerns regarding off-target effects. Standard siRNA controls, such as scrambled siRNA

and a second independent siRNA, are essential to validate the specificity and efficiency of knockdown. Furthermore, siRNA injection at the PN2 stage may result in diluted knockdown efficiency by the 2-cell to morula stage, making it difficult to conclusively attribute later phenotypes to early knockdown effects. Another issue is the potential fragility of cultured morula-stage embryos, which often display spontaneous defects even without manipulation. The study utilizes triple siRNA knockdown along with B2 ASO treatment, creating a highly stressful condition for the embryos. This raises concerns about whether the observed phenotypes truly reflect YTHDF depletion or are instead a consequence of cumulative stress. Given that YTHDF KO mice are available, the authors should consider using KO embryos as a proof-of-principle experiment to support their findings.

We thank the reviewer for the thoughtful evaluation of our manuscript and the constructive comments provided. In the original version, we employed siRNA microinjection to achieve *Ythdf* knockdown (KD), which raised concerns about potential off-target effects. To address this, we have now conducted additional experiments using both Cas9-mediated KO and CasRx-mediated KD approaches as controls for siRNA-mediated KD. These new results confirm that microinjection of *Ythdf*-targeting guide RNAs (either individually or in combination) into zygotes leads to defects in blastocyst formation when YTHDF2, YTHDF3, or YTHDF1-3 are disrupted, but not when YTHDF1 alone is targeted (New Figure 1h-k, New Extended Figures 2c-k, and New Extended Figures 3h and j). These findings support the conclusion that the developmental phenotypes observed in siRNA-mediated YTHDF-KD embryos are unlikely to be artifacts caused by off-target effects.

Furthermore, to rule out any manipulation-induced effects, we used scrambled siRNA as the negative control in all zygote microinjection experiments. Both control and YTHDF-targeted groups received injections of 10 pl of siRNA solution at the same concentration (30 μ M), ensuring that the embryos were exposed to equivalent experimental conditions. We also performed western blot analysis on morula-stage embryos following triple *Ythdf* knockdown (siDF1-3), which confirmed high knockdown efficiency for each YTHDF protein at this stage (New Extended Figure 1c). This suggests that YTHDF proteins remain effectively depleted from the 2-cell stage to the morula stage.

Together, these results provide robust validation that the phenotypes observed in siRNA-mediated YTHDF knockdown embryos are reliable and not due to technical artifacts.

3. The conclusions drawn from the data remain somewhat unclear. In Fig. 1, the authors suggest that YTHDF2 and YTHDF3 exhibit dynamic expression from the 2-cell to morula stage, implying they are key regulators of B2 RNA. However, later in the study, they focus on YTHDF1 as the major regulator, despite its relatively constant expression. This shift in focus needs clearer justification. Additionally, the conclusion oscillates between B2 RNA function and cytoplasmic YTHDF function in cis and trans, but the causal link remains weak. The most critical dataset in this manuscript is presented in Fig. 5, yet its interpretation and presentation are complex. A more streamlined and structured explanation of Fig. 5, aligning it clearly with the proposed model, would significantly

improve the manuscript's clarity.

We apologize for any confusion. As highlighted in previous studies, the cellular functions of YTHDF proteins (YTHDF1, YTHDF2, and YTHDF3) remain a subject of ongoing debate. The Canonical Model proposes distinct roles: YTHDF1 enhances translation (Cell, 2015; PMID: 26046440), YTHDF2 promotes mRNA decay (Nature, 2014; PMID: 24284625), and YTHDF3 supports both translation and decay (Cell Research, 2017; PMID: 28106072). In contrast, the Unified Model suggests that all three YTHDFs redundantly bind m⁶A-modified transcripts to promote degradation (Cell, 2020; PMID: 32492408; Genes & Development, 2020; PMID: 32943573). Central to this debate is whether YTHDF1, like YTHDF2 and YTHDF3, contributes to transcript decay. One of our key questions was to determine whether YTHDF1 facilitates degradation of its m⁶A-modified targets during early mouse embryonic development. Our findings demonstrate that all three YTHDF proteins can promote transcript degradation, though m⁶A site accessibility to each YTHDF may vary at specific developmental stages.

In response to the reviewer's criticism regarding the interpretation of B2 RNA function in Fig. 5, we have revised our presentation for improved clarity. Specifically, we have separated the *trans* and *cis* functions of B2 RNA into two figures: **New Fig. 5** and **New Fig. 6**. In **New Fig. 5**, we demonstrate that the upregulated B2 RNA acts in *trans* to repress RNA Polymerase II (Pol II) transcription, contributing to gene downregulation in siDF1-3 embryos. In contrast, **New Fig. 6** illustrates that DF1-3 knockdown leads to reduced B2 RNA transcription in *cis*, which in turn diminishes Pol II activity near SINE/B2 loci. These changes help streamline the mechanistic interpretation and have been incorporated into the revised manuscript.

4. Most qPCR and many statistical analyses seem to use technical replicates with SEM analysis; it is suggested that biological replicates be used with SD analysis.

We thank the reviewer for pointing this out. We have updated all relevant experiments involving biological replicates to present the data as mean \pm standard deviation (SD), and we have included this information in the corresponding figure legends.

This study provides valuable insights into m⁶A regulation in early embryogenesis, but addressing the concerns above will be crucial for strengthening the mechanistic conclusions. Ensuring consistency in the proposed model, incorporating appropriate controls for knockdown experiments, and clarifying the interaction between cytoplasmic YTHDF proteins and nuclear B2 RNAs will greatly enhance the impact of this work. We appreciate the reviewer's positive comments and constructive suggestions. In response, we have newly conducted (1) Cas9-mediated KO and (2) CasRx-mediated KD of YTHDFs as complementary approaches to validate the embryonic phenotypes observed in the siRNA-mediated KD experiments. Additionally, we have updated the immunostaining and RNA-FISH data to more accurately reflect the cytoplasmic distribution of B2 signals, in addition to their nuclear enrichment. We have also clarified the interaction between cytoplasmic YTHDF proteins and cytoplasmic B2 RNAs. These improvements have been incorporated into the revised manuscript as recommended.

Minor

1. It is recommended to include an experimental workflow in Figure 1, similar to Figure 2a, to provide a clear overview of the study design. This would help readers better understand the sequencing and logic of experiments.

We thank the reviewer for this suggestion. We have added an experimental workflow in Figure 1 (New Figure 1c).

2. Please include the n numbers in the Extended Data figures to ensure transparency regarding sample size and statistical robustness.

We thank the reviewer for pointing this out. We have included the n numbers in the Extended Data figures and Source Data file.

3. In Extended Data Figure 2b, the term “Wes” immunoblotting assays should be corrected to “Western” immunoblotting assays for accuracy. Additionally, the Western blot bands appear atypical compared to those commonly presented in the literature. To ensure data integrity, the authors should provide the original, unprocessed blots as supplementary data.

We apologize for the lack of clarity. To detect YTHDF proteins in early embryos, we employed the Simple Western™ Wes system, which utilizes capillary electrophoresis for protein separation and presents results in a digital format, differing from traditional Western blotting techniques. This method is particularly suitable for low-input samples and has been widely adopted in such contexts (*Nature Methods*, 2011, <https://doi.org/10.1038/nmeth.f.353>). We have included a detailed description of the Simple Western procedure in the Methods section and provided the original blot data in the Source Data file.

4. In lines 295–296, the authors state that they will focus on YTHDF1 in subsequent analyses. However, based on the data, YTHDF1’s function does not appear to be clearly distinct from that of other YTHDF proteins. For instance, Extended Data Figure 7i shows that knockdown of any YTHDF leads to upregulation of MERVL and B2 RNA, suggesting functional redundancy among the three proteins. The authors should clarify why YTHDF1 is prioritized and provide additional evidence to support its unique role, if applicable.

We apologize for the lack of clarity. Please refer to our response to Major Question #3 from Reviewer #2. The cellular functions of YTHDF proteins (YTHDF1, YTHDF2, and YTHDF3) remain a topic of active debate. The Canonical Model suggests that YTHDF1 enhances translation of its mRNA targets (*Cell*, 2015, PMID: 26046440), YTHDF2 promotes mRNA decay (*Nature*, 2014, PMID: 24284625), and YTHDF3 supports both translation and degradation (*Cell Research*, 2017, PMID: 28106072). In contrast, the Unified Model posits that all three YTHDF proteins primarily mediate degradation of m⁶A-modified transcripts through binding to overlapping m⁶A sites (*Cell*, 2020, PMID: 32492408; *Genes & Development*, 2020, PMID: 32943573). Central to this controversy is whether YTHDF1, like YTHDF2 and YTHDF3, also facilitates transcript degradation.

Our study specifically addresses this question in the context of mouse early embryonic development. Our data show that all three YTHDF proteins can promote the degradation of their target transcripts (Figure 3k and l). Although m⁶A site accessibility may vary for each YTHDF protein at different developmental stages, we observed that any YTHDF can bind MERVL and B2 RNAs and redundantly mediate

their degradation (Figure 4f and g; Extended Figure 7i-l). We have now incorporated this interpretation into the revised manuscript.

5. The manuscript benefits from clear labeling of developmental stages and time points, which improves figure readability. However, some panels could be further clarified:

- Figure 1c: Adding “4.5 dpc” on the left side would clearly indicate the developmental stage.

We thank the reviewer for pointing this out. We have now added “4.5 dpc” to the left side of Figure 1d as suggested.

- Other time points: Consistently label time points in similar experiments throughout both the main and Extended Data figures.

we have consistently labeled the time points in similar experiments across both the main and Extended Data figures (Figure 1d, New Figure 1h, Figure 4h, New Extended Figure 2f and i, Extended Figure 4b, New Extended Figure 4k, Extended Figure 8a, and Extended Figure 9c and f).

- Cumulative distribution plots (Extended Data Fig. 7h-k): Clearly specify what is being compared by adding titles such as “MERVL abundance” at the top of the respective panels.

We thank the reviewer for the helpful suggestion. We have added titles to the top of the respective cumulative distribution plots in Figure 4f and 4g, as well as in Extended Data Figure 7i-l.

6. The authors state that *Ythdf* triple-KO embryos form more blastocysts (Fig. 1e), yet in line 186, they mention that *Ythdf* triple-KO results in the lowest blastocyst formation rate (36%). This apparent contradiction needs clarification. A similar issue arises in line 238, where treatment with STM2457 is said to reduce blastocyst formation, yet increase blastomere number (line 244). The authors should clarify whether these observations reflect differences in embryonic viability, cell proliferation, or other developmental parameters.

We thank the reviewer for pointing this out. Based on our results from siRNA-mediated *Ythdf* triple knockdown (KD) embryos, we observed that *Ythdf* triple KD leads to the lowest blastocyst formation rate (36%) at 4.5 dpc and results in increased blastomere numbers per embryo at 3.0 and 3.5 dpc (Figure 1d-g). These findings suggest a possible link between accelerated blastomere proliferation and impaired blastocyst formation. Furthermore, we observed similar phenotypes in embryos treated with STM2457, including reduced blastocyst formation at 4.5 dpc and increased blastomere numbers at 3.0 and 3.5 dpc (Extended Figure 4b, c, and j). Transcriptomic analysis of *Ythdf* triple KD and STM2457-treated morulae revealed commonly downregulated genes such as *Trp53*, *Trp73*, *Ccng1*, and *Cdkn1a*, which are associated with the p53 signaling pathway (New Extended Figure 4n and Extended Figure 4o). This suggests a potential mechanism whereby suppression of p53 signaling may promote cell proliferation and accelerate blastomere division. We have included this interpretation in the revised manuscript.

7. In line 226, the authors claim that YTHDF proteins do not influence early embryonic lineage differentiation based on mRNA and protein expression analysis. However, this contrasts with the findings of Lior et al. (doi: 10.1101/gad.340695.120), who reported that

Ythdf triple-KO (tKO) in mESCs results in higher expression of pluripotency markers and poor differentiation ability. Could the authors reconcile these differences? Potential factors such as experimental conditions, differences between ESCs and embryos, or the timing of YTHDF depletion should be discussed. This, again, caused the worrisome of siRNA off target and stressful condition for PN2 injection.

We thank the reviewer for pointing this out. Previous studies have shown that m⁶A-marked mRNAs are critical for the transition from naïve pluripotency to differentiation during post-implantation stages (e.g., E5.5 embryos and mESCs). Specifically, *Mettl3* KO blastocysts at E3.5 display normal morphology and expression of pluripotency markers and do not exhibit defects in early embryonic lineage differentiation. However, developmental abnormalities become apparent at post-implantation stages (E5.5-E7.5) (Science, 2015, PMID: 25569111). Similarly, *Ythdf* triple-KO embryos fail to develop beyond E7.5. *Ythdf* triple-KO mESCs also exhibit a “hyper-pluripotency” phenotype—characterized by elevated expression of pluripotency markers and impaired differentiation potential—resembling the phenotype observed in *Mettl3*-KO mESCs (Genes & Development, 2020, PMID: 32943573). Therefore, our observations that both DF1-3 depletion and METTL3 inhibition do not affect early lineage differentiation before post-implantation are consistent with prior findings. These results collectively suggest that *Ythdf* and *Mettl3* proteins may serve distinct regulatory roles in pre-implantation embryos versus mESCs.

Additionally, to corroborate the siRNA-mediated knockdown (KD) results, we have newly performed (1) Cas9-mediated KO of YTHDFs and (2) CasRx-mediated KD of YTHDFs as alternative approaches. Notably, CasRx-mediated *Ythdf* triple-KD morulae showed a phenotype comparable to that observed in siRNA-mediated KD, with no significant change in the expression of the pluripotency marker Oct4 compared to control embryos (New Extended Figure 3o). We have added this discussion to the revised manuscript.

8. Some figure annotations are unclear, requiring frequent reference to figure legends. Improving the labeling of axes, conditions, and key comparisons would significantly enhance clarity and accessibility for readers.

We apologize for the lack of clarity. We have carefully reviewed all figures and ensured that each panel is appropriately labeled. Additional labels have been added where necessary to improve clarity and presentation. We hope these revisions will facilitate a better understanding and interpretation of the data.

9. In line 271 and Extended Data Fig. 5d, the authors state that DF1/2/3 peaks are enriched around stop codons. However, the data show binding enrichment across intronic, exonic, and intergenic regions. The authors should clarify this discrepancy and provide further validation of m⁶A-binding sites.

Thank you for pointing this out. One critical factor influencing the distribution of m⁶A is the RNA type used for detection. Previous studies have shown that more m⁶A sites are found in promoter, intronic, and intergenic regions in total RNA samples than in polyA⁺ RNA samples (*Nature Methods*, 2015, PMID: 26121403). Thus, while m⁶A modifications can occur throughout the transcriptome, they are typically enriched in

long coding exons and the 3' UTRs of polyA⁺ RNAs.

In our study, DF1/2/3 LACE-seq was performed on total RNA from cell lysates, which allowed us to capture m⁶A peaks and DF1/2/3 binding events across exonic, intronic, and intergenic regions. A major contributor to the observed DF1/2/3 enrichment in intronic and intergenic regions is the abundant expression of retrotransposon RNAs during early embryonic development. These transposable elements (TEs), which often reside in intronic and intergenic loci, represent a substantial portion of DF-bound RNAs. Statistical analysis showed that 20%-44% of DF1/2/3 binding peaks are located on TEs within these regions (New Extended Figure 5f and g).

To further examine the relationship between DF binding and m⁶A, we compared the DF1/2/3 LACE-seq datasets with picoMeRIP-seq m⁶A datasets on non-intronic transcripts. We observed a reciprocal overlap of approximately 65%-89% between DF binding peaks and m⁶A-modified sites (Extended Figure 6a), reinforcing that DF protein binding is strongly m⁶A-dependent. These findings have now been incorporated into the revised manuscript.

10. The study would benefit from integrating previously published pico-MeRIP-seq data (<https://doi.org/10.1038/s41594-023-00969-x>) to compare m⁶A-modified vs. unmodified sites in embryos. This would strengthen the m⁶A-binding site analysis and help verify whether DF-binding regions correspond to m⁶A modifications.

We thank the reviewer for the helpful suggestion. Please refer to our response to Minor Question #9 from Reviewer #2. As part of our analysis, we examined the overlap between DF1/2/3 LACE-seq datasets and picoMeRIP-seq datasets on non-intronic transcripts. Our results showed a reciprocal overlap of approximately 65%-89% between DF protein binding peaks and m⁶A-modified sites (Extended Figure 6a). Furthermore, 69%-80% of DF-targeted genes harbored picoMeRIP-seq m⁶A peaks (Extended Figure 6b), providing strong evidence that the function of YTHDF proteins is largely m⁶A-dependent. These findings have been included in the revised manuscript.

11. Figure 4b does not clearly indicate which RNAs are most abundant upon Ythdf loss. Similar data presentation issues are observed in Extended Data Fig. 4c, 5d, and 8b. The authors should refine these figures to improve clarity and interpretability. We thank the reviewer for pointing this out. To enhance clarity, we have exchanged Figure 4a and Figure 4b and revised the corresponding sentence to: “Although DF proteins broadly target retrotransposon RNAs, B2 RNAs, as well as the 2C-specific transposon MERVL, were markedly upregulated in si*Ythdf1-3* knockdown morula embryos.”

To improve the interpretability of Extended Data Figure 4c, we have revised the sentence to: “To this end, we employed a METTL3 inhibitor (STM2457) to treat embryos, which greatly reduced m⁶A levels and blastocyst formation (Extended Data Fig. 4a-c). Next, we conducted RNA-seq analysis on STM2457-treated and control embryos at the L2C and morula stages (Extended Data Fig. 4d-f).”

For Extended Data Figure 5d, we added the following clarification: “It is noteworthy that the DF1/2/3 LACE-seq libraries were generated using total RNA from cell lysates,

allowing the detection of DF1/2/3 binding events across exonic, intronic, and intergenic regions. This is consistent with previous findings showing that total RNA samples, as opposed to poly(A)⁺ RNA, contain a greater proportion of m⁶A sites within promoter, intronic, and intergenic regions”.

Regarding **Extended Data Figure 8b**, we have added the following explanation: “As expected, embryos microinjected with 1× B2 RNA (without m⁶A modification, ~170 ng/μl) exhibited notable developmental delays beginning at 1.5 dpc and a significant reduction in blastocyst formation (46.6%). In contrast, embryos injected with a lower concentration of B2 RNA showed milder developmental delays starting at 2.5 dpc, suggesting that the phenotype induced by B2 RNA is dose-dependent (Extended Data Fig. 8a-c). Next, we conducted RNA-seq analysis on B2 overexpression (OE) and control embryos at the L2C and morula stages (Extended Data Fig. 8d-f).”

Reviewer #3 (Remarks to the Author):

RNA epigenetic regulation plays significant roles in mammalian early embryonic development. The study focuses on DF1/2/3 proteins, key cytosolic m⁶A-binding factors, in mediating RNA decay during mouse embryonic development. The strengths of this study includes: DF1/2/3 target identification by LACE-seq and PolIII, repeat RNA interference by ASO to study their roles in the early embryos, and PolIII mapping in the *Ythdf3* 1-3 KD embryos. However, this study fall short in the following aspects: siRNA KD models used rather than previously established KO models, specificity and reproducibility of KD models, and some conclusions not supported by the data presented. We appreciate the reviewer’s insightful comments. In response, we have conducted and included multiple additional experimental panels to reinforce the embryonic phenotypes observed upon siRNA-mediated KD of YTHDFs. These supplementary data support a more rigorous and well-substantiated interpretation of our findings and conclusions.

Here are some specific concerns for consideration.

1. Line 104, the CRE line used in the cKO shall be specified.

We thank the reviewer for pointing this out. We have revised the sentence to include the CRE line information, which now reads: “In mice, conditional knockout (cKO) of *Ythdf2* (*Ythdf2*^{HA-FI/HA-FI}; *Zp3*-Cre) leads to female-specific infertility, and the embryos mainly arrest at the 2-cell (2C) stage.”

2. Line 110 - 112, the sentence that summarizes the key finding of the manuscript is hard to follow.

Please make it clearer.

We thank the reviewer for pointing this out. We have revised the sentence to read, “Here, we present an expanded model in which all *Ythdf* proteins mediate the decay of their target retrotransposon RNAs and mRNAs. The accessibility of m⁶A sites to individual *Ythdf* proteins varies across specific developmental stages, and their coordinated regulation is essential for proper progression of pre-implantation development.”

3. Figure 1A, B, it is recommended to also include oocytes for the profiles. What does the number in Y-axis mean?

Thank you for your suggestion. We have now included oocytes in the profiles shown in Figure 1a and 1b. The Y-axis in Figure 1a represents the expression levels of *Ythdf1/2/3* from SMART RNA-seq datasets of oocytes and early embryos, as quantified using the DESeq2 software package. In Figure 1b, the Y-axis indicates the translation intensities of *Ythdf1/2/3* based on Ribo-lite datasets from oocytes and early embryos, measured as ribosome-protected fragments (RPFs) (Nature Cell Biology, 2022, PMID: 35697785). We have updated the Y-axis labeling in Figures 1a and 1b accordingly.

4. The role of *Ythdfs* in early embryonic development has been previously shown by KO and cKO mouse models. This study, however, use KD models, which might comprise its impact to the field.

Thank you for the thoughtful comment. While *Ythdf* KO and cKO mouse models have been previously reported, their prolonged genetic deletion, starting from gametogenesis or early ontogeny, makes it difficult to dissect the specific roles of *Ythdf* genes during preimplantation embryonic development. In contrast, *Ythdf* KD models offer a temporal advantage, allowing us to investigate *Ythdf* function specifically during defined early embryonic stages. Moreover, the developmental defects associated with previously reported *Ythdf* KO and cKO models have limited their utility for generating sufficient embryos or conducting mechanistic studies at various early developmental stages.

Based on our *Ythdf* KD models, we demonstrated that all three *Ythdf* proteins (*Ythdf1*, *Ythdf2*, and *Ythdf3*) promote the decay of both mRNAs and retrotransposon RNAs, with their access to m⁶A sites varying across specific developmental stages. This provides an expanded model for understanding the distinct and overlapping functions of *Ythdf1/2/3* during early embryogenesis. Notably, we found that *YTHDF1/2/3*-mediated decay of SINE/B2 RNAs is critical for proper early embryonic development. This finding is reminiscent of recently reported roles of m⁶A-modified retrotransposon RNAs in mouse embryonic stem cells (*Nature*, 2021, PMID: 33658714; *Nature*, 2021, PMID: 33442060; *Science*, 2020, PMID: 31949099). Together, our results highlight the essential regulatory role of m⁶A-modified B2 RNAs—distinct from LINEs and LTRs—in modulating the developmental potential of early embryos.

5. The siRNA mediated KD of *Ythdfs* were carried out by zygote injection. Since both the mRNA level and translation of *Ythdf1* are minimal. It is hard to imagine how siRNAs would affect the protein level of *Ythdf1*. This raises significant concerns about the validity of the KD approach used in this study.

Thank you for the thoughtful comment. Although *Ythdf1* exhibits relatively lower mRNA expression and translation levels compared to *Ythdf2* and *Ythdf3* (Figure 1a and b), we confirmed efficient *Ythdf1* knockdown at both the transcript and protein levels. RT-qPCR and RNA-seq analyses demonstrated substantial reduction of *Ythdf1* mRNA following siRNA treatment (Extended Figure 1a, Extended Figure 2a, and New Extended Figure 3a and b). In addition, multiple approaches, including Western blot, immunofluorescence, and the Simple Western™ Wes system, validated efficient knockdown of *Ythdf1* protein across various developmental stages (Extended Figure

1b, d, and e, New Extended Figure 1c, and Extended Figure 2b). Collectively, these results provide strong evidence that siRNA treatment effectively reduces Ythdf1 protein levels.

6. Line 156 - 159, the conclusion about DF2 and DF3 in compensating DF1 loss is not supported by the data in Extended Data Figure 2b. In addition, this reviewer is not sure about how to explain the protein level compensation, as well as the overexpression in exacerbating the phenotype.

Thank you for pointing this out. A notable previous study demonstrated that knockdown of any individual YTHDF protein in HeLa cells led to compensatory upregulation of the remaining YTHDF paralogs, potentially masking the effects of single or double knockdowns (Cell, 2020, PMID: 32492408). This finding supports the notion of functional redundancy among YTHDF1/2/3 proteins. To explore this possibility in early embryonic development, we conducted RT-qPCR and Simple Western™ analyses to assess DF1/2/3 expression following single, double, and triple YTHDF knockdowns at the E2C and L2C stages. Our RT-qPCR results revealed no compensatory upregulation at the RNA level (Extended Figure 2a). However, at the protein level, DF2 and DF3 partially compensate for the loss of DF1 or DF1/2 in L2C embryos, whereas DF1 does not compensate for the loss of DF2, DF3, or DF2/3 (Extended Figure 2b). These findings suggest asymmetric compensatory dynamics among the YTHDF proteins during early embryogenesis.

Consistently, our LACE-seq datasets profiling DF1/2/3 binding across various developmental stages confirmed that, while all DF proteins share similar m⁶A recognition capabilities, their accessibility to m⁶A sites differs depending on the developmental stage (New Figure 3b and h). This developmental variation in m⁶A site accessibility likely explains why overexpression of any single DF protein cannot rescue the phenotype resulting from double DF knockdown and, in some cases, may even worsen it. These findings underscore the importance of stage-specific m⁶A site availability in defining the functional redundancy and specificity of DF proteins during early development.

7. The data of blastomeres in morula in Figure 1e is not consistent with that in Figure 1d, since no morula observed in KD embryo in 1d.

We apologize for the lack of clarity. We have updated the labeling of New Figure 1f to “No. of blastomeres (3.0 dpc)” and revised the corresponding sentence to read: “In addition, we observed another abnormal phenotype in *Ythdf* triple knockdown embryos—a notable increase in the number of blastomeres per embryo at 3.0 and 3.5 dpc.

8. Line 248, the conclusion that “the phenotype of Ythdf1-3 knockdown is dependent on m6A marks” can not be drawn from the data.

Thank you for pointing this out. We have now performed experiments involving si*Ythdf1-3* knockdown in embryos and successfully rescued the phenotype by treating si*Ythdf1-3* KD embryos with the METTL3 inhibitor STM2457. These results demonstrate that the phenotype resulting from si*Ythdf1-3* knockdown can be rescued by STM2457 treatment (New Extended Figure 4k and j). Based on these data, we conclude that the phenotype of si*Ythdf1-3* knockdown is dependent on m⁶A marks.

We have included the new data in the revised manuscript and removed the original Extended Figure 4k and j accordingly.

9. How is the overlap between DEGs in *Ythdf1-3* KD and STM2457 treated embryos? Thank you for pointing this out. We have analyzed the overlap of DEGs between *siYthdf1-3* knockdown and STM2457-treated embryos. Specifically, we identified 256 genes commonly upregulated in both conditions, including well-known 2C markers such as *Gm8994*, *Tdpoz2*, and *Tdpoz4*. Additionally, 216 genes were commonly downregulated, including *Trp53*, *Notch1*, and *Trp73* (New Extended Figure 4m). Notably, the common downregulated genes are enriched in the p53 signaling pathway, suggesting a potential mechanism for enhanced cell proliferation and accelerated blastomere division (Extended Figure 4o). We have incorporated these findings into the revised manuscript.

10. The difference of limited target redundancy in Extended Data Fig. 5g and reciprocal overlap of DF targets between two DF proteins reads confusing. Please describe it clearer to make it understandable to readers.

We apologize for the lack of clarity. In the original manuscript, we used the PureCLIP software package to identify DF1/2/3 binding peaks and calculated the overlap rates between any two DF binding peaks. While the overlap rates at the peak level are relatively low (3.2%-28.0%), the reciprocal overlap rates at the gene level are substantially higher, ranging from 23.6% to 72.2%. This difference arises because individual transcripts often contain multiple m⁶A sites, and DF1/2/3 binding peaks can localize to distinct m⁶A sites within the same transcript.

To more accurately address this point, we followed Reviewer #1's suggestion (please refer to our response to Major Question #2 from Reviewer #1) and analyzed the overlap of DF1/2/3 binding at m⁶A sites by directly comparing DF1/2/3 signals at putative m⁶A sites, rather than relying on called binding peaks. Our results showed a high correlation between DF proteins at m⁶A sites across early developmental stages (combined DF LACE-seq datasets from the E2C, L2C, and morula embryos), with Pearson's *R* values ranging from 0.48 to 0.70 (New Figure 3b). This suggests that DF proteins largely bind overlapping m⁶A sites during early embryogenesis. However, when analyzing the correlation around m⁶A sites at individual stages (E2C, L2C, or Morula), the correlation coefficients were more variable and relatively lower (*R* = 0.30 to 0.81) (New Figure 3b). This indicates that the accessibility of m⁶A sites to DF proteins differs depending on the developmental stage, likely due to differences in expression levels and subcellular localization of the DF proteins throughout early embryonic development. Therefore, we conclude that DF proteins have similar or identical m⁶A recognition capacities, but the accessibility of m⁶A sites to these proteins varies dynamically at individual stages (New Figure 3h). We have replaced the original Extended Data Fig. 5 g with New Figure 3b and incorporated this conclusion into the revised manuscript.

11. How is the overlap between DEGs in B2OE and *Ythdf1-3* KD embryos? Is that consistent with the claimed developmental defect in *Ythdf1-3* KD caused by B2OE?

Thank you for pointing this out. We have analyzed the overlap of DEGs between *Ythdf1-3* KD and B2 OE embryos. Specifically, we identified 191 common

upregulated genes, including well-known 2C markers such as *Gm4340*, *Tdpoz1/4/5*, *Zfp352*, and *Zscan4c*. Additionally, 205 common downregulated genes were found, including *Lrp1*, *Notch1*, and *Rgs14*, which are involved in developmental processes (New Extended Figure 8h). The number of common DEGs between B2 OE and *Ythdf1-3* KD embryos is comparable to that observed between STM2457-treated and *Ythdf1-3* KD embryos. Notably, the downregulated genes shared by STM2457-treated and *Ythdf1-3* KD embryos are associated with the p53 signaling pathway, suggesting a potential mechanism for enhanced cell proliferation and accelerated blastomere division. In contrast, no significant difference in blastomere number was observed between B2 OE embryos and controls at 3.5 dpc, and the common downregulated genes between B2 OE and *Ythdf1-3* KD embryos are related to developmental processes. These findings indicate that overexpression of B2 RNA partially recapitulates the molecular features seen in *Ythdf1-3* knockdown early embryos.

12. The developmental phenotype is similar between *Ythdf1-3* KD and B2ASO embryos, while the phenotype is largely alleviated in embryos with both *Ythdf1-3* KD and B2ASO. How to interpret this?

Thank you for pointing this out. Based on our B2 ASO knockdown (KD) and B2 overexpression (OE) results (Figure 4h and i, and Extended Figure 8a and b), we conclude that the normal expression of B2 RNAs is essential for early embryonic development. Both elevated and reduced levels of B2 RNAs lead to embryonic defects, likely due to dysregulation of RNA Pol II transcription. In si*Ythdf1-3* KD embryos, B2 RNA expression progressively accumulates during development, suggesting that excessive B2 RNA abundance may contribute to the observed embryonic defects. To test this, we simultaneously knocked down *Ythdf1-3* while reducing B2 RNA levels by injecting B2 ASO at a low concentration (0.2 μ M) to maintain B2 RNA near control levels (Figure 4j). Our results indicate that decreasing B2 RNA accumulation can rescue the *Ythdf1-3* KD phenotype, demonstrating that the phenotype is predominantly driven by B2 RNA accumulation.

13. The changes in PolIII binding might be a result of transcriptional changes, but not the reason. The link that increased B2 leads to decreased polIII binding and then decreased expression is not convincing to this reviewer.

Thank you for the thoughtful comment. Several landmark studies have demonstrated that B2 RNA directly binds to RNA polymerase II (Pol II) to repress transcription both *in vivo* and *in vitro*, particularly in mouse cells, thereby attenuating global mRNA synthesis in response to stress conditions such as heat shock (Cell, 2016, PMID: 27984727; PNAS, 2009, PMID: 19307572; Nat Struct Mol Biol, 2004, PMID: 15300239; Nat Struct Mol Biol, 2004, PMID: 15300240). Under such stress conditions, B2 RNA levels increase, allowing it to bind new targets and suppress RNA Pol II elongation activity. Building on these findings, we observed the following: 1) upon *DF1-3* knockdown, B2 RNA levels increase and prominently colocalize with RNA polymerase II in the nucleus, particularly at the morula stage (Figure 5a); 2) genes downregulated in both B2 OE and *DF1-3* KD embryos exhibit decreased Pol II occupancy across gene bodies or an elevated Pol II pausing index (defined as the ratio of Pol II density near the promoter to that within the gene body) (Figure 5c, d, and g);

3) these same genes show restored Pol II transcriptional elongation in B2 ASO rescue embryos compared to DF1-3 KD embryos (Figure 5i). Together, these results support the conclusion that elevated B2 RNA impairs Pol II elongation, contributing to the downregulation of target gene expression.

14. Figure 5i, how does this data imply that B2 RNA transcription promotes adjacent polIII transcription?

We apologize for the lack of clarity. We have analyzed changes in RNA Pol II CUT&Tag signals in siDF1-3 knockdown embryos compared to controls. The results revealed a significant reduction in Pol II signal near SINE/B2 loci (<50 kb), whereas regions located farther away (>50 kb) remained unaffected (Figure 6d). Interestingly, at the L2C stage, DF1-3 knockdown embryos showed reduced transcription of Pol III-driven B2 RNAs, which corresponded with diminished Pol II transcription near SINE/B2 loci (Figure 6b). In contrast, at the morula stage, B2 RNA transcription was mildly elevated in siDF1-3 knockdown embryos, leading to increased Pol II transcription around SINE/B2 loci (Figure 6b). These results support the conclusion that B2 RNA transcription can positively regulate adjacent Pol II transcription in a *cis*-acting manner.

15. Line 523 - 526, this reviewer is still confused by which model this study propose? What is the exact difference from the previous models?

We apologize for the lack of clarity. Please refer to our response to Major Question #3 from Reviewer #2. As previously reported, the cellular functions of the YTHDF proteins (YTHDF1, YTHDF2, and YTHDF3) remain a topic of active debate. The Canonical Model proposes that YTHDF1 enhances translation of its mRNA targets (Cell, 2015, PMID: 26046440), YTHDF2 promotes mRNA decay (Nature, 2014, PMID: 24284625), and YTHDF3 supports both translation and degradation (Cell Research, 2017, PMID: 28106072). In contrast, the Unified Model suggests that all three YTHDF proteins redundantly facilitate degradation of shared m⁶A-modified targets (Cell, 2020, PMID: 32492408; Genes & Development, 2020, PMID: 32943573). The central point of contention is whether YTHDF1, like YTHDF2 and YTHDF3, can promote degradation of its targets. In this study, we investigated whether YTHDF1 contributes to transcript degradation during mouse early embryonic development. Our findings demonstrate that YTHDF1, alongside YTHDF2 and YTHDF3, indeed facilitates degradation of its targeted transcripts. Furthermore, all three YTHDF proteins share similar m⁶A recognition capacity throughout early development, although the accessibility of specific m⁶A sites appears to differ at distinct developmental stages. Importantly, we also found that YTHDF1/2/3 proteins promote the degradation of retrotransposon RNAs, particularly B2 RNAs, across various stages of early embryogenesis.

Based on the results presented in the manuscript, we demonstrated that siDF1-3 KD leads to significant embryonic defects and widespread changes in gene expression. Our proposed model attributes these developmental abnormalities primarily to disrupted m⁶A regulation and the accumulation of B2 RNAs. Mechanically, the upregulated genes in siDF1-3 KD embryos are driven by the increased stability of m⁶A-marked RNAs, due to impaired degradation. In contrast, the downregulated genes are largely due to transcriptional repression mediated by the accumulation of B2 RNAs, which interfere with RNA polymerase II activity.

Reviewers' comments:

Reviewer #2 (Remarks to the Author):

The revised manuscript has substantially improved in clarity and presentation, and most of the previous concerns have been adequately addressed. The organization of the revision is coherent and easy to follow, reflecting the authors' thoughtful response to the reviewers' feedback. Notably, the inclusion of both Cas9-mediated knockout and CasRx-mediated knockdown of YTHDF proteins provides important orthogonal validation of the embryonic phenotypes initially observed with siRNA-mediated depletion. These complementary genetic approaches significantly strengthen the causal link between YTHDF function and the developmental phenotypes reported.

In addition, the updated immunostaining and RNA-FISH analyses offer a more nuanced view of the subcellular distribution of B2 RNAs during embryogenesis. The authors now appropriately emphasize both the cytoplasmic and nuclear localization patterns, which are critical for interpreting the functional interplay between YTHDF proteins and B2 transcripts. The expanded discussion on the interaction between cytoplasmic YTHDFs and B2 RNAs adds further mechanistic insight and enhances the overall depth of the study.

Taken together, these revisions improve the rigor and clarity of the manuscript and elevate the quality of the work.

Our Response: We thank this reviewer for the warm and constructive comments throughout the review process, and we sincerely hope this report could help recognize the regulated mechanism of Ythdf in early embryos.

Reviewer #3:

Many of my concerns have not been addressed. I have serious concerns about this manuscript. I describe several of them below. Therefore, this manuscript is not suitable for publication in *[journal name redacted]*.

For my previous comment 4, I did not buy the response. For the current study, cKO mouse models will no doubt fully give more clean results with minimal off-target and dosage effect compared to KD models. In addition, siRNA mediated KD starting in 1C did not give temporal advantage. Rather, it makes the system more complicated by maternal protein carry over from oocytes.

Our Response: We acknowledge the advantages of cKO mouse models in providing cleaner results with minimal off-target and dosage effects compared to KD models. While previous studies have shown that Ythdf2 cKO leads to embryonic arrest at the 2-cell stage and Ythdf1/2/3 triple KO results in complete lethality before postnatal day 30, obtaining early knockout embryos (such as YTHDF1-3 KO morula embryos) for studying Ythdf1/2/3 protein functions during early embryonic development remains biologically unfeasible. Our research focus is not on investigating the impact of maternal or ZGA-derived Ythdf1/2/3

proteins, but rather on investigating the regulatory mechanisms of Ythdf1/2/3 proteins in early embryos, with particular emphasis on the post-major zygotic genome activation (ZGA) period (specifically from the L2C stage onward). In this context, Ythdf KD models offer distinct advantages by enabling stage-specific functional analysis during preimplantation development, particularly from the L2C to morula stages, where effective protein knockdown can be achieved and embryos remain viable for experimental analysis. Furthermore, our study incorporates multiple orthogonal approaches, such as Cas9-mediated knockout and CasRx-mediated knockdown. These methods provide independent validation of the embryonic phenotypes initially observed with siRNA-mediated depletion, thereby strengthening the causal relationship between Ythdf protein function and the observed developmental outcomes.

For my previous comment 5, my concern has not been addressed by the response. Repeating the statement in the manuscript did not solve the question. How can siDF1 affect DF1 protein level in E2C stage? There is minimal RNA and translation of DF1 before L2C stage. This makes the reviewer seriously doubt about the validity of the data in this manuscript.

Our Response: While Ythdf1 exhibits relatively lower mRNA expression and translation levels compared to Ythdf2 and Ythdf3, we have consistently detected its presence in early embryos through multiple orthogonal methods, including RNA-seq, RT-qPCR, Western blot, and immunofluorescence. We carry out Ythdf1 siRNA microinjection at the zygote stage, which is ~18 h before the E2C stage; thus, the Ythdf1 protein is highly likely to be knocked down at the E2C stage. Our RT-qPCR results showed that significant *Ythdf1* mRNA reduction at the E2C stage (Extended Data Fig.2a), and the Simple Western and immunofluorescent results showed moderate Ythdf1 protein knockdown at E2C stage and marked Ythdf1 protein depletion at the L2C stage (Extended Data Fig.1e and Extended Data Fig.2b). These findings provide conclusive evidence that siDF1 effectively reduces DF1 protein levels by the E2C stage. More importantly, even considering the theoretical possibility of incomplete DF1 knockdown at E2C, this would not impact our core findings because the study primarily examines Ythdf protein functions at L2C and morula stages, and all three Ythdf proteins (DF1/2/3) show robust knockdown at these stages.

For my previous comment 6, my concern has not been addressed by the response. The compensation may arise from RNA levels of close homologs. However, in the case in the current manuscript, the RNA level remains unchanged. Then, how can the protein level make a dramatic change? Where was the protein from? This is theoretically not understandable.

Our Response: Our results demonstrate compensatory upregulation occurring at the DF protein level, but not at the RNA level, before the L2C stage. This observation aligns with the known biological context that transcriptional activity remains largely quiescent before L2C in mouse embryos. Consequently, single or double DF knockdowns would not be expected to induce compensatory upregulation of the remaining DF paralog mRNAs, as evidenced by our experimental findings. Regarding the observed protein-level compensation, we propose several potential regulatory mechanisms that may account for

this phenomenon: 1) Translational control of existing DF mRNAs; 2) Post-translational modifications affecting protein stability; 3) Feedback regulation through signaling pathways; 4) Alterations in protein degradation kinetics. These mechanisms could collectively modulate DF paralog protein expression in response to individual DF protein depletion, providing a plausible explanation for the compensatory effects observed at the protein level.

For my previous comment 7, I did not see the change mentioned in the response "We have updated the labeling of New Figure 1f to "No. of blastomeres (3.0 dpc)". The original label is like this." More importantly, my concern is not responded at all. My concern is that there is no morula mentioned in original Figure 1d for siDF1-3 embryos, with embryos reaching 5-16 cells at most. How can the quantification in original Figure 1e with many data points more than 16 cells? This also makes the reviewer seriously doubt about the validity of the data in this manuscript.

Our Response: To clarify, the apparent discrepancy between the original Figures 1d and 1e stems from our use of independent sample batches for these analyses. In the original Figure 1d, we captured developmental progression from 2-cell to blastocyst stages and counted the embryo population at each developmental time point. In the original Figure 1e, we performed immunofluorescence analysis to count blastomeres at specific time points (dpc 3.0 and 3.5) by using separate embryo batches. This experimental design explains the observed variation in developmental rates between batches. Specifically, siDF1-3 embryos in Figure 1d primarily reached 5-16 cells, and siDF1-3 embryos in Figure 1e predominantly attained the 16-cell stage. In any case, siDF1-3 treatment increases blastomere number per embryo compared to same-batch controls.

For my previous comment 8, my concern has not been addressed by the response. The siDF1-3 led to decreased blastocyst rate, and the STM2457 treatment led to decreased blastocyst rate. How can combined treatment leads to normal development comparable to normal WT control? If m6A binding by Ythdfs is important to their function, we'd expect that the absence of m6A shall give similar phenotype to the absence of m6A together with siDF1-3. This also makes the reviewer seriously doubt about the validity of the data in this manuscript.

Our Response: While both siDF1-3 and STM2457 treatments similarly reduce blastocyst rates, our data suggest distinct regulatory mechanisms based on partial overlap between DEGs in siDF1-3 versus STM2457 treatments (Extended Data Fig.4m). And MeRIP-seq and LACE-seq data demonstrate that DF1/2/3 proteins bind only a subset of m6A sites (Extended Data Fig.6a and b). We agree with the theoretical expectation that the absence of m6A shall give a similar phenotype to the absence of m6A together with siDF1-3. While the phenotype of siDF1-3 should differ from the combined absence of m6A with siDF1-3, what is more, the absence of m6A together with siDF1-3 can partially rescue the phenotype of siDF1-3. Because the m6A-marked RNAs normally targeted by DF1-3 escape proper degradation in siDF1-3 embryos, and these RNAs may be aberrantly regulated by alternative m6A readers (e.g., IGF2BPs). While STM2457-treated siDF1-3 embryos will deplete m6A sites and eliminate the ill effects of potentially harmful misregulated RNAs recognized by DF1-3 proteins. Our results (Extended Data Fig.4k) clearly showed that the STM2457-treated siDF1-3 embryos could partially rescue the developmental phenotype of siDF1-3.

Dear Dr Su,

Thank you very much for submitting your manuscript to The EMBO Journal. Given that the manuscript was previously reviewed at a different journal we had decided to ask an arbitrating referee to comment on the manuscript, the referee comments and your response.

As you can see from the report, this referee thinks that the concerns raised during the last round of revisions at the previous journal while being justified can be addressed by appropriately acknowledging the limitations of the study. All together this referee however does not agree that the manuscript can convincingly argue that there are functionally relevant differences between the binding patterns of the three YTHDF proteins and thinks that the proposed stage-specific differences would need additional evidence and with the current experimental evidence could be caused by technical artifacts or by other mechanisms such as PTMs which were not investigated here.

On the other hand, this referee thinks that the identification of B2 elements as a functionally important target of YTHDF proteins is interesting and should be emphasized more throughout the manuscript.

Taken together, we think that the assessment all together supports publication of the manuscript, and we would therefore ask you to revise the manuscript following the suggestion by the arbitrating referee.

Do not hesitate to contact us if you have any additional questions.

With best regards,

Cornelius Schneider

Cornelius Schneider, PhD
Editor
The EMBO Journal
c.schneider@embojournal.org

For more details on our Transparent Editorial Process, please visit our website:
<https://www.embopress.org/page/journal/14602075/authorguide#transparentprocess>

Please remember: Digital image enhancement is acceptable practice, as long as it accurately represents the original data and conforms to community standards. If a figure has been subjected to significant electronic manipulation, this must be noted in the figure legend or in the 'Materials and Methods' section. The editors reserve the right to request original versions of figures and

the original images that were used to assemble the figure.

We realize that it is difficult to revise to a specific deadline. In the interest of protecting the conceptual advance provided by the work, we recommend a revision within 3 months (30th Dec 2025). Please discuss the revision progress ahead of this time with the editor if you require more time to complete the revisions. Use the link below to submit your revision:

Referee #1:

I think the manuscript is improved, and some of the weaknesses raised by reviewer 3 can be acknowledged as limitations of the study by the authors.

Comments:

1. The abstract and the text uses the phrase "with m⁶A site accessibility to individual Ythdf proteins varying across different developmental stages" Is unclear. The authors say they have found that the Ythdf proteins have highly similar binding patterns and thus they bind to the same sites. This should be clearly stated. But the authors also state that there are stage-specific changes in binding to specific sites (but not specific Ythdf proteins). This can be stated, and likely reflects competing RNA-binding protein expression. The abstract should make the clear point of overall similarity in binding patterns of all three YTHDF proteins. If there are differences between the Ythdf isoforms at a specific stage at a specific site, these sites should be named and tested biochemically, as mentioned earlier. If there are differences, it is likely to be outliers in sample-to-sample variability.
2. Related to this, the authors state "However, when analyzing the correlation around m6A sites at individual stages (E2C, L2C, or Morula), the correlation coefficients were more variable and relatively lower (R = 0.30 to 0.81) (Figure 3b)." The correlation between datasets is related to the coverage level for each dataset. Are the datasets equally deep and have the same quality? Metrics should be shown, and if the datasets are not equivalent in depth, they can be downsampled to make equal comparisons between stages. Correlates between biological replicates should be reported and compared based on read depth. I think this will show that variability even exists just for biological replicates. It is critical to make sure depth of each sample is discussed and labeled in figures since variations in depth could cause these effects.
3. The authors should keep in mind that the Ythdf proteins have considerable PTMs, with glycosylation appearing to be particularly important which regulates their functions. It is possible that the Ythdfs may have different modification levels at different stages in development which affects whether they can compensate efficiently. Additionally it has been reported that expressing the Ythdf proteins is difficult due to aggregate formation (liquid phase separation). This can make a protein appear non-functional since the protein becomes aggregated. These should at least be mentioned as alternatives for the inability of YTHDF1 to compensate.
4. The statement "These findings suggest that DF-mediated regulation B2 elements may be specific to the mESC context" does not seem well founded to me since no other mouse cell type was tested. How do we know that DFs only regulate B2 in mESC and not other mouse cell types?
5. I recommend that the authors consider emphasizing the retroelement/B2 part of their story more in the abstract and title since this is the most interesting/new part of the study

Reviewers' Comments:

We sincerely thank Referee #1 for the insightful and constructive comments that have helped improve our manuscript. In the following pages, we provide a point-by-point response to all specific comments. The reviewer's comments are presented in blue, and our corresponding responses are shown in black. In addition, all revisions in the manuscript text file are highlighted in red.

Referee #1:

I think the manuscript is improved, and some of the weaknesses raised by reviewer 3 can be acknowledged as limitations of the study by the authors.

We are very grateful to the Referee for carefully reviewing our manuscript and providing insightful comments. We have revised and improved the manuscript accordingly.

Comments:

1. The abstract and the text uses the phrase "with m⁶A site accessibility to individual Ythdf proteins varying across different developmental stages" Is unclear. The authors say they have found that the Ythdf proteins have highly similar binding patterns and thus they bind to the same sites. This should be clearly stated. But the authors also state that there are stage-specific changes in binding to specific sites (but not specific Ythdf proteins). This can be stated, and likely reflects competing RNA-binding protein expression. The abstract should make the clear point of overall similarity in binding patterns of all three YTHDF proteins. If there are differences between the Ythdf isoforms at a specific stage at a specific site, these sites should be named and tested biochemically, as mentioned earlier. If there are differences, it is likely to be outliers in sample-to-sample variability.

Fig. R1A. Venn diagram showing the overlap between m⁶A peaks identified from 2C picoMeRIP-seq datasets and m⁶A sites identified from mESC GLORI datasets.

Fig. R1B. Venn diagram showing the overlap between m⁶A peaks identified from 8C picoMeRIP-seq datasets and m⁶A sites identified from mESC GLORI datasets.

New **Fig. 3G.** Density plots showing the pairwise comparison of normalized Ythdf1/2/3 binding signals (LACE-seq reads) surrounding individual m⁶A peaks (picoMeRIP-seq

datasets from 2C and 8C embryos) across the E2C, L2C, morula, and combined developmental stages. The DF1/2/3 LACE-seq datasets from E2C and L2C embryos were referenced against the m⁶A peaks identified in the 2C picoMeRIP-seq datasets; the DF1/2/3 LACE-seq datasets from morula embryos were referenced against the m⁶A peaks identified in the 8C picoMeRIP-seq datasets. The combined DF1/2/3 LACE-seq datasets from E2C, L2C, and morula embryos were compared with the combined m⁶A peaks from the 2C and 8C picoMeRIP-seq datasets.

Fig. R1C. Representative images of IF/RNA-FISH for mRNA with DF1/2 counterstain in L2C embryos. Scale bars, 20 μ m. Fluorescence intensity traces from the image are plotted at right. Black arrowheads indicate DF puncta colocalize with mRNA signals.

We appreciate the referee's insightful suggestion. In response, we have clarified our findings in the abstract and performed additional experiments and analyses to address the concern regarding the paradoxical conclusion that stage-specific binding of Ythdf isoforms might arise from improper use of the mESC GLORI datasets. Specifically, in the original manuscript, we analyzed the correlations among Ythdf isoforms at individual developmental stages (E2C, L2C, and Morula) using m⁶A sites referenced from the mESC GLORI datasets. Based on this analysis, we concluded that stage-dependent differences exist among Ythdf isoforms at specific m⁶A sites. We have now carefully re-evaluated the mESC GLORI datasets together with publicly available mouse MeRIP-seq datasets from the 2C and 8C stages. We found that only 18% of m⁶A peaks from the 2C MeRIP-seq dataset and 28% from the 8C MeRIP-seq dataset overlap with those identified in the mESC GLORI dataset (**Fig. R1A, B**), indicating that the m⁶A methylome of early embryos differs substantially from that of mESCs. Therefore, the m⁶A sites from the mESC GLORI datasets are not sufficiently accurate as a reference for comparing Ythdf1/2/3 binding in early embryos. Accordingly, we reanalyzed the correlations among Ythdf isoforms at each developmental stage (E2C, L2C, and Morula) using m⁶A peaks derived from the 2C and 8C MeRIP-seq datasets.

This updated analysis showed a substantial improvement in the correlation coefficients between the Ythdf isoforms, both at individual developmental stages and when the stages were combined (New **Fig. 3G**). In addition, we performed RNA-FISH/IF co-staining of the putative DF1-unique target *Arg1* mRNA with DF1 and DF2 proteins, as well as the putative DF2-unique target *Tmem60* mRNA with DF1 and DF2 proteins at the L2C stage, to assess whether DF proteins exhibit stage-specific binding to particular transcripts. We observed that both DF1 and DF2 colocalize with *Arg1* and *Tmem60* mRNAs (**Fig. R1C**), indicating that the putative DF-unique binding sites are likely outliers resulting from sample-to-sample variability rather than true biological specificity. Taken together, we have now clearly stated in the abstract (Lines 43-45) that “we demonstrate that all three Ythdf proteins mediate the decay of their target transcripts by binding to similar m⁶A sites, including maternal mRNAs, mid-preimplantation-activated transcripts, and retrotransposon RNAs.” We have also updated the correlation coefficients among Ythdf isoforms (new Fig. 3G) and removed the previous statement that “m⁶A site accessibility to individual Ythdf proteins varies across developmental stages.”

2. Related to this, the authors state "However, when analyzing the correlation around m6A sites at individual stages (E2C, L2C, or Morula), the correlation coefficients were more variable and relatively lower (R = 0.30 to 0.81) (Figure 3b)." The correlation between datasets is related to the coverage level for each dataset. Are the datasets equally deep and have the same quality? Metrics should be shown, and if the datasets are not equivalent in depth, they can be downsampled to make equal comparisons between stages. Correlates between biological replicates should be reported and compared based on read depth. I think this will show that variability even exists just for biological replicates. It is critical to make sure depth of each sample is discussed and labeled in figures since variations in depth could cause these effects.

Stage	Sample	Used reads	Pairwise of DFs	Original correlation coefficients (R)	Downsample (R)	Refer to MeRIP (R)
E2C	DF1_rep1	1,302,704	DF1-DF2	0.8608	0.8601 ± 0.0008	0.9141 ± 0.0011
	DF1_rep2	1,083,236				
E2C	DF2_rep1	6,933,063	DF2-DF3	0.4748	0.4747 ± 0.0005	0.8950 ± 0.0004
	DF2_rep2	7,017,721				
E2C	DF3_rep1	1,394,804	DF3-DF1	0.3801	0.3801 ± 0.0015	0.8821 ± 0.0019
	DF3_rep2	1,038,503				
L2C	DF1_rep1	894,583	DF1-DF2	0.7098	0.7076 ± 0.0033	0.7065 ± 0.0029
	DF1_rep2	849,914				
L2C	DF2_rep1	3,532,964	DF2-DF3	0.5298	0.5297 ± 0.0008	0.5380 ± 0.0011
	DF2_rep2	4,236,165				
L2C	DF3_rep1	2,781,399	DF3-DF1	0.3144	0.3131 ± 0.0045	0.8006 ± 0.0022
	DF3_rep2	3,317,315				
Morula	DF1_rep1	1,520,269	DF1-DF2	0.8299	0.8298 ± 0.0008	0.8858 ± 0.0014
	DF1_rep2	1,173,811				
Morula	DF2_rep1	6,316,160	DF2-DF3	0.6881	0.6881 ± 0.0000	0.6268 ± 0.0008
	DF2_rep2	6,617,866				
Morula	DF3_rep1	5,387,558	DF3-DF1	0.5313	0.5312 ± 0.0004	0.5851 ± 0.0046
	DF3_rep2	5,248,610				
Combined	DF1	6,824,517	DF1-DF2	0.9900	0.9900 ± 0.0000	0.9018 ± 0.0006
Combined	DF2	34,653,939	DF2-DF3	0.7264	0.7264 ± 0.0001	0.6252 ± 0.0016
Combined	DF3	19,221,248	DF3-DF1	0.6026	0.6026 ± 0.0006	0.6252 ± 0.0005

Table R1. Statistical analysis for DF LACE-seq datasets. The correlation coefficients

referred to MeRIP-seq datasets are labeled in red.

We thank Referee #1 for this insightful and constructive suggestion. Following the reviewer's recommendation, we reanalyzed the correlations among Ythdf isoforms at individual stages (E2C, L2C, or Morula) as well as across combined stages by down-sampling all datasets to equal read depth across biological replicates. The results showed that the correlations among Ythdf isoforms remained largely unchanged compared to the original analysis when using m⁶A sites derived from the mESC GLORI datasets. As noted in Comment #1, only 18% of the m⁶A peaks from 2C MeRIP-seq datasets and 28% of the peaks from 8C MeRIP-seq datasets overlap with the mESC GLORI datasets (**Fig. R1A, B**), indicating that the methylome of early embryos is substantially different from that of mESCs. Consequently, m⁶A sites from the mESC GLORI datasets are not suitable as a reference for evaluating Ythdf1/2/3 binding in early embryos. We therefore analyzed the correlation of Ythdf isoforms at individual stages (E2C, L2C, or Morula) using m⁶A peaks from 2C and 8C MeRIP-seq datasets. This analysis revealed a substantial improvement in the correlation coefficients between Ythdf isoforms at both individual and combined stages (New **Fig. 3G; Table R1**). We have incorporated this update in the revised manuscript (Lines 296, 343–351) and annotated the sequencing depth of each sample in **Table EV2**.

3. The authors should keep in mind that the Ythdf proteins have considerable PTMs, with glycosylation appearing to be particularly important which regulates their functions. It is possible that the Ythdfs may have different modification levels at different stages in development which affects whether they can compensate efficiently. Additionally it has been reported that expressing the Ythdf proteins is difficult due to aggregate formation (liquid phase separation). This can make a protein appear non-functional since the protein becomes aggregated. These should at least be mentioned as alternatives for the inability of YTHDF1 to compensate.

We appreciate the referee's insightful suggestion and have expanded the discussion accordingly. Specifically, we now highlight that PTMs and liquid-liquid phase separation (LLPS) of DF proteins may influence their compensatory efficiency during early embryonic development. This discussion has been incorporated into the Results and Discussion section (Lines 190-197 and 620-630) with appropriate citations.

4. The statement "These findings suggest that DF-mediated regulation B2 elements may be specific to the mESC context" does not seem well founded to me since no other mouse cell type was tested. How do we know that DFs only regulate B2 in mESC and not other mouse cell types?

We sincerely apologize for the confusion. We have clarified as follows: "These findings suggest that DF-mediated regulation of B2 elements may be integral to the mESC and mouse early embryo context"(Line 406-408).

5. I recommend that the authors consider emphasizing the retroelement/B2 part of their story more in the abstract and title since this is the most interesting/new part of the study. We thank the referee for the positive comments and constructive suggestions. We have emphasized DF-mediated regulation of B2 elements in both the manuscript title, "m⁶A reader Ythdf proteins control B2 repeats expression and guard early embryo development", and the abstract: "In parallel, Ythdf1-3 deficiency represses Pol III-

driven B2 transcription, which in turn modulates RNA Pol II activity at genomic regions adjacent to B2 loci. Altogether, the coordinated regulatory axis of Ythdf-SINE/B2-gene expression governs a broad transcriptional network that is crucial for embryogenesis” (Lines 48-52).

Dear Dr. Su,

Thank you for submitting a revised version of your manuscript. We think that the revisions adequately address the concerns raised by the arbitrating referee. There remain only a few mainly editorial points that have to be addressed before I can extend formal acceptance of the manuscript:

- 1) Please remove figures and track changes from the manuscript file
- 2) AFFILIATIONS (research institution or university vs. biotech company): employment in a biotech company should be stated in DCIS
- 3) FUNDING INFO: missing info in eJP: the National Natural Science Foundation of China (32230028, 32300952; 32471171, and 81971357 and 82271728), and the Science and Technology Program of Guangzhou, China (2023A03J0258)
- 4) On the abstract page of the manuscript, please include 4-5 general keyword terms to enhance searchability.
- 5) Please adjust the format of the reference list and of the in-text citations according to EMBO Journal format (alphabetical order, author name et al + year.../up to 10 author names in the reference list before et al / please refer to our Guide to Authors for additional information on EMBO J reference format).
- 6) Please rename the Conflict of Interest section into "Disclosure and Competing Interests Statement", in accordance with our updated Guide to Authors (<https://link.springer.com/partners/embo-press/editorial-policies#Competing%20interest%20disclosures>)
- 7) As we are switching from a free-text author contribution statement towards a more formal statement based on Contributor Role Taxonomy (CRediT) terms, please remove the present Author Contribution section and instead specify each author's contribution(s) directly in the Author Information page of our submission system during upload of the final manuscript. See <https://casrai.org/credit/> for more information.
- 8) FIGURE CALLOUTS: there is a callout for the missing Supplementary Table 1; missing callout for Fig. 4J
- 9) Please provide the missing response in "Experimental animals" section in the authors checklist
- 10) DATASET EV LEGENDS: legends for Datasets should be added as a separate tab/sheet in each Excel file (legends for EV Tables are correctly placed above the table in Excel files)
- 11) Thank you for providing the source data. Could you please ZIP the SD for EV and appendix figures in one folder.
- 12) Please provide suggestions for a short 'blurb' text prefacing and summing up the conceptual aspect of the study in two sentences (max. 250 characters), followed by 3-5 one-sentence 'bullet points' with brief factual statements of key results of the paper; they will form the basis of an editor-written 'Synopsis' accompanying the online version of the article. Please also provide an altered synopsis image, making sure that the aspect ratio conforms to our website's format - it should be exactly 550 pixels wide and between 300-600 pixels high.
- 13) Please note that the exact p values are not provided in the legends of figures 1E, F, G, H, I, J, K; 2E, 4F, I, J; S1B, D, F; S2 G, H, I, K, P; S3D, S4B, C; EV1 D, F, G, H, K, L, M, N, P, Q; EV2 C, K, L; EV5 A, B, E, G, H
- 14) Please indicate the statistical test used for data analysis in the legends of figures 2B, C; 3E, I, J, K; 4D, F, G; 5C, D, F, I; 6B, D; S2Q, S3F, S4F, G; S5C, D; EV2 F, N; EV4 I-L
- 15) Please note that the box plots need to be defined in terms of minima, maxima, centre, bounds of box and whiskers, and percentile in the legends of figures 1A, 6C, S2 K, S4 J, S5 H, EV4 G, H
- 16) Please note that information related to n is missing in the legends of figures 1A, F, G, K; 2F, 6C, S1 B, D, F; S2K, M, N; S4 J, S5 H, EV1 F-H, L-N; EV2 H, I; EV4 C, E, G, H
- 17) Please note that the error bars are not defined in the legends of figures 2F, S2 M, N; S4 L, M; EV2 H
- 18) Please note that the dotted borders are not defined in the legend of figures 1F, G, K. This needs to be rectified.

19) Sections need to be named and the order should be corrected: Title page - Abstract - Keywords - Introduction - Results - Discussion - Methods - Data Availability - Acknowledgements - Disclosure and Competing Interests Statement - References - Figure Legends - Table(s) - Expanded View Figure Legends.

With best regards,

Cornelius Schneider

Cornelius Schneider, PhD
Editor | The EMBO Journal
c.schneider@embojournal.org

Please refer to our figure preparation guideline in order to ensure proper formatting and readability in print as well as on screen:

<https://link.springer.com/journal/44318/submission-guidelines#cms-Figure-and-data-presentation>

All minor editorial requests have been addressed by the authors.

Dear Dr. Su,

I am pleased to inform you that your manuscript has been accepted for publication in the EMBO Journal.

You may qualify for financial assistance for your publication charges - either via a Springer Nature fully open access agreement or an EMBO initiative. Check your eligibility: <https://link.springer.com/journal/44318/how-to-publish-with-us>

Yours sincerely,

Cornelius Schneider, PhD
Editor
The EMBO Journal
c.schneider@embojournal.org

Please note that it is The EMBO Journal policy for the transcript of the editorial process (containing referee reports and your response letters) to be published as an online supplement to each paper. If you should prefer removal of any referee-only figures included in the point-by-point response(s), e.g. because they may still be used for future publication or because they have been reproduced from published work by others, please do let us know immediately via response email.

More information is available here: <https://link.springer.com/partners/embo-press/editorial-policies#Peer%20review>